# A multicolor suite for deciphering population coding of calcium and cAMP in vivo

**Tatsushi Yokoyama** [1,2,3,4] ✉, **Satoshi Manita** [5], **Hiroyuki Uwamori** [6], **Mio Tajiri** [7], **Itaru Imayoshi** [2,3,4], **Sho Yagishita** [7], **Masanori Murayama** [6], **Kazuo Kitamura** [5] & **Masayuki Sakamoto** [1,2,3,4,8] ✉

cAMP is a universal second messenger regulated by various upstream pathways including Ca²⁺ and G-protein-coupled receptors (GPCRs). To decipher in vivo cAMP dynamics, we rationally designed cAMPinG1, a sensitive genetically encoded green cAMP indicator that outperformed its predecessors in both dynamic range and cAMP affinity. Two-photon cAMPinG1 imaging detected cAMP transients in the somata and dendritic spines of neurons in the mouse visual cortex on the order of tens of seconds. In addition, multicolor imaging with a sensitive red Ca²⁺ indicator RCaMP3 allowed simultaneous measurement of population patterns in Ca²⁺ and cAMP in hundreds of neurons. We found Ca²⁺-related cAMP responses that represented specific information, such as direction selectivity in vision and locomotion, as well as GPCR-related cAMP responses. Overall, our multicolor suite will facilitate analysis of the interaction between the Ca²⁺, GPCR and cAMP signaling at single-cell resolution both in vitro and in vivo.

cAMP is a universal second messenger in a variety of cell types and organisms. Individual cell types express various adenylate cyclases (ACs) and phosphodiesterases that synthesize and degrade cAMP, respectively. The upstream regulators of the ACs include GPCRs and another central second messenger, Ca²⁺, through Ca²⁺-dependent ACs[1]. cAMP is regulated by these multiple upstream signaling pathways and modulates diverse cellular functions through cAMP-dependent kinases, channels and transcription factors. Therefore, technologies to visualize the spatiotemporal dynamics of cAMP in vivo are crucial for biological research in various organs or species.

Since cAMP was visualized in 1991, more than 50 cAMP indicators have been developed[2,3], and recently, several cAMP indicators have been used in vivo[4–8]. The circularly permuted green fluorescent protein (cpGFP)-type cAMP indicators have been intensively developed more recently due to their large dynamic range[6,7,9–11]. However, the use of these indicators has been limited due to their cAMP affinity (>1 μM), which is lower than that of endogenous cAMP-dependent kinases and channels (typically in the hundreds of nanomolar range)[12,13].

Here, we present cAMPinG1, a green cAMP indicator with a high dynamic range and more than 4-fold higher cAMP affinity than the existing green cAMP indicators. We also introduce RCaMP3, an improved red calcium indicator. Dual-color imaging of RCaMP3 and cAMPinG1 revealed dynamic interaction and information flow between Ca²⁺, cAMP

[1]Department of Optical Neural and Molecular Physiology, Graduate School of Biostudies, Kyoto University, Kyoto, Japan. [2]Center for Living Systems Information Science, Graduate School of Biostudies, Kyoto University, Kyoto, Japan. [3]Department of Brain Development and Regeneration, Graduate School of Biostudies, Kyoto University, Kyoto, Japan. [4]Laboratory of Deconstruction of Stem Cells, Institute for Frontier Life and Medical Sciences, Kyoto University, Kyoto, Japan. [5]Department of Neurophysiology, Graduate School of Medicine, University of Yamanashi, Chuo, Yamanashi, Japan. [6]Laboratory for Haptic Perception and Cognitive Physiology, Center for Brain Science, RIKEN, Wako, Saitama, Japan. [7]Department of Structural Physiology, Graduate School of Medicine, The University of Tokyo, Tokyo, Japan. [8]Precursory Research for Embryonic Science and Technology (PRESTO), Japan Science and Technology Agency, Kyoto, Japan. ✉e-mail: yokoyama.tatsushi.2w@kyoto-u.ac.jp; sakamoto.masayuki.2e@kyoto-u.ac.jp

and GPCRs in vivo. In addition, we demonstrated the application of cAMPinG1 imaging in cultured cells for studying GPCR biology.

## Results

### Rational engineering of a sensitive cAMP sensor

To develop a high-affinity cAMP sensor, we chose a mammalian protein kinase A regulatory subunit (PKA-R) as the cAMP-sensing domain. PKA-Rs are widely distributed and functional in mammalian neurons even when tagged by GFP and overexpressed[14] and have been well characterized in terms of their biochemical, evolutionary and structural properties[13,15–17]. We used the cAMP-binding domain A of mammalian PKA-R type 1α (PKA-R1α) because of its affinity for cAMP (around 150 nM)[18], which was within the range of the affinity of cAMP-binding domains in multiple PKAs and cyclic nucleotide-gated ion channels[7,8] (Extended Data Fig. 1a). We inserted cpGFP into the β4–β5 loop of PKA-R1α for the following reasons: (1) it is close to cAMP in the cAMP-bound three-dimensional structure, (2) it is exposed on the surface, and (3) it is structurally flexible, as demonstrated by the crystal structure analysis and evolutionarily conserved sequences[15,19], to avoid possible structural perturbations[20] (Fig. 1a and Extended Data Fig. 1b,c). We also removed the N-terminal PKA-R1α region to avoid interaction with endogenous proteins (Extended Data Fig. 1a). Instead, we fused the RSET sequence to the N terminus of the sensor to promote stable expression[19]. To improve dynamic range ($\Delta F/F$), we then screened over 250 mutants with mutations on the putative interface between the cpGFP and cAMP-binding domain, including two linkers and residues in cpGFP close to the interface. The variant with the largest $\Delta F/F$ was named cAMP indicator green 1 (cAMPinG1; Fig. 1b). Excitation spectra of cAMPinG1 indicated green fluorescence intensity of cAMPinG1 increased by 1,000% upon binding to cAMP with blue light (488 nm), while the green fluorescence intensity decreased by 61% with violet light (405 nm), resulting in a 2,700% ratio change with a combination of blue and violet excitation in HEK293T cell lysate (Fig. 1c). In contrast, G-Flamp1 showed less fluorescence decrease with violet excitation[7] (Extended Data Fig. 2a).

We next developed a cAMP-insensitive indicator (cAMPinG1mut (inactive mutant)) by introducing the p.Arg211Glu (in the numbering of mouse PKA-R1α) mutation to block cAMP binding (Fig. 1d). To detect cAMP changes selectively in the cytoplasm, we added a self-cleaving peptide (F2A) to the C termini of cAMPinG1 (named nuclear-excluded cAMPinG1 (cAMPinG1-NE)[21] (Fig. 1d). Furthermore, to avoid contamination of somatic neuropil fluorescence signals, we linked ribosomal subunit protein (RPL10) to the C terminus of cAMPinG1 as soma-targeting (named cAMPinG1-ST)[22] (Fig. 1d).

### In vitro characterization of cAMPinG1

Side-by-side comparison revealed that cAMPinG1 had the largest $\Delta F/F$ at 490 nm of excitation and fluorescence intensity in a cAMP-saturated state in HEK293T cell lysate than the existing cAMP sensors, Flamindo2, gCarvi and G-Flamp1 (refs. 7,9,23; Fig. 1e,

Extended Data Fig. 2a and Supplementary Table 1). The $\Delta F/F$ of cAMPinG1 at 450 nm of excitation was less than that of G-Flamp1, indicating that cAMPinG1 was tuned to the 470–490 nm of excitation, frequently used for green fluorescence imaging (Fig. 1e). cAMPinG1mut did not respond to cAMP, confirming that the fluorescence change of cAMPinG1 depended on cAMP binding. The $K_d$ value of cAMPinG1 was 181 nM, less than a quarter of those of Flamindo2, gCarvi and G-Flamp1 (Fig. 1f,g). Furthermore, we validated the linear relation of cAMP concentration and cAMPinG1 fluorescence intensity, low affinity to cGMP ($K_d = 12.2$ μM), in vitro properties of cAMPinG1-NE, pH sensitivity, in vitro kinetics and undetectable binding to PKA-C (Extended Data Fig. 2c–j and Supplementary Table 2).

As expected by excitation spectra (Fig. 1c), time-lapse imaging for HEK293T cells showed an increase in green fluorescence of cAMPinG1 with blue (488 nm) excitation and decreased with violet (405 nm) excitation in response to the application of forskolin (FSK), an activator of ACs (Fig. 1h–j). While cAMPinG1 demonstrated a high $\Delta F/F$ (~400%) for intensiometric measurement with 488 nm of excitation, ratiometric measurement had an even higher $\Delta F/F$ (~800%). Moreover, cAMPinG1 showed a fluorescence lifetime increase from 2.461 to 2.644 in response to forskolin (Extended Data Fig. 2k).

In acute brain slices, the elevation of cAMPinG1-NE fluorescence by forskolin/isobutylmethylxanthine (IBMX) administration was observed with single-cell resolution (Extended Data Fig. 3a–d). Moreover, the cAMPinG1-NE response to local puff application of dopamine in the striatum and bath application of norepinephrine in the cortex was also observed, which was significantly larger than that of G-Flamp1 ($P = 1.2 \times 10^{-3}$ and $4.6 \times 10^{-6}$, unpaired two-tailed $t$-test; Extended Data Fig. 3e–l). We did not observe any abnormalities in the excitability and synaptic transmission of neurons expressing cAMPinG1-NE (Extended Data Fig. 3m–o).

### In vivo imaging of cAMP dynamics with subcellular resolution

We introduced cAMP sensors into pyramidal neurons in layer 2/3 (L2/3) of the mouse primary visual cortex (V1) by in utero electroporation (Fig. 2a). We applied an aversive airpuff stimulus in the awake condition to induce an elevation of norepinephrine and cAMP in the cortex[5]. Somatic cAMPinG1 imaging visualized transient cAMP signals induced by a 20-s airpuff with single-cell resolution (Fig. 2b–e). We also confirmed that cAMPinG1mut did not respond to the airpuff stimulus. (Fig. 2d,e). In addition, the $\Delta F/F$ of cAMPinG1 was significantly larger than that of G-Flamp1 ($P = 9.2 \times 10^{-6}$ between cAMPinG1 and G-Flamp1, unpaired two-tailed $t$-test; Fig. 2d,e). The half-decay time of the cAMP transients was around 20 s, in contrast to the Ca$^{2+}$ transients on the order of hundreds of milliseconds (Fig. 2f).

Then, we imaged cAMPinG1-NE in dendritic spines and shafts in vivo under lightly anesthetized conditions (Fig. 2g). We observed sensory-evoked cAMP transients in both dendritic spines and shafts (Fig. 2h,i). Thus, cAMPinG1 allows in vivo cAMP imaging, which requires advanced sensitivity of the indicators such as spine imaging.

---

**Fig. 1 | Sensor design and in vitro characterization of cAMPinG1. a**, Top, primary structure of cAMPinG1. Bottom, tertiary structures of cAMP-binding PKA-R1α (PDB 1RGS) with cAMP and cpGFP (PDB 3WLD, calmodulin and RS20 domains are hidden) are shown with the linkers between the two domains (dotted lines). **b**, In vitro screening results of 251 variants with resultant cAMPinG1. **c**, Excitation and emission spectra of cAMPinG1 in cAMP-free and cAMP-saturated states. **d**, Primary structures of cAMPinG1mut, cAMPinG1-NE and cAMPinG1-ST. **e**, $\Delta F/F$ of cAMP sensors for cAMP change from 0 to 300 μM in HEK cell lysate at 490 nm (left) or 450 nm (right) of excitation. $n = 4$ wells in each sensor. Tukey's post hoc test following one-way ANOVA. gCarvi versus cAMPinG1 at 490 nm: $P = 1.5 \times 10^{-14}$; G-Flamp1 versus cAMPinG1 at 490 nm: $P = 2.4 \times 10^{-8}$; cAMPinG1 versus cAMPinG1mut at 490 nm: $P = 1.9 \times 10^{-15}$; gCarvi versus G-Flamp1 at 450 nm: $P = 1.7 \times 10^{-9}$; gCarvi versus cAMPinG1 at 450 nm: $P = 3.7 \times 10^{-6}$; G-Flamp1 versus cAMPinG1 at 450 nm: $P = 1.9 \times 10^{-7}$. **f**, cAMP

titration curves. Response of Flamindo2 is inversed ($-\Delta F/F$). The $x$ axis is logarithmic. $n = 4$ wells in each sensor. **g**, $K_d$ values. The $y$ axis is logarithmic. $n = 4$ wells in each sensor. Tukey's post hoc test following one-way ANOVA. gCarvi versus cAMPinG1: $P = 1.5 \times 10^{-14}$; G-Flamp1 versus cAMPinG1: $P = 3.1 \times 10^{-10}$. **h**, Schematic of the imaging settings. 488-nm and 405-nm excitation lights were used in turns for ratiometric imaging in HEK293T cells. **i**, Representative images of HEK293T cells expressing cAMPinG1 excited by 488 nm (left) and 405 nm (right) before (top) and after (bottom) 50 μM forskolin (FSK) application. Scale bar, 10 μm. **j**, $\Delta F/F$ of cAMPinG1 (left) and the inactive mutant cAMPinG1mut (right) in response to 50 μM forskolin application. The ratio (488 ex/405 ex) was calculated with fluorescence excited by 488 nm and 405 nm. $n = 196$ (cAMPinG1), $n = 164$ (cAMPinG1mut) cells. All shaded areas and error bars denote the s.e.m.

## Engineering improved red Ca²⁺ indicator RCaMP3

$Ca^{2+}$ signaling is one of the upstream modulators of cAMP signaling through $Ca^{2+}$-dependent ACs. To facilitate the combinational use of the green cAMP indicator in vivo, we developed a red $Ca^{2+}$ indicator. A series of red $Ca^{2+}$ indicators has been developed based on R-GECO1 (refs. [21],[24]–[28]; Extended Data Fig. 4a). We introduced the following mutations into jRGECO1a; p.Met128Arg/p.Ile131Val/p.Val139Leu/p.Phe188Leu/p.Asn231Ser/p.Ser234Thr (in R-GECO1 numbering) from R-GECO1.2 to blue-shift the excitation spectra, and p.Glu217Asp from sRGECO to generate a large dynamic range. In addition, we used

a self-cleaving peptide (F2A) instead of the nuclear export signal of jRGECO1a[21]. We termed this hybrid indicator RCaMP3 (Fig. 3a).

In vitro characterization revealed that, while RCaMP3 exhibits similar pH sensitivity and photostability to jRGECO1a and XCaMP-R, it shows a larger dynamic range and more blue-shifted excitation spectum[24],[26] (Fig. 3b and Extended Data Fig. 4b–l). The blue-shifted excitation spectrum of RCaMP3 is well suited for two-photon imaging because the excitation peaks of jRGECO1a and XCaMP-R[24],[26] are longer than 1,040 nm, the fixed wavelength of the commonly available dual-wavelength lasers. Indeed, RCaMP3 showed higher fluorescence

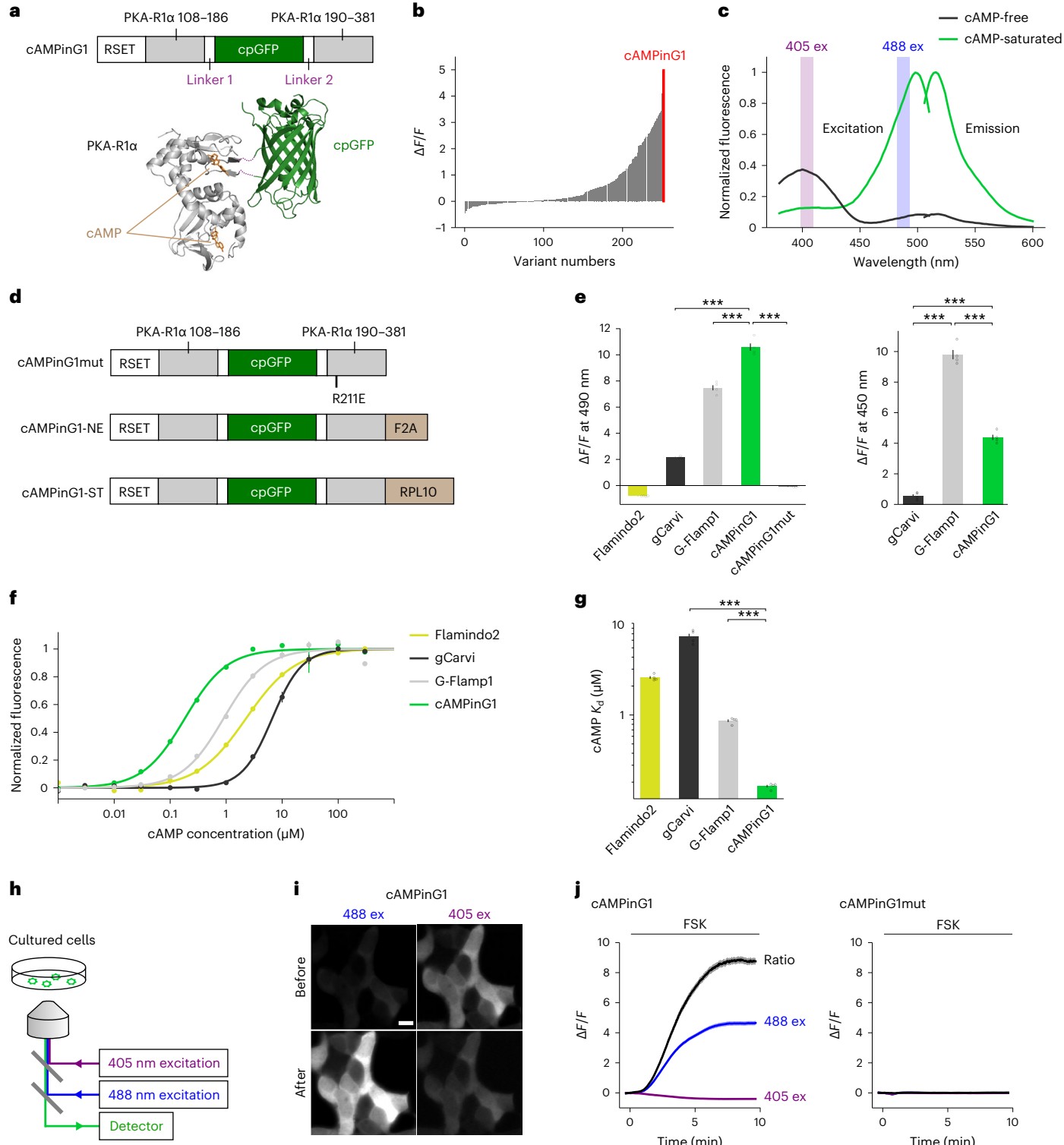

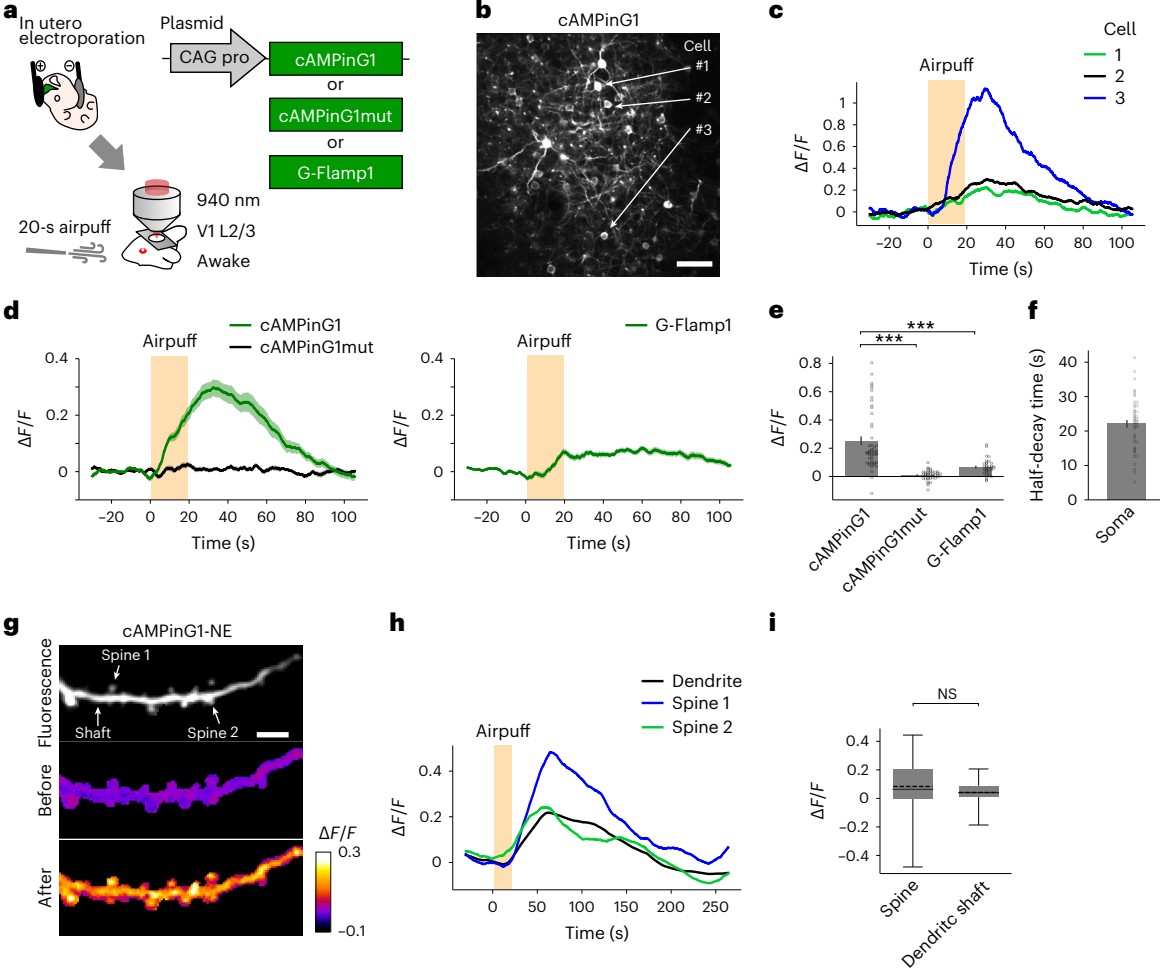

**Fig. 2 | In vivo two-photon cAMPinG1 imaging in somata and dendritic spines. a**, Schematic of the experimental procedure of cAMPinG1 somatic imaging. cAMPinG1 or cAMPinG1mut was delivered to neurons in L2/3 of the mouse V1 by in utero electroporation. **b**, A representative in vivo two-photon fluorescence image of cAMPinG1. Scale bar, 50 μm. **c**, Single-trial cAMP traces of 3 representative cells. The orange box indicates the timing of the stimulus. **d**, Averaged traces of somatic signals of cAMPinG1 (left), cAMPinG1mut (left) and G-Flamp1 (right) in response to airpuff stimulation. $n$ = 47 neurons in 4 mice (cAMPinG1), $n$ = 39 neurons in 4 mice (cAMPinG1mut), $n$ = 32 neurons in 3 mice (G-Flamp1). **e**, Averaged $\Delta F/F$ of cAMPinG1 and cAMPinG1mut in response to airpuff stimulation. $n$ = 47 neurons in 4 mice (cAMPinG1), $n$ = 39 neurons in 4 mice (cAMPinG1mut), $n$ = 32 neurons in 3 mice (G-Flamp1). Unpaired two-tailed $t$-test.

cAMPinG1 versus cAMPinG1mut: $P$ = 6.9 × 10$^{-10}$, cAMPinG1 versus G-Flamp1: $P$ = 9.2 × 10$^{-6}$. **f**, Half-decay time of somatic cAMP transients in response to airpuff. $n$ = 47 neurons in 4 mice. **g**, A representative image of cAMPinG1 imaging in spines and their shaft. cAMPinG1 fluorescence, $\Delta F/F$ before and $\Delta F/F$ after airpuff. Scale bar, 5 μm. **h**, Representative traces of a dendritic shaft and two spines. The orange square indicates the timing of the stimulus. **i**, Averaged $\Delta F/F$ of cAMPinG1 in dendritic shafts and spines. $n$ = 56 spines, $n$ = 11 shafts in 4 mice. Boxes indicate the 25th and 75th percentiles, solid lines indicate the median, dashed lines indicate the mean, and whiskers indicate the total range of data. Unpaired two-tailed $t$-test. $P$ = 4.3 × 10$^{-1}$. All shaded areas and error bars denote the s.e.m. NS, not significant.

than jRGECO1a and XCaMP-R in the Ca$^{2+}$-saturated state excited at 1,040 nm (Fig. 3c).

In acute brain slices, RCaMP3 had larger $\Delta F/F$ to single action potentials than jRGECO1a with comparable rise and decay kinetics (Fig. 3d–h). In addition, we performed loose-seal cell-attached electrical recording and two-photon Ca$^{2+}$ imaging simultaneously in vivo (Fig. 3i). The $\Delta F/F$ of RCaMP3 correlated with the number of action potentials and was consistently higher than that of jRGECO1a across all spike numbers, despite their comparable kinetics. (Fig. 3k–n). We note that the aggregation of RCaMP3 was comparable with that of jRGECO1a (Supplementary Fig. 1).

To demonstrate the sensitivity of RCaMP3, we used fast-scanning high optical invariant two-photon microscopy (FASHIO-2PM)[29]. We performed RCaMP3 imaging in a large field of view (3.0 × 3.0 mm$^2$) including the primary somatosensory cortices (Fig. 3o and Supplementary Video 1). Somatic Ca$^{2+}$ transients of several thousands of

L5 neurons were simultaneously monitored with single-cell resolution (Fig. 3p). In addition, somatic Ca$^{2+}$ transients of L2/3 neurons and dendritic Ca$^{2+}$ transients of L5 neurons could be monitored when imaged in L2/3 (Extended Data Fig. 5).

**Dual-color Ca$^{2+}$ and cAMP imaging during forced running**

To visualize Ca$^{2+}$ and cAMP dynamics with single-cell resolution in vivo, we coexpressed RCaMP3 and cAMPinG1-ST in L2/3 neurons of the V1 by adeno-associated virus (AAV) injection (Fig. 4a). Using a piezo objective scanner, we performed multiple z-plane imaging of head-fixed awake mice (Fig. 4b). cAMPinG1-ST and RCaMP3 were excited with 940 nm and 1,040 nm, respectively. cAMPinG1-ST fluorescence was localized to somata due to the soma-targeting signal RPL10 (ref. 22) (Supplementary Fig. 2). The fluorescence intensity of cAMPinG1 was considerably higher than that of G-Flamp1 when coexpressing Ca$^{2+}$ and cAMP sensors (Supplementary Fig. 3). Here, we simultaneously detected Ca$^{2+}$ and

cAMP signals of more than 400 neurons in L2/3 of a mouse (Fig. 4c,d and Supplementary Video 2). During the imaging, we used a forced running task to increase norepinephrine, cAMP and PKA activities in the cortex[7,8,30]. The majority of neurons showed cAMP increase during running (Fig. 4d,e). The fluorescence changes in cAMPinG1-ST resulted from cAMP binding to the sensor and were significantly larger than those of G-Flamp1 ($P = 4.6 \times 10^{-2}$ between cAMPinG1-ST and G-Flamp1, unpaired two-tailed $t$-test; Extended Data Fig. 6a–d). In addition, this global cAMP increase was less cell specific and dependent on norepinephrine[8,31] and β-adrenoceptors (Extended Data Fig. 6e–g). Furthermore, consistent with a previous report[32], around 25% of cells showed calcium transients during running, as detected by RCaMP3 (Fig. 4d,e). These motion-related cells had larger cAMP transients than the other non-motion-related cells (Fig. 4f–h). This additional cAMP elevation with $Ca^{2+}$ responses may be attributed to $Ca^{2+}$-dependent ACs.

To demonstrate the application of cAMPinG1 and RCaMP3 for cell types other than neurons, we expressed RCaMP3 and cAMPinG1-NE in astrocytes in L2/3 of the V1 (ref. 33). Forced running induced $Ca^{2+}$ increase followed by cAMP increase in astrocytes (Extended Data Fig. 6h–j).

### Dual-color $Ca^{2+}$ and cAMP imaging during visual stimulation

To further investigate the relation between $Ca^{2+}$ and cAMP in vivo, a drifting grating stimulus of eight directions was applied to induce cell-specific $Ca^{2+}$ transients in L2/3 neurons of the V1 (Fig. 5a). RCaMP3 showed direction-selective $Ca^{2+}$ transients in response to 4 s of drifting gratings, consistent with previous studies[26,34] (Fig. 5b–d and Extended Data Fig. 7a,b). Interestingly, cAMPinG1-ST also showed direction-selective cAMP transients preceded by $Ca^{2+}$ transients, which were selective to the same direction and positively correlated with $Ca^{2+}$ (Fig. 5b–d and Extended Data Fig. 7a,b).

Then, we conducted an experiment with repetitive moving gratings of the same direction (8 s, three times). Neurons responding to the direction in $Ca^{2+}$ levels exhibited a cAMP increase, as observed above (Fig. 5e–g). We observed that the cAMPinG1-ST fluorescence began to decrease in the majority of $Ca^{2+}$-responsive and $Ca^{2+}$-unresponsive cells in the middle of visual stimuli. This global cAMP decrease lasted beyond the end of the stimulus and cAMPinG1 fluorescence was lower than the baseline observed before the stimulation (Fig. 5e–g). These fluorescence changes in cAMPinG1-ST resulted from cAMP binding to the sensor and were significantly higher than those of G-Flamp1 ($P = 7.2 \times 10^{-4}$ between cAMPinG1-ST and G-Flamp1, unpaired two-tailed $t$-test after removing outliers with Smirnov–Grubbs' test; Extended Data Fig. 7c,d). We did not find any contamination of the fluorescence signal from RCaMP3 in the green channel (Extended Data Fig. 7e). Overall, two-photon imaging of RCaMP3 and cAMPinG1-ST revealed bidirectional cAMP change and strong correlation between $Ca^{2+}$ and cAMP with single-cell resolution in vivo.

### Action potentials induce somatic cAMP elevation

To determine whether the calcium-related cAMP increase detected above was related to action potentials, we applied single-cell optogenetic stimulation and cAMP imaging in vivo. To achieve spatially precise optical stimulation to the soma of a neuron, we sparsely expressed cAMPinG1-NE and soma-targeted ChRmine in L2/3 neurons of the V1 by utilizing the Cre/loxP recombination system[35] (Fig. 5h,i). We observed an increase in cAMP levels in response to two-photon single-cell stimulation at 1,040 nm (Fig. 5j,k), indicating that action potentials are sufficient to induce somatic cAMP elevation.

### Dual-color fiber photometry for $Ca^{2+}$ and cAMP imaging

We next evaluated the use of cAMPinG1 and RCaMP3 in single-photon fiber photometry in deep brain areas. In line with previous studies[7,8,36], we observed cAMP elevation during the running period following the increase in $Ca^{2+}$ levels in the dorsal striatum (Extended Data Fig. 8). These fluorescence signals arose from cAMP binding and were more intense than those of G-Flamp1.

### cAMPinG1 imaging for GPCR biology and drug screening

To demonstrate the utility of cAMPinG1 in cultured cells, we performed ratiometric imaging by measuring the 488 nm excitation and 405 nm excitation ratio (488 ex/405 ex) at a single timepoint after forskolin or mock application. The 488 ex/405 ex ratio in each cell was higher in forskolin-stimulated cells than in non-stimulated cells (Extended Data Fig. 9a). Furthermore, the ratio change of cAMPinG1 was significantly larger than those of G-Flamp1 and cAMPFIRE-L ($P < 1.0 \times 10^{-13}$ between cAMPinG1 and G-Flamp1, $P < 1.0 \times 10^{-13}$ between cAMPinG1 and cAMPFIRE-L, Tukey's post hoc test following one-way analysis of variance (ANOVA); Extended Data Fig. 9a–c).

To apply this single-cell, single-timepoint cAMPinG1 imaging approach for GPCR biology and pharmacology, we expressed a dopamine receptor D1 (DRD1), a Gs-coupled GPCR with constitutive activity in the cAMP pathway, in the cAMPinG1 stable cell line (Fig. 6a and Extended Data Fig. 9c,d). We fused a self-cleaving 2A peptide (P2A) and RFP to DRD1 to monitor the expression level of DRD1 in each cell. cAMPinG1 imaging revealed a positive correlation between DRD1 expression and cAMP level[37], which diminished with dopamine (Fig. 6a,b). Consistent with previous studies reporting high constitutive activity of GPCR GPR52 (refs. 37,38), we also observed higher cAMP levels of GPCR GPR52 than those of Gs-coupled β2-adrenoceptor (ADRB2) and DRD1 (Fig. 6e). These results suggest that the cAMPinG1 imaging can be available to compare the constitutive activity of cAMP-related GPCRs. The cAMPinG1 imaging also allowed quantifying the agonist activity of a Gi-coupled dopamine receptor D2 (DRD2) as well as that of DRD1 (Fig. 6c,d). The cAMPinG1 imaging also detected inverse agonist activity of clozapine for a Gs-coupled serotonin receptor HTR6 as well as the agonist activity of serotonin (Fig. 6f). Furthermore, the

---

**Fig. 3 | Engineering and characterization of RCaMP3. a**, Top, primary structure of RCaMP3 with amino acids mutated relative to R-GECO1 (in R-GECO1 numbering). Bottom, tertiary structures of R-GECO1 (PDB 4I2Y) with amino acids mutated in RCaMP3 (sphere). **b**, $\Delta F/F$ of red $Ca^{2+}$ sensor in HEK lysate. $n = 4$ wells in each sensor. Tukey's post hoc test following one-way ANOVA. jRGECO1a versus RCaMP3: $P = 5.8 \times 10^{-11}$; XCaMP-R versus RCaMP3: $P = 4.0 \times 10^{-13}$. **c**, Two-photon (1,040 nm) fluorescence in live HEK cells with ionomycin. $n = 360$ (jRGECO1a), $n = 267$ (XCaMP-R), $n = 376$ (RCaMP3) cells. Tukey's post hoc test following one-way ANOVA. jRGECO1a versus RCaMP3: $P = 6.9 \times 10^{-14}$; XCaMP-R versus RCaMP3: $P = 6.9 \times 10^{-14}$. **d**, $Ca^{2+}$ imaging with whole-cell patch-clamp recording in acute brain slices. **e**, Representative jRGECO1a and RCaMP3 responses to single action potentials (APs, vertical lines). Thin lines denote individual traces (jRGECO1a: 12 trials, RCaMP3: 10 trials), and thick lines denote average response. **f**–**h**, $\Delta F/F$ (**f**), rise time (**g**) and half-decay time (**h**) for single APs. $n = 7$ neurons (jRGECO1a), $n = 6$ neurons (RCaMP3). Unpaired two-tailed $t$-test. $P = 4.7 \times 10^{-2}$ (**f**); $P = 9.7 \times 10^{-1}$ (**g**); $P = 4.6 \times 10^{-1}$ (**h**). **i**, $Ca^{2+}$ imaging under a cell-attached recording in vivo.

**j**, Representative trace of simultaneous measurement of RCaMP3 fluorescence and APs in vivo. The number of APs for each event is indicated below the trace. The image shows a neuron expressing RCaMP3 with the recording pipette. Scale bar, 20 μm. **k**, $\Delta F/F$ of jRGECO1a (152 events from 12 cells) and RCaMP3 (228 events from 15 cells) for single APs. Thin lines represent individual traces, and thick lines represent the average traces. **l**,**m**, Half-rise time (**l**) and half-decay time (**m**) for single APs. $n = 12$ cells (jRGECO1a), $n = 15$ cells (RCaMP3). Unpaired two-tailed $t$-test. $P = 8.8 \times 10^{-1}$ (**l**); $P = 2.1 \times 10^{-1}$ (**m**). **n**, $\Delta F/F$ in response to APs (jRGECO1a, $n = 152, 198, 124$ and 44 events for 1, 2, 3 and 4 APs, respectively; RCaMP3, $n = 228$, 139, 159 and 17 events for 1, 2, 3 and 4 APs, respectively). Unpaired two-tailed $t$-test. $P = 4.5 \times 10^{-58}$ (1AP); $P = 1.8 \times 10^{-93}$ (2AP); $P = 1.2 \times 10^{-60}$ (3AP); $P = 3.8 \times 10^{-17}$ (4AP). **o**, Mesoscale $Ca^{2+}$ imaging using FASHIO-2PM. **p**, Left, a representative full field of view. Scale bar, 500 μm. Right, magnified images and $Ca^{2+}$ traces of 12 representative neurons. Scale bar, 10 μm. Error bars denote the s.e.m. Boxes indicate the 25th and 75th percentiles, solid lines indicate the median, dashed lines indicate the mean, and whiskers indicate the total range of data.

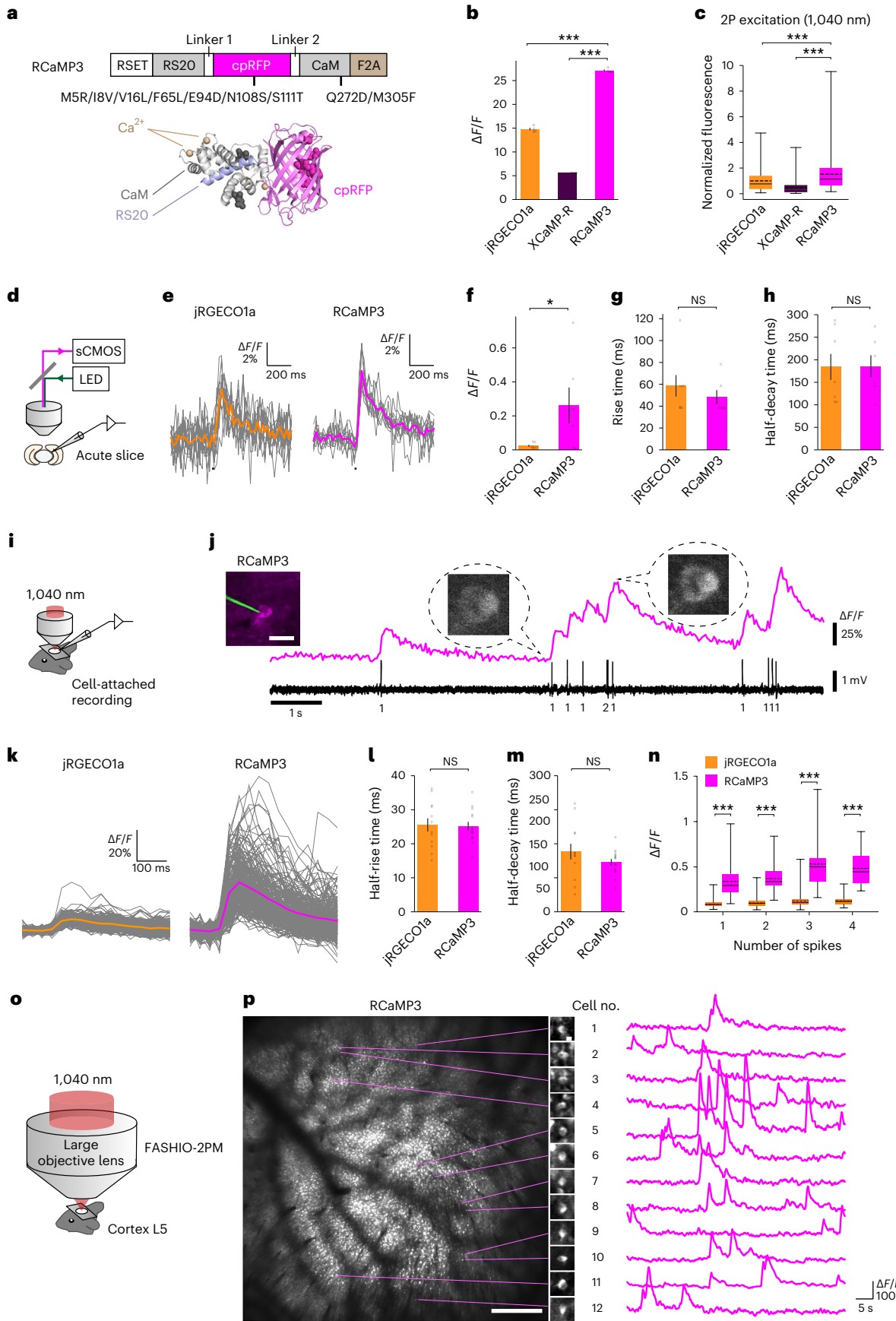

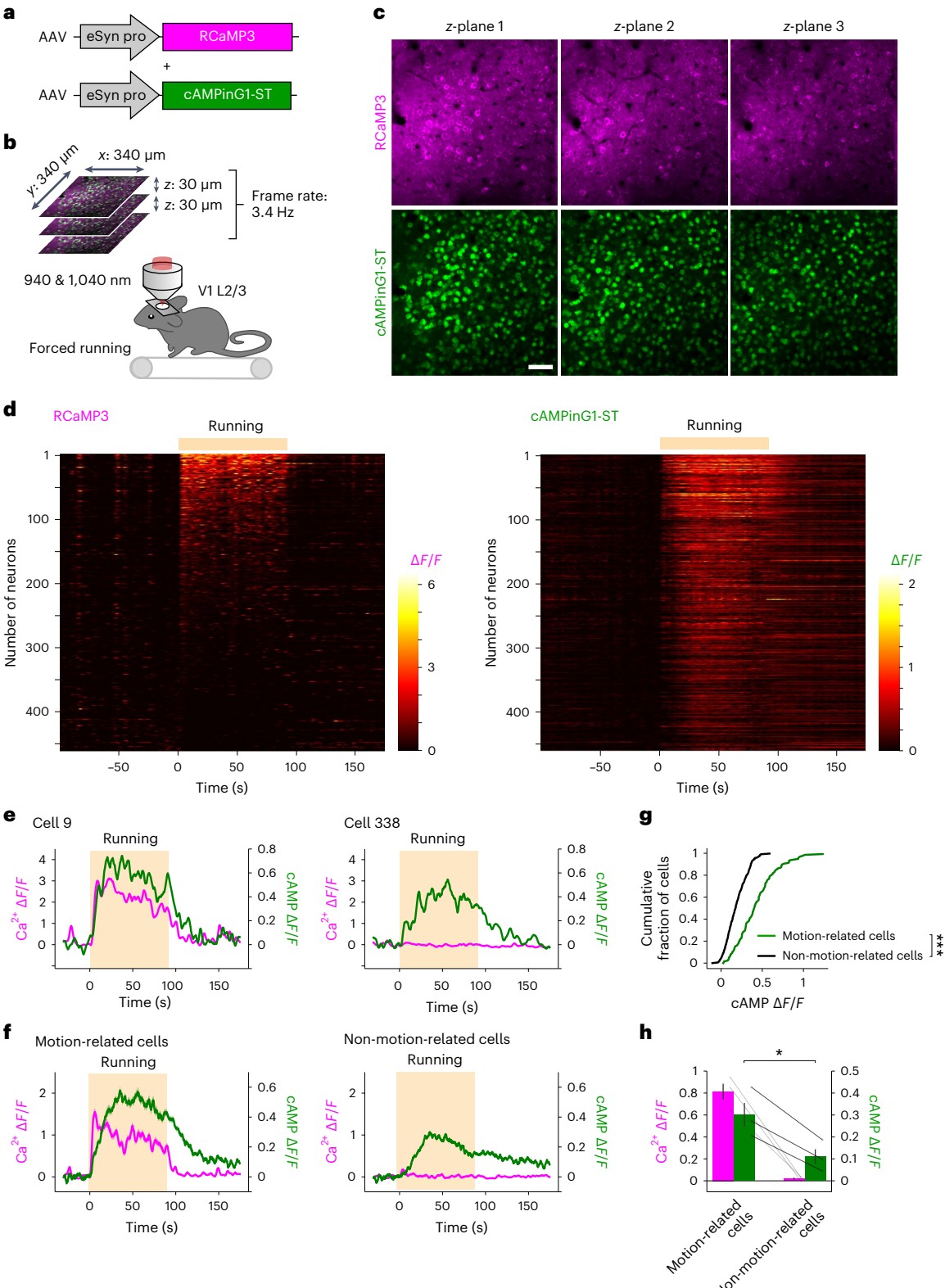

**Fig. 4 | In vivo dual-color imaging for Ca²⁺ and cAMP during forced running.** **a**, Schematic of AAVs. AAVs encoding RCaMP3 and cAMPinG1-ST were co-injected into the L2/3 of the V1. **b**, Schematic of the experimental procedure. Sequential excitation at 940 nm and 1,040 nm was used for dual-color imaging of cAMPinG1-ST and RCaMP3. Three optical planes spaced 30 μm apart were imaged at 3.4 Hz per plane using a piezo objective scanner. **c**, Representative images of RCaMP3 and cAMPinG1-ST. Scale bar, 50 μm. **d**, Single-trial traces of RCaMP3 and cAMPinG1-ST. Cells are sorted according to ΔF/F of RCaMP3 during running. *n* = 461 cells in 1 mouse. **e**, Single-trial traces of RCaMP3 and

cAMPinG1-ST of two representative cells. The box indicates the period of forced running. The cell number on the top corresponds to the number in **d**. **f**, Averaged fluorescence transients of RCaMP3 (magenta) and cAMPinG1-ST (green). *n* = 137 cells in 1 mouse (left), *n* = 324 cells in 1 mouse (right). **g**, Cumulative plot of mean cAMP ΔF/F of motion-related (green) and non-related (black) cells during forced running. *n* = 137 cells in 1 mouse (green), *n* = 324 cells in 1 mouse (black). Kolmogorov–Smirnov test. $P = 1.1 \times 10^{-16}$. **h**, Averaged ΔF/F of RCaMP3 (magenta) and cAMPinG1-ST (green) during forced running. *n* = 3 trials, *n* = 3 mice. Paired two-tailed *t*-test. $P = 1.5 \times 10^{-2}$. All shaded areas and error bars denote the s.e.m.

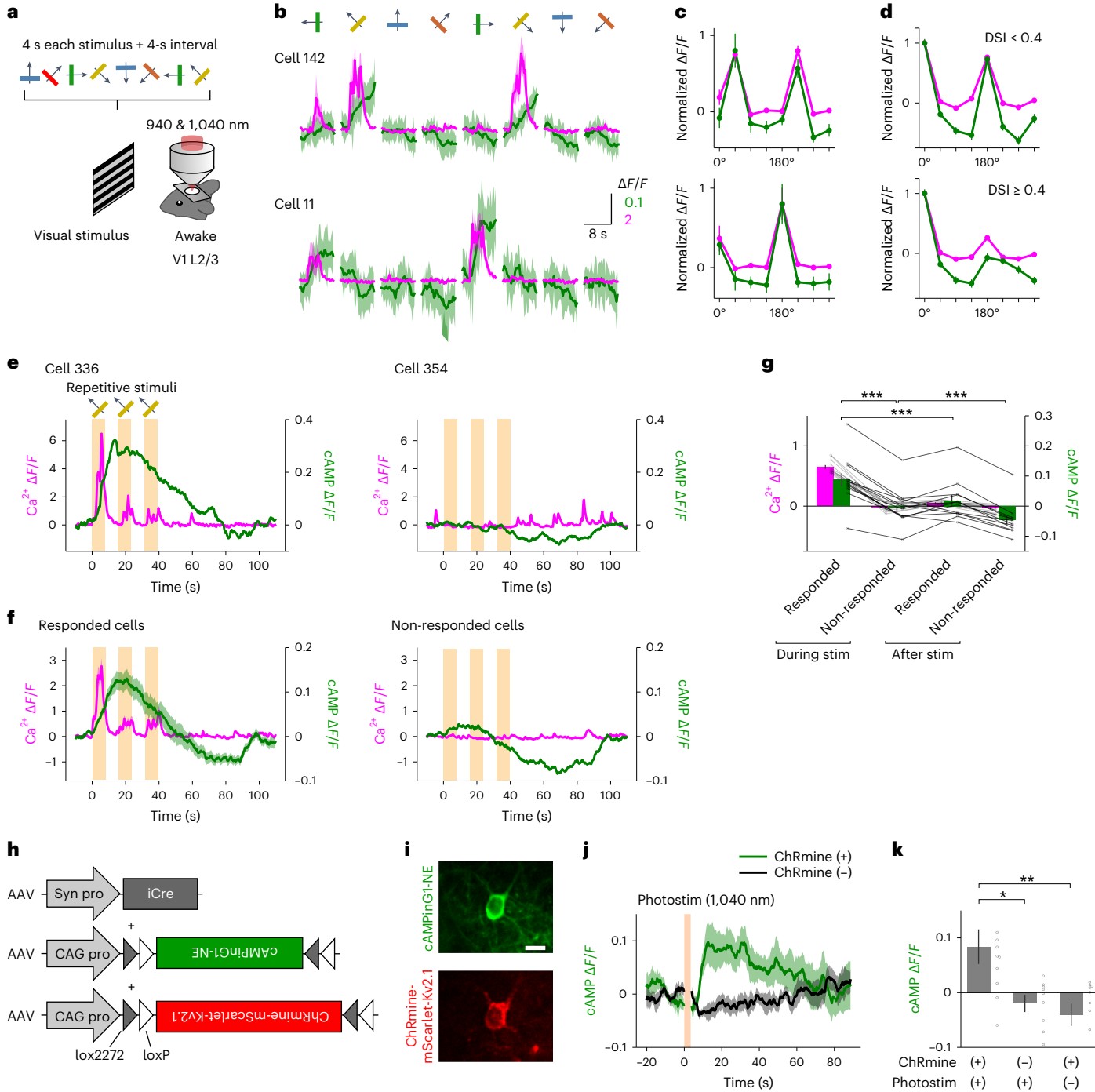

**Fig. 5 | In vivo dual-color imaging for Ca²⁺ and cAMP during visual stimulation. a**, Schematic of the experimental procedure. Moving gratings of 8 directions were used to induce cell-specific Ca²⁺ transients in L2/3 neurons of the V1. **b**, ΔF/F of RCaMP3 and cAMPinG1-ST of 2 representative cells. n = 6 times in each cell. **c**, Direction-selective visual responses of RCaMP3 and cAMPinG1-ST of the two representative cells in **b**. **d**, Direction-selective visual responses of RCaMP3 and cAMPinG1-ST. Top, neurons showing the direction selectivity index (DSI) < 0.4 in Ca²⁺ response. n = 94 cells in 3 mice. Bottom, neurons showing the DSI ≥ 0.4 in Ca²⁺ response. n = 101 cells in 3 mice. **e**, Single-trial traces of RCaMP3 and cAMPinG1-ST of 2 representative cells. The box indicates the period of visual stimuli. **f**, Averaged traces of RCaMP3 (magenta) and cAMPinG1-ST (green). n = 53 cells in 1 mouse (left), n = 408 cells in 1 mouse (right). **g**, ΔF/F of RCaMP3 and cAMPinG1-ST during and after the visual stimuli. n = 12 trials, n = 3 mice. Paired two-tailed t-test. Responded (during stim) versus non-responded

(during stim): $P = 1.5 \times 10^{-7}$; responded (during stim) versus responded (after stim): $P = 2.1 \times 10^{-4}$; non-responded (during stim) versus non-responded (after stim): $P = 5.1 \times 10^{-4}$. **h**, Schematic of AAVs for sparse expression of cAMPinG1-NE and soma-targeted ChRmine. **i**, Representative fluorescence images of cAMPinG1-NE and soma-targeted ChRmine. Scale bar, 10 μm. **j**, ΔF/F of cAMPinG1-NE in response to 1,040 nm of photostimulation. n = 11 neurons in 3 mice (ChRmine (+), green), n = 11 neurons in 3 mice (ChRmine (−), black). **k**, ΔF/F of cAMPinG1-NE in response to 1,040 nm of photostimulation. n = 11 neurons in 3 mice (ChRmine (+), photostim (+)), n = 11 neurons in 3 mice (ChRmine (−), photostim (+)), n = 9 neurons in 3 mice (ChRmine (+), photostim (−)). Tukey's post hoc test following one-way ANOVA. ChRmine (+), photostim (+) versus ChRmine (−), photostim (+): $P = 1.4 \times 10^{-2}$; ChRmine (+), photostim (+) versus ChRmine (+), photostim (−): $P = 2.0 \times 10^{-3}$. All shaded areas and error bars denote the s.e.m.

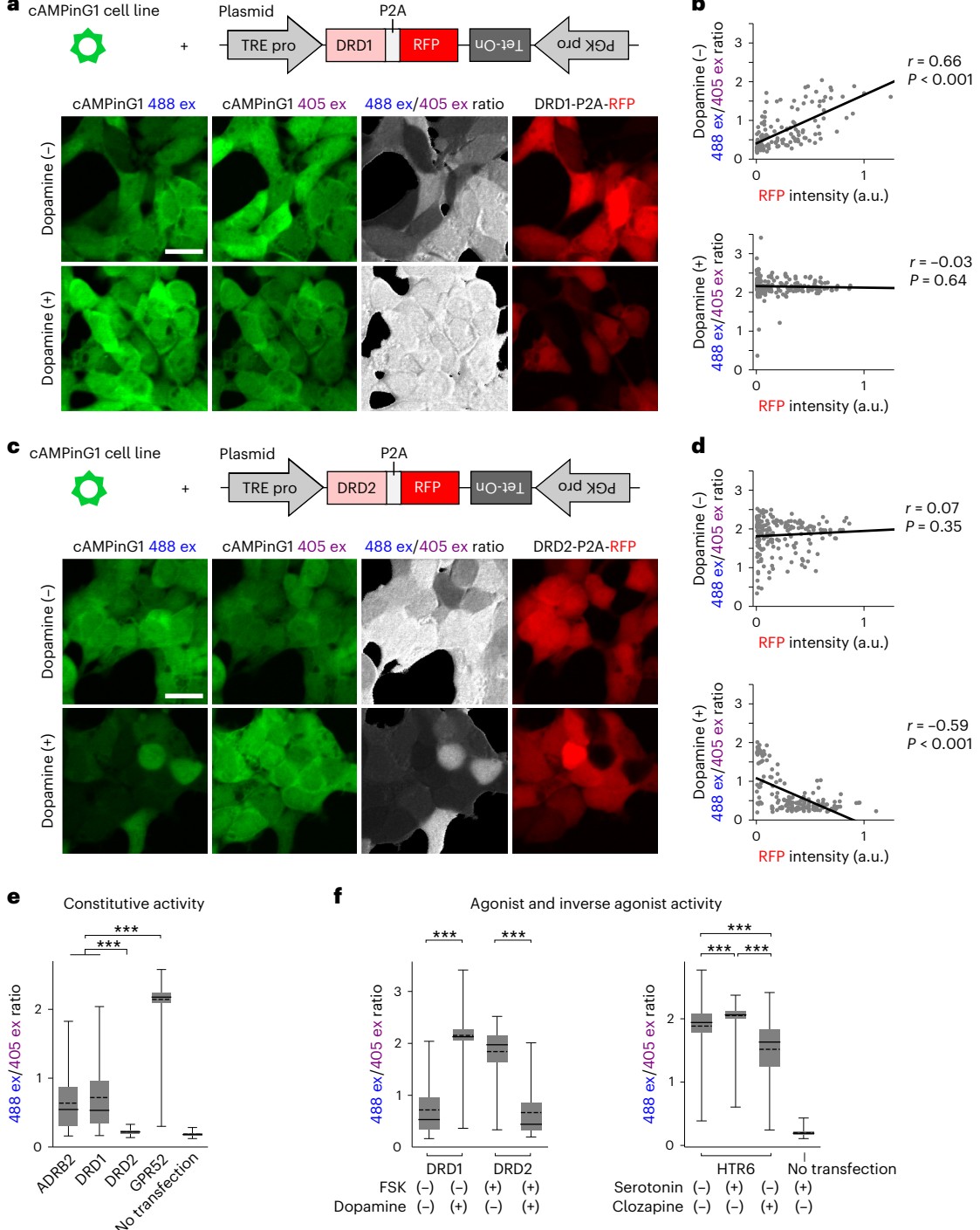

**Fig. 6 | Single-cell, single-timepoint cAMPinG1 imaging for GPCR biology.**
**a**, Schematic of expression system (top). Representative images of the cAMPinG1 stable cell line transiently expressing DRD1–P2A–RFP in the absence (middle) or presence (bottom) of 1,000 nM dopamine were taken by alternating 405/488/561-nm lasers. Scale bar, 20 μm. **b**, Correlation between 488 ex/405 ex ratio of cAMPinG1 and DRD1–P2A–RFP expression level. Individual dots indicate single cells. $n = 141$ (dopamine (−), top) and 190 (dopamine (+), bottom) cells. Pearson correlation coefficient in linear regression, $r = 0.66$; $P = 2.7 \times 10^{-19}$ (top) and $r = −0.03$; $P = 0.64$ (bottom). **c**, Schematic of expression system (top). Representative images of cAMPinG1 cells expressing DRD2–P2A–RFP in the absence (middle) or presence (bottom) of 1,000 nM dopamine. 0.5 μM forskolin in both conditions. Scale bar, 20 μm. **d**, Correlation between 488 ex/405 ex ratio and DRD2–P2A–RFP expression level. $n = 180$ (dopamine (−), top) and 154 (dopamine (+), bottom) cells. Pearson correlation coefficient in linear regression, $r = 0.07$; $P = 0.35$ (top) and $r = −0.59$; $P = 4.9 \times 10^{-16}$ (bottom). **e**, 488 ex/405 ex ratio

of cAMPinG1 cells expressing GPCRs–P2A–RFP without ligands. $n = 158$ (ARDB2), 141 (DRD1), 151 (DRD2), 178 (GPR52) and 126 (no transfection) cells. Tukey's post hoc test following one-way ANOVA. ARDB2 versus DRD2: $P = 9.7 \times 10^{-14}$; DRD1 versus DRD2: $P = 9.7 \times 10^{-14}$; ARDB2 versus GPR52: $P = 9.7 \times 10^{-14}$; DRD1 versus GPR52: $P = 9.7 \times 10^{-14}$. **f**, 488 ex/405 ex ratio of cAMPinG1 cells expressing GPCRs–P2A–RFP with or without agonists. $n = 141$ (DRD1), 190 (dopamine + DRD1), 180 (FSK + DRD2) and 154 (dopamine + FSK + DRD2) cells (left). $n = 159$ (HTR6), 151 (5HT + HTR6), 135 (clozapine + HTR6), 204 (5HT + no transfection) cells (right). Unpaired two-tailed $t$-test (left). DRD1: $P = 6.3 \times 10^{-113}$; DRD2: $P = 2.7 \times 10^{-69}$. Tukey's post hoc test following one-way ANOVA (right). No drug versus serotonin: $P = 5.4 \times 10^{-5}$; no drug versus clozapine: $P = 1.5 \times 10^{-14}$; serotonin versus clozapine: $P = 1.5 \times 10^{-14}$. Boxes indicate the 25th and 75th percentiles, solid lines indicate the median, dashed lines indicate the mean, and whiskers indicate the total range of data. a.u., arbitrary units.

cAMPinG1 imaging visualized that multiple GPCRs in the same field of view specifically responded to their respective agonists (Extended Data Fig. 10a–d). These results suggest that single-cell, single-timepoint cAMPinG1 imaging has the potential for multiplex high-throughput drug screenings.

## Discussion

In this study, we reported cAMPinG1 and RCaMP3, a multicolor suite for cAMP and $Ca^{2+}$ imaging, and showed applications that required advanced sensitivity of these indicators. Furthermore, we addressed an important biological question: the relation between $Ca^{2+}$ and cAMP signaling in vivo.

Each existing cAMP sensor has some drawbacks for in vivo imaging[7,8]. While the basal cAMP concentration varies depending on cell types and experimental conditions, it is estimated to be at ~100 nM in neurons in vitro[2,39]. Therefore, the high affinity of cAMPinG1 ($K_d$ = 181 nM) is well suited to capture these changes, although potential cAMP buffering effects should be considered. G-Flamp1 limited the number of cells that could be imaged and quantified cAMP dynamics in the mouse cortex[7], possibly due to low cAMP affinity ($K_d$ = ~1 μM) as well as low expression efficacy. Another cAMP sensor, cAMPFIRE[8], exceeds the packaging capacity of the AAV vector due to the size of cDNA (approximately 4.5 kb), limiting its in vivo delivery methods. cAMPinG1 (about 1.7 kb) with high cAMP affinity and a large dynamic range overcome these issues. Further investigation with cAMPinG1 will shed light on in vivo dynamics and actual concentrations of cAMP in various neuronal and glial compartments, which are assumed to be modulated within microdomains[40,41].

The global cAMP increase in the V1 during forced running depended on Gs-coupled β-adrenoceptors[5,7,8,31]. On the contrary, the mechanisms of the global cAMP decrease observed during and after stimulations in the cortex and dorsal striatum remain unknown. Some Gi-coupled GPCRs and phosphodiesterases activated by $Ca^{2+}$ or kinases[42] can be involved. The detection of these bidirectional cAMP changes indicates that the cAMP affinity of cAMPinG1 is suitable for in vivo cAMP imaging.

Simultaneous RCaMP3 and cAMPinG1 imaging visualized population dynamics of $Ca^{2+}$ and cAMP in hundreds of cells in the mouse cortex, demonstrating a robust positive correlation between $Ca^{2+}$ and cAMP. In vivo optogenetic experiments demonstrated that action potentials are sufficient to induce somatic cAMP transients. This suggests that the spike-induced cAMP increase can be mediated by voltage-dependent $Ca^{2+}$ channels and $Ca^{2+}$-dependent ACs[43], while the other non-cell-autonomous mechanisms cannot be ruled out. Our results suggest the possibility that cAMP can encode specific information, such as direction selectivity in vision or locomotion encoded in action potentials and $Ca^{2+}$ signaling. This cell-specific cAMP elevation cooperated or competed with global cAMP increase or decrease, respectively, leading to the formation of population patterns of cAMP. Notably, cAMP transients last for tens of seconds, much longer than the hundreds of milliseconds of $Ca^{2+}$ transients. Therefore, information encoded in $Ca^{2+}$ and GPCR signaling can be integrated and stored for a longer period through cAMP transients.

Single-cell, single-timepoint ratiometric cAMP imaging in cultured cells depended on the spectral property of cAMPinG1. This brief and robust technique will contribute to GPCR analysis, such as the systematic determination of cAMP-mediated downstream pathways of GPCRs to address some discrepancies between the results of receptor–G-protein 'couplome' obtained from different assays[44], as well as high-throughput drug screening.

## Online content

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

## Methods

### Animals

All experimental procedures were performed using protocols that were approved by the Institutional Animal Care and Use Committee at Kyoto University (Lif-K23008), University of Yamanashi (A1-20), RIKEN (W2022-2-012) and The University of Tokyo (P18-118). Wild-type mice (C57BL/6N, ICR) and A2A-Cre ((B6.FVB (Cg)-Tg (Adora2a-cre) KG139Gsat/Mmucd)) mice were group-housed and kept on a 12-h light–dark cycle with ad libitum food and water. The housing conditions were controlled at room temperature of approximately 22–24 °C and a relative humidity of 40–60%. Wild-type mice used in this study were purchased from Japan SLC. Experiments were performed using both male and female sex between 0 and 20 weeks of age.

### Cell lines

HEK293T cells were obtained from the American Type Culture Collection (CRL-11268). Cells were cultured in DMEM (08458-16, Nacalai) supplemented with 10% fetal bovine serum (FBS) (Sigma-Aldrich), 50 units per ml penicillin and 50 µg ml$^{-1}$ streptomycin (26252-94, Nacalai) at 37 °C and 5% $CO_2$ in a humidified atmosphere. *Escherichia coli* (*E. Coli*) DH5α and DH10B, and Stbl3 cells were obtained from Toyobo (DNA-9303), Invitrogen (18297010) and Invitrogen (C737303), respectively. Bacteria were incubated in Lysogeny Broth (LB) medium supplemented with antibiotics at 37 °C.

### Plasmids

To develop cAMP sensors, PKA-R1α (amino acids 108–186 and 190–381) was obtained from a mouse cDNA library, and cpGFP and RSET domain were subcloned from GCaMP6f (Addgene plasmid, 52924). The F2A sequence was from XCaMP-R[24].The RPL10 domain was from GCaMP6m-RPL10a[22]. G-Flamp1 was synthesized (GENEWIZ). To develop red $Ca^{2+}$ sensors, the cpRFP domain was synthesized (GENEWIZ), and RSET, RS20 and CaM domains were obtained from jRGECO1a[26]. For site-directed mutagenesis, plasmid libraries were made using the inverse PCR method with PrimeSTAR Max DNA polymerase (R045A, Clontech), In-Fusion HD Cloning Kit (639650, Clontech), and primers which included NNK codons, where K = G or T. For expression in *E. coli*, cAMP sensors were subcloned into a pBAD vector[45]. For optogenetic stimulation, ChRmine-mScarlet-Kv2.1 was synthesized (GENEWIZ). For expression in HEK293T cells, $Ca^{2+}$ or cAMP sensors were subcloned into a plasmid encoding CAG promoter and woodchuck hepatitis virus posttranscriptional regulatory element (WPRE). For expression of GPCRs–P2A–mCherry in HEK293T cells, pTRE3G-HA signal-Flag–GPCRs–P2A–mCherry-reverse-PGK–TetOn3G was made of Tet-ON 3G inducible expression system (Clontech) and PRESTO-Tango GPCR Kit[46].

### In vitro fluorometry for cAMP sensor screening

The plasmids for bacterial expression of cAMP sensors were transformed into *E. coli* strain DH10B. *E. coli* cells were plated and cultured at 37 °C on an LB agar plate with ampicillin and 0.0004% arabinose. Each colony was used to inoculate 1.5 ml of LB liquid medium with ampicillin and 0.2% arabinose and grown at 37 °C overnight. After centrifugation, cells were resuspended in 150 µl suspension buffer (20 mM MOPS (pH 7.2), 100 mM KCl, 1 mM dithiothreitol (DTT), cOmplete EDTA-free protease inhibitor cocktail; 11836170001, Roche), sonicated at 4 °C, and centrifuged. The supernatant was collected. For fluorometry, the supernatant was diluted 20-fold with the suspension buffer. The diluted supernatant was applied to 96-well plates. cAMP (A2381, Tokyo Chemical Industry) was added to a final concentration of 300 µM for cAMP-saturated conditions. Fluorometric measurements were performed on a microplate reader (Spark, TECAN) at room temperature at 485 nm of excitation (bandwidth of 20 nm) and 535 nm of emission (bandwidth 20 nm).

### In vitro cAMP fluorometry for HEK cell lysate

HEK293T cells were incubated on six-well plates. Next, 1 µg DNA was transfected using X-tremeGENE HP DNA Transfection Reagent (6366244001, Roche). Two days after the transfection, the cells were harvested into the 150 µl suspension buffer described above, sonicated at 4 °C, and centrifuged. The supernatant was collected. For cAMP-saturated conditions, the supernatant was diluted 20-fold with suspension buffer, and cAMP was added to a final concentration of 300 µM. Fluorometric measurements were performed at 485 nm or 450 nm of excitation (bandwidth 20 nm) and 545 nm of emission (bandwidth 20 nm). Excitation and emission spectra were taken at 555 nm and 460 nm, respectively. For measurement of cAMP affinity, suspension buffer with 0, 3, 10, 30, 100, 300, 1,000, 3,000, 10,000, 30,000, 100,000 and 300,000 nM cAMP was prepared. Then, the supernatant was diluted 40-fold with each suspension buffer. For measurement of cGMP affinity, suspension buffer with 0, 100, 300, 1,000, 3,000, 10,000, 30,000, 100,000, 300,000 and 1,000,000 nM cGMP (ab120805, Abcam) was prepared. Then, the supernatant was diluted 40-fold with each suspension buffer. For measurement of pKa, the supernatant was diluted 40-fold with pH buffer (20 mM citrate (for pH 5–6), 20 mM MOPS (for pH 6.5–10))[25], and cAMP was added to a final concentration of 100 µM. pKa values were determined from the inflection point of a sigmoid fit to fluorescence versus pH.

### In vitro $Ca^{2+}$ fluorometry for HEK cell lysate

Cell incubation, transfection and collection were performed as described above. For $Ca^{2+}$-saturated or $Ca^{2+}$-free conditions, the supernatant was diluted 20-fold with the $Ca^{2+}$-EGTA buffer (30 mM MOPS (pH 7.2), 100 mM KCl, 10 mM EGTA, 10 mM $CaCl_2$ and 1 mM DTT) or EGTA buffer (30 mM MOPS (pH 7.2), 100 mM KCl, 10 mM EGTA and 1 mM DTT), respectively. Fluorometric measurements were performed at 560 nm of excitation (bandwidth of 20 nm) and 610 nm of emission (bandwidth of 20 nm). Excitation and emission spectra were taken at the emission of 635 nm and at the excitation of 520 nm, respectively. For measurement of $Ca^{2+}$ affinity, the supernatant was diluted 40-fold with a series of solutions with free $Ca^{2+}$ concentration ranges from 0 nM to 3,900 nM (ref. 27). For measurement of pKa, the supernatant was diluted 40-fold with pH buffer (20 mM citrate (for pH 5–6), 20 mM MOPS (for pH 6.5–10)) containing 2 mM $CaCl_2$ or 2 mM EGTA[25]. pKa values were determined from the inflection point of a sigmoid fit to fluorescence versus pH.

### In vitro kinetics analysis

The plasmids for bacterial expression of cAMPinG1, which included His-tag sequence, were transformed into *E. coli* strain DH10B. *E. coli* cells were plated and cultured at 37 °C on an agar plate with ampicillin. A colony was used to inoculate 200 ml of LB liquid medium with ampicillin and 0.2% arabinose and grown at 18 °C for 44 h. After centrifugation, cells were resuspended in 10 ml suspension buffer (25 mM Tris-HCl (pH 8.0), 300 mM NaCl, 1 mM DTT and cOmplete EDTA free (Sigma-Aldrich)), sonicated at 4 °C and centrifuged. The supernatant was collected.

Coverslips (no. 1, 15 mm diameter, Matsunami) were washed with 1% acetic acid solution for 6 h. Then, they were immersed in silanized solution containing 50% ethanol, 2% mercaptosilane and 1% acetic acid overnight, followed by drying at 143 °C for 1 h. They were immersed sequentially in reducing solution containing 50% ethanol, 2.5 mM ethylenediaminetetraacetic acid, 2 mM DTT and 100 mM phosphate buffer, pH = 7.0 for 1 h, 2.5 mg ml$^{-1}$ maleimido-C3-NTA solution for 2 h, and 50 mM nickel sulfate solution for 1 h (ref. 9). The resultant nickel complex-coated coverslips were incubated in the supernatant including cAMPinG1 protein for 1 h at 4 °C. The cAMPinG1-binding coverslips were placed on the stage of an FVMPE-RS (Olympus) microscope equipped with a water-immersion ×25 objective lens (N.A.: 1.05, Olympus), a femtosecond laser (Insight DS+, Spectra-Physics) and

GaAsP detector (Hamamatsu Photonics) with a 495–540-nm emission filter (Olympus). The laser was tuned at 920 nm. For measurement of binding kinetics (Extended Data Fig. 2g), cAMP was microperfused to the surface of the coverslips from 0 µM to 2 µM with a micropipette and a microinjector (BEX). Images ($288 \times 18$ µm², $128 \times 8$ pixels) were collected at 81.6 Hz. For measurement of dissociation kinetics (Extended Data Fig. 2h), cAMP was microperfused from 0.5 µM to 0 µM. Images ($576 \times 108$ µm², $256 \times 48$ pixels) were collected at 12.7 Hz.

### cAMP imaging in HEK293T cells
For time-lapse imaging (Fig. 1h–j), HEK293T cells were incubated in 35-mm glass-bottom dishes. DNA encoding the sensors (1 µg) was transfected as described above. One day after the transfection, the culture medium was replaced with Tyrode's solution (129 mM NaCl, 5 mM KCl, 30 mM glucose, 25 mM HEPES–NaOH, pH 7.4, 2 mM CaCl₂, 2 mM MgCl₂). Imaging for cAMPinG1 was performed using an LSM 880 confocal microscope (Carl Zeiss) with an air-immersion ×20 objective lens (N.A.: 0.80, Carl Zeiss). Then, 405-nm and 488-nm lasers were used for excitation in turns. Forskolin (16384-84, Nacalai) was added to a final concentration of 50 µM.

For single-timepoint imaging for GPCRs (Fig. 6), HEK293T cells were incubated in 96-well glass-bottom plates. Around 0.1 µg DNA encoding the sensors was transfected as described above. One day after the transfection, the culture medium was replaced with Tyrode's solution. Imaging was performed using an LSM 880 confocal microscope with the ×20 objective lens. For tetracycline-dependent expression of GPCRs–P2A–mCherry, doxycycline was added to a final concentration of 100 ng ml⁻¹ 3 h before the imaging. Twenty minutes before the imaging, the culture medium was replaced with Tyrode's solution with or without forskolin, dopamine (14212-71, Nacalai), clozapine (12059, Cayman), adrenocorticotropic hormone (AP3295, AdooQ,) or serotonin (18961-41, Nacalai). Lasers (405 nm and 488 nm) were used for ratiometric cAMP imaging, and 561-mn and 633-nm lasers were used for the visualization of GPCR-expressing cells.

For single-timepoint imaging for sensor comparison (Extended Data Fig. 9a–c), HEK293T cells were incubated in 96-well glass-bottom plates, transfected and imaged as described above. Imaging was performed to calculate the relative change ($\Delta R/R$) in fluorescence ratio ($R$) in the absence or presence of 10 µM forskolin. For cAMPinG1 and G-Flamp1, $R$ is the ratio of green fluorescence (491–553 nm of emission) with 488 nm of excitation to that with 405 nm of excitation. For cAMPFIRE-L, $R$ is the ratio of cyan fluorescence (464–499 nm of emission) with 458 nm of excitation to yellow fluorescence (526–597 nm of emission) with 514 nm of excitation.

### Binding assay in HEK293T cells
HEK293T cells were incubated in 35-mm glass-bottom dishes. DNA encoding the sensors (1 µg) was transfected as described above. One day after the transfection, the culture medium was replaced with Tyrode's solution as described above. Imaging was performed using an LSM 880 confocal microscope with an oil-immersion ×40 objective lens (N.A.: 1.30, Carl Zeiss). Images ($53.1 \times 53.1$ µm², $512 \times 512$ pixels) were collected using 488-nm and 561-nm lasers.

### Fluorescence lifetime measurement
HEK293T cells in 96-well glass-bottom plates were prepared as described above. Twenty minutes before the imaging, the culture medium was replaced with Tyrode's solution with or without 10 µM forskolin. Imaging was performed using a TCS SP8 FALCON microscope (Leica) at a pulse frequency of 80 MHz with an air-immersion ×20 objective lens (N.A.: 0.75, Leica). Excitation was 488 nm by white-light laser and emission was 500–550 nm. Images ($233 \times 233$ µm², $256 \times 256$ pixels) were collected. The lifetime was analyzed using the LAS X FLIM/FCS software (Leica).

### Two-photon Ca²⁺ imaging in HEK293T cells
HEK293T cells were incubated in 35-mm glass-bottom dishes. A mixture of 0.8 µg DNA encoding the red Ca²⁺ sensors and 0.2 µg DNA of pCMV-mCerulean was transfected as described above. One day after the transfection, the culture medium was replaced with Tyrode's solution as described above. Thirty seconds after bath application of ionomycin (Cayman Chemical, 11932) to a final concentration of 5 µM, two-photon imaging was performed with an FVMPE-RS (Olympus) equipped with a water-immersion ×25 objective lens (N.A.: 1.05, Olympus), a femtosecond laser (Insight DS+, Spectra-Physics) and two GaAsP detectors (Hamamatsu Photonics) with 495–540-nm and 575–645-nm emission filters (Olympus). Images ($339 \times 339$ µm², $1,024 \times 1,024$ pixels, single optical section) were collected. The laser was tuned to 880 nm for mCerulean and 1,040 nm at the front aperture of the objective for the red Ca²⁺ sensors.

### Photostability analysis
HEK293T cells were prepared in 96-well glass-bottom plates as described above. For one-photon bleaching measurements (Extended Data Fig. 4k), imaging was performed using an LSM 880 confocal microscope with a ×20 objective lens immersed with Tyrode's solution. A 561-nm laser was used for excitation. Images ($170 \times 170$ µm², $512 \times 512$ pixels) were collected at 0.6 Hz. For two-photon bleaching measurement (Extended Data Fig. 4l), imaging was performed using a FVMPE-RS microscope with the ×20 objective lens in Tyrode's solution. A 1,040-nm laser was used for excitation. Images ($255 \times 255$ µm², $256 \times 256$ pixels) were collected at 0.82 Hz.

### Stable cell lines generation
Lentiviral particles were produced by transfection of the packaging plasmids with polyethylenimine into HEK293T cells[47]. Lentivirus-infected HEK293T cells were dissociated and isolated into multi-well plates. Single clones with bright fluorescence were picked, grown and stored at −80 °C. To establish a triple stable cell line (Extended Data Fig. 9), cAMPinG1 single stable cell line was infected with lentivirus encoding TRE3G–DRD1–EF1a–TetOn3G–P2A–mCherry–NLSx3 and TRE3G–MC3R–EF1a–TetOn3G–P2A–iRFP670 (ref. 48).

### Cell proliferation assay
The HEK293T cells were seeded on six-well plates with 2 ml DMEM and 10% FBS or 1.5% FBS and incubated at 37 °C. The 1.5% FBS group was a positive control of slow cell proliferation. Twenty hours after the beginning of the culture, cells in half of the wells were harvested, and the number of cells was counted using a counting chamber as a timepoint of zero. Sixty hours after the beginning of the culture, cells in the other half of the wells were harvested and counted as a timepoint of 48 h.

### In utero electroporation
ICR pregnant mice (Japan SLC) were anesthetized with an anesthetic mixture (0.075 mg ml⁻¹ medetomidine hydrochloride, 0.40 mg ml⁻¹ midazolam and 0.50 mg ml⁻¹ butorphanol tartrate) and administered at 100 µl per 10 g of body weight intraperitoneally. Then, 2.0 µl of purified plasmid (1.0 µg µl⁻¹ final concentration in each sensor) was injected into the right lateral ventricle of embryos at embryonic day 15. pCAG-green cAMP sensors-WPRE (cAMPinG1-NE, cAMPinG1mut-NE, G-Flamp1) and pCAG-RCaMP3-WPRE were delivered to induce the expression of cAMP and Ca²⁺ indicators in L2/3 pyramidal neurons in the V1. After soaking the uterine horn with warm saline (37 °C), each embryo's head was carefully held between tweezers with platinum 5-mm disk electrodes (CUY650P5, Nepagene). Subsequently, five electrical pulses (45 V, 50-ms duration at 1 Hz) were delivered by an electroporator (NEPA21, Nepagene)[49,50]. Electroporated mice were used for cAMP and calcium imaging 4–10 weeks after birth.

## AAV production and injection

Recombinant AAVs were produced using HEK293T cells and purified by AVB Sepharose[51]. The final titers were: AAV2/1-CAG-DIO-cAMPinG1-NE ($5.0 \times 10^{13}$ genome copies (GC) per ml), AAV2/1-CAG-DIO-cAMPinG-1mut-NE ($2.0 \times 10^{13}$ GC per ml) and AAVPHP.eB-CaMKII-Cre ($1.0 \times 10^{12}$ GC per ml) for forskolin/IBMX bath application in acute brain slices (Extended Data Fig. 3a–d); AAV2/1-eSyn-cAMPinG-1mut-NE ($1.5 \times 10^{13}$ GC per ml) and AAV2/1-CaMKII (0.3 kb)-loxFAS-H2B-mCherry-loxFAS ($1.0 \times 10^{13}$ GC per ml) for local dopamine application in acute brain slices (Extended Data Fig. 3e–h); AAV2/1-eSyn-cAMPinG1mut-NE ($1.5 \times 10^{13}$ GC per ml) and AAVPHP. eB-hSyn-EGFP ($2.0 \times 10^{13}$ GC per ml) for electrophysiology in acute brain slices (Extended Data Fig. 3m–o); AAV2/1-eSyn-NES-jRGECO1a ($3.0 \times 10^{13}$ GC per ml) and AAV2/1-eSyn-RCaMP3 ($2.0 \times 10^{13}$ GC per ml) for one-photon and two-photon calcium imaging in the barrel cortex (Fig. 3d–n); AAV2/1-eSyn-NES-jRGECO1a ($1.0 \times 10^{13}$ GC per ml) and AAV2/1-eSyn-RCaMP3 ($1.0 \times 10^{13}$ GC per ml) for two-photon mesoscale Ca$^{2+}$ imaging (Fig. 3o,p); AAV2/1-eSyn-NES-jRGECO1a ($1.0 \times 10^{13}$ GC per ml) and AAV2/1-eSyn-RCaMP3 ($1.0 \times 10^{13}$ GC per ml) for inclusion counting (Extended Data Fig. 5a,b); AAV2/1-eSyn-cAMPinG1-ST ($1.0 \times 10^{13}$ GC per ml), AAV2/1-eSyn-cAMPinG1mut-ST ($1.0 \times 10^{13}$ GC per ml) and AAV2/1-eSyn-RCaMP3 ($1.0 \times 10^{13}$ GC per ml) for two-photon imaging in the V1 (Figs. 4 and 5a–g). AAV2/1-eSyn-cAMPinG1-NE ($7.0 \times 10^{12}$ GC per ml) and AAV2/1-eSyn-G-Flamp1 ($7.0 \times 10^{12}$ GC per ml) for confocal imaging in fixed tissues (Extended Data Fig. 5f, g); AAV2/1-hSyn-iCre ($2.0 \times 10^{10}$ GC per ml), AAV2/1-CAG-DIO-cAMPinG1-NE ($3.0 \times 10^{13}$ GC per ml), AAV2/1-CAG-DIO-RCaMP3 ($1.0 \times 10^{13}$ GC per ml, for an infection marker) and AAV2/1-CAG-DIO-ChRmine-mScarlet-Kv2.1 ($1.0 \times 10^{12}$ GC per ml) for cAMP imaging and optogenetic stimulation (Fig. 5h–k); AAV2/1-eSyn-cAMPinG1-NE ($7.0 \times 10^{12}$ GC per ml) and AAV2/1-eSyn-RCaMP3 ($1.0 \times 10^{13}$ GC per ml, for an infection marker) for fiber photometry in the V1 (Extended Data Fig. 6e–g); AAV2/1-gfaABC1D-cAMPinG1-NE ($1.0 \times 10^{13}$ GC per ml) and AAV2/1-gfaABC1D-RCaMP3 ($1.0 \times 10^{13}$ GC per ml) for astrocyte imaging (Extended Data Fig. 6h–j). AAV2/1-eSyn-G-Flamp1 ($3.0 \times 10^{13}$ GC per ml) for two-photon imaging in the V1 (note that the G-Flamp1 fluorescence diminishes when injecting a mixture of G-Flamp1 and RCaMP3 AAVs for coexpression; Extended Data Fig. 7c,d); AAV2/1-eSyn-cAMPinG1-NE ($7.0 \times 10^{12}$ GC per ml), AAV2/1-eSyn-cAMPinG1mut-NE ($7.0 \times 10^{12}$ GC per ml) or AAV2/1-eSyn-G-Flamp1 ($7.0 \times 10^{12}$ GC per ml) and AAV2/1-eSyn-RCaMP3 ($1.0 \times 10^{13}$ GC per ml) for fiber photometry in the dorsal striatum (Extended Data Fig. 8).

Stereotaxic virus injection was performed to C57BL/6N male mice aged 4–6 weeks anesthetized by the anesthetic mixture described above except for two-photon mesoscale imaging. A2A-Cre transgenic mice were used for the slice experiments for local dopamine application (Extended Data Fig. 3e–h). A micropipette was inserted into the right V1 (A/P −3.85 mm, M/L +2.7 mm from bregma, D/V −0.30 mm from the pial surface), the barrel cortex (A/P −1.0 mm, M/L −3.0 mm from bregma, D/V −0.20 mm from the pial surface), the right dorsal striatum (A/P +0.5 mm, M/L +1.8 mm from bregma, D/V −1.5 mm from the pial surface), the nucleus accumbens (A/P +1.3 mm, M/L ±1.25 mm from bregma, D/V −4.5 mm) or the medial prefrontal cortex (A/P +1.8 mm, M/L +0.3 mm from bregma, D/V −2.4 mm). Then, the virus solution (volume: 500–1,000 nl) was injected. Carprofen (5 mg per kg of body weight, Zoetis) was administered intraperitoneally just after the injection experiment. Mice were subjected to imaging after 4–12 weeks of the injection.

## cAMP imaging in acute brain slices

For cAMP imaging with forskolin/IBMX bath application (Extended Data Fig. 3a–d), AAV (AAVPHP.eB-CaMKII-Cre and AAV2/1-CAG-DIO-cAMPinG1, or AAV2/1-CAG-DIO-cAMPinG1mut) was injected into the V1 at a total volume of 500 nl (ref. 52). For cAMP imaging with local dopamine application (Extended Data Fig. 3e–h),

AAV (AAV2/1-eSyn-cAMPinG1mut-NE, AAV-CaMKII (0.3 kb)-loxFAS-H2B-mCherry-loxFAS) was injected into the nucleus accumbens at a total volume of 1,000 nl. After 2 weeks of expression, mice were killed by rapid decapitation after anesthesia with isoflurane. The brains were immediately extracted and immersed in gassed (95% O$_2$/5% CO$_2$) and ice-cold solution containing: 220 mM sucrose, 3 mM KCl, 8 mM MgCl$_2$, 1.25 mM NaH$_2$PO$_4$, 26 mM NaHCO$_3$ and 25 mM glucose. Acute coronal brain slices (280 µm thick) of the visual cortex were cut in gassed ice-cold solution with a vibratome (VT1200, Leica). Brain slices were then transferred to an incubation chamber containing gassed artificial cerebrospinal fluid (ACSF) containing: 125 mM NaCl, 2.5 mM KCl, 1.25 mM NaH$_2$PO$_4$, 26 mM NaHCO$_3$, 1 mM CaCl$_2$, 2 mM MgCl$_2$ and 20 mM glucose at 34 °C for 30 min and subsequently maintained at room temperature before transferring them to the recording chamber and perfused with the ACSF solution described above, except using 2 mM CaCl$_2$ and 1 mM MgCl$_2$ at 30–32 °C. cAMP imaging was performed with an upright microscope (BX61WI, Olympus) equipped with an FV1000 laser-scanning system (FV1000, Olympus) and a water-immersion ×60 objective (N.A.: 1.0, Olympus), a femtosecond laser (MaiTai, Spectra-Physics) and a GaAsP detector (Hamamatsu Photonics) with a 500–550-nm emission filter (Semrock). The laser was tuned at 940 nm (2 mW at the front aperture of the objective). For cAMP imaging with forskolin/IBMX bath application (Extended Data Fig. 3a–d), images (105.6 × 105.6 µm$^2$, 640 × 640 pixels) were taken every 30 s. During the imaging, forskolin (067-02191, Wako) and IBMX (3758, Tocris) were added to a final concentration of 25 µM and 50 µM, respectively. For cAMP imaging with local dopamine application (Extended Data Fig. 3e–h), images were taken at 2 Hz. During the imaging, 1 µM dopamine was microperfused with a micropipette.

For cAMP imaging with norepinephrine bath application (Extended Data Fig. 3i–l), in utero electroporation was performed as described above. Four weeks after birth, mice were killed by rapid decapitation after anesthesia with the anesthetic mixture described above. The brains were immediately extracted and immersed in gassed (95% O$_2$/5% CO$_2$) and ice-cold solution containing: 222 mM sucrose, 3.6 mM KCl, 2 mM MgCl$_2$, 2 mM CaCl$_2$, 1.5 mM NaH$_2$PO$_4$ and 27 mM NaHCO$_3$. Acute coronal brain slices (300 µm thick) of the visual cortex were cut in gassed ice-cold solution with a vibratome (VT1200, Leica). Brain slices were then transferred to an incubation chamber containing gassed ACSF containing: 126 mM NaCl, 3 mM KCl, 1.1 mM NaH$_2$PO$_4$, 26 mM NaHCO$_3$, 2 mM CaCl$_2$, 2 mM MgCl$_2$ and 10 mM glucose at room temperature for 30 min. The brain slices were transferred to the recording chamber and perfused with the ACSF solution described above. cAMP imaging was performed with FVMPE-RS (Olympus) equipped with a water-immersion ×25 objective lens (N.A.: 1.05, Olympus), a femtosecond laser (Insight DS+, Spectra-Physics) and two GaAsP detectors (Hamamatsu Photonics) with 495–540-nm emission filters (Olympus). The laser was tuned at 940 nm. Images (339 × 339 µm$^2$, 256 × 256 pixels) with 36 optical planes with planes spaced 2 µm apart in depth were collected every 31 s. During the imaging, norepinephrine (A0906, Tokyo Chemical Industry) was added to a final concentration of 0.5 µM.

## Electrophysiology for characterization of cAMPinG1

For characterization of cAMPinG1 in acute brain slices (Extended Data Fig. 3m–o), AAV (AAV2/1-eSyn-cAMPinG1-NE, AAVPHP.eB-hSyn-EGFP) was injected into the medial prefrontal cortex of mice aged 8 weeks at a volume of 500 nl. After 2 weeks of expression, mice were decapitated, and acute brain slices were prepared as described above. Slices were perfused with oxygenated ACSF (125 mM NaCl, 2.5 mM KCl, 2 mM CaCl$_2$, 1 mM MgCl$_2$, 1.25 mM NaH$_2$PO$_4$, 26 mM NaHCO$_3$, 20 mM glucose and 200 µM Trolox). Whole-cell recordings were performed by 5–6 MΩ glass pipettes. Patch pipettes were filled with an internal solution (120 mM potassium gluconate, 20 mM KCl, 10 mM disodium phosphocreatine, 4 mM ATP (magnesium salt), 0.3 mM GTP (sodium salt), 10 mM HEPES (pH 7.25, 293 mOsm). Electrophysiological data were

acquired using a patch-clamp amplifier (MultiClamp 700B, Molecular devices) and stored using a Digidata 1440A converter and pCLAMP software (Molecular Devices). To assess the excitability, the spike number was measured by injecting pulses of increased intensity in steps of 25 pA (from 0 to 250 pA, 500 ms duration). For miniature excitatory postsynaptic current measurement, tetrodotoxin (0.2 μM to a final concentration), APV (50 μM) and picrotoxin (25 μM) were added and membrane potential voltage clamped at −70 mV after correction of liquid-junction potential was used.

## Tissue preparation

Tissue blocks were cut into 50-μm-thick slices with a cryostat (CM1950, Leica) and floated in PBS. For nuclear staining, brain slices were incubated with DAPI for 10 min at room temperature before being mounted on slides. Imaging was performed using an LSM 880 confocal microscope with a ×20 objective lens and 405-nm, 488-nm and 561-nm lasers. For inclusion counting of $Ca^{2+}$ sensors, images ($142 \times 142$ μm$^2$, $512 \times 512$ pixels) were acquired across multiple optical planes, each spaced 2 μm apart in depth from L2/3 neurons in the V1.

## Simultaneous Ca²⁺ imaging and whole-cell recordings in acute brain slices

AAV (AAV2/1-eSyn-jRGECO1a, AAV2/1-eSyn-RCaMP3) was injected into the barrel cortex (A/P −1.0 mm, M/L −3.0 mm from the bregma, D/V −0.2 mm from the pial surface) at 20 nl min$^{-1}$ at a volume of 500 nl. After 4 weeks of expression, mice were killed by rapid decapitation after anesthesia with pentobarbital (100 mg per kg body weight). The brains were immediately extracted and immersed in gassed (95% $O_2$/5% $CO_2$) and ice-cold ACSF containing: 124 mM NaCl, 2.5 mM KCl, 1.25 mM NaH$_2$PO$_4$, 26 mM NaHCO$_3$, 2 mM CaCl$_2$, 2 mM MgCl$_2$ and 10.1 mM glucose. Acute coronal brain slices (300 μm thick) of the barrel cortex were cut in gassed, ice-cold ACSF with a vibratome (VT1200S, Leica). Brain slices were then transferred to an incubation chamber containing gassed ACSF at 30 °C for 60 min and subsequently maintained at room temperature before transferring them to the recording chamber at 35 °C.

Whole-cell recordings were performed in the L2/3 pyramidal neurons of the barrel cortex with glass recording electrodes (5–8 MΩ) filled with the intracellular solution containing: 130 mM K-gluconate, 4 mM NaCl, 10 mM HEPES, 4 mM Mg-ATP, 0.3 mM Na-GTP, 7 mM dipotassium-phosphocreatine and pH adjusted to 7.0 with potassium hydroxide (296 mOsm). Electrophysiological data were acquired using a patch-clamp amplifier (MultiClamp 700B, Molecular devices) filtered at 10 kHz and sampled at 20 kHz. Single action potentials were evoked by injecting a series of current pulses (2 ms in duration) through the patch pipette. Each trial was repeated, and the mean value was presented.

Calcium imaging was performed using an upright microscope (BX51WI, Olympus) with a water-immersion ×40 objective lens (N.A.: 0.8, Olympus). To acquire RCaMP3 and jRGECO1a images with LED light (MCWHLP1, Thorlabs), a U-MWIG3 fluorescence mirror unit (Olympus) was used. Fluorescent images were captured by a sCMOS camera (Orca-Flash 4.0 v3, Hamamatsu Photonics) controlled by HC Image software (Hamamatsu Photonics). Images were acquired at 50 Hz with 1 × 1 binning.

## Simultaneous calcium imaging and cell-attached recordings in vivo

AAV (AAV2/1-eSyn-jRGECO1a, AAV2/1-eSyn-RCaMP3) was injected into the barrel cortex (A/P −1.0 mm, M/L −3.0 mm from the bregma, D/V −0.2 mm from the pial surface) at 20 nl min$^{-1}$ at a volume of 500 nl. After 4 weeks of expression, mice were head-fixed and anesthetized with isoflurane (-1.5–2.0%) throughout the experiment, and body temperature was kept at 37 °C with a heating pad. A craniotomy was made in the barrel cortex. The exposed brain was covered with 1.5% agarose in ACSF containing the following: 150 mM NaCl, 2.5 mM KCl,

10 mM HEPES, 2 mM CaCl$_2$ and 1 mM MgCl$_2$, pH 7.3. A glass coverslip was then placed over the agarose to suppress the brain motion artifacts. A glass electrode (5–8 MΩ) was filled with ACSF containing Alexa 488 (200 μM). jRGECO1a or RCaMP3-expressing neurons were targeted using two-photon microscopy (Movable Objective Microscope, Sutter) with a tunable laser (InSight X3, Spectra-Physics) and a water-immersion ×16 objective lens (N.A.: 0.80, Nikon). Fluorescence signals were collected using a GaAsP photomultiplier tube (Hamamatsu Photonics) with a 590–660-nm emission filter. After establishing the cell-attached configuration (20–100 MΩ seal), simultaneous spike recording and calcium imaging were performed at the soma (sampling rate = 30 Hz, $512 \times 512$ pixels). Electrophysiological data were acquired using a patch-clamp amplifier (MultiClamp 700B; Molecular devices) in current-clamp mode, filtered at 10 kHz, and sampled at 20 kHz. The laser was tuned to 1,040 nm (40 mW at the front aperture of the objective).

## Fiber photometry

A 400-μm-diameter mono fiber-optic cannula (Kyocera) was implanted. A custom-made metal head plate was attached to the skull with dental cement. Mice were subjected to imaging after more than 2 days of the surgery.

Dual-color fiber photometry for green cAMP sensors and RCaMP3 was performed using the GCaMP and Red Fluorophore Fiber Photometry System (Doric) with 405-nm, 470-nm and 560-nm LED, 500–540-nm and 580–680-nm emission filter, and 400-μm-diameter 0.57-N.A. Mono Fiber-optic Patch Cords (Doric). Photometry data were recorded at a sampling rate of 30 Hz by lock-in amplifier detection. Mice were head-fixed during the recordings.

For cAMP imaging in the V1, on the first day, just after the first recording for 30-s forced running, 25 mg per kg body weight propranolol (168-28071, Wako) or mock solution was intraperitoneally administered. Thirty minutes later, the second recording was performed. On the subsequent day, the same two recordings were performed for the same mice, but with the reversed treatments. Half of the mice received propranolol on the first day and the other half received mock solution on the first day.

## Cranial window implantation

Mice were anesthetized by the anesthetic mixture described above. Before surgery, dexamethasone sodium phosphate (2 mg per kg body weight, Wako) and carprofen (5 mg per kg of body weight) were administered to prevent inflammation and pain. During surgery, mice were put on a heating pad, and body temperature was kept at 37 °C. A custom-made stainless-steel head plate was fixed to the skull using cyanoacrylate adhesive and dental cement (Sun Medical) above the right visual cortex. A craniotomy was drilled with a 2.5-mm diameter, and the brain was kept moist with saline. A cover glass (3-mm diameter, no. 0 thickness, Warner Instruments) was placed over the craniotomy site with surgical adhesive glue (Aron Alpha A, Sankyo)[49,50]. The mice were subjected to imaging more than 18 h after the surgery.

## In vivo two-photon imaging

In vivo two-photon imaging was performed with an FVMPE-RS (Olympus) equipped with a water-immersion ×25 objective (N.A.: 1.05, Olympus), a femtosecond laser (Insight DS+, Spectra-Physics) and two GaAsP detectors (Hamamatsu Photonics) with 495–540-nm and 575–645-nm emission filters (Olympus). For somatic cAMP imaging of cAMPinG1, cAMPinG1mut and G-Flamp1 expressed by in utero electroporation, images ($339 \times 339$ μm$^2$, $512 \times 512$ pixels, single optical section) were collected at 15 Hz in the awake condition. The laser was tuned to 940 nm (48.6 mW at the front aperture of the objective). For cAMPinG1-NE spine imaging, images ($28.8 \times 38.4$ μm$^2$, $96 \times 128$ pixels, single optical section) were collected at 7.5 Hz in the condition anesthetized lightly by isoflurane (0.5% vol/vol). The laser power was 23.5 mW at the front aperture

of the objective. For RCaMP3 and cAMPinG1-ST imaging, sequential excitation at 940 nm and 1,040 nm was used for dual-color imaging. Images (339 × 339 µm², 512 × 512 pixels) with three optical planes with plane spaced 30 µm apart in depth were collected at 3.4 Hz per plane using a piezo objective scanner (Olympus). The laser power was set to 47.6 mW for 940 nm excitation, and to 113.7 mW for 1,040 nm excitation. The imaging with visual stimuli (Fig. 5a–g) was followed by the imaging during forced running (Fig. 4) on the same cell population. For cAMP and Ca²⁺ imaging in astrocytes, images (288 × 384 µm², 192 × 256 pixels) were collected at 1.6 Hz in the awake condition. The laser power for 940 nm of excitation was set to 26.5 mW, and for 1,040 nm of excitation, it was set to 52.2 mW. For single-cell cAMP imaging with optical stimulation using soma-targeted ChRmine, images (86.4 × 115.2 µm², 192 × 256 pixels) were collected at 3.2 Hz in the awake condition with 940 nm of excitation. Next, 1,040-nm two-photon excitation was used for 4-s, 16-Hz spiral scanning with 11-µm diameter. The imaging with a 940-nm laser was temporally stopped during the optical excitation. The laser power for 940-nm excitation was set to 4.1 mW, and for 1,040 nm of photostimulation, it was set to 39.8 mW.

## Two-photon mesoscale Ca²⁺ imaging with FASHIO-2PM

AAV (AAV2/1-eSyn-RCaMP3) was injected into the neonatal somatosensory cortex[53]. After 8 weeks of AAV injection, a 4.5-mm-diameter craniotomy was performed over an area including the primary somatosensory area of the right hemisphere. A head plate was also fixed to the skull above the cerebellum.

Two-photon imaging was performed with FASHIO-2PM[29] equipped with a femtosecond laser (Chameleon Discovery, Coherent). The laser was tuned to 1,040 nm. The field of view was 3.0 × 3.0 mm² (2,048 × 2,048 pixels). The sampling rate was 7.5 Hz. Laser power of 270 mW and 360 mW at the front of the objective lens was used to observe L2/3 and L5 neurons of awake mice, respectively.

## Physical stimulation

Airpuff stimuli (2 Hz, 0.1 s duration, 40 times) were generated using a microinjector (BEX). For the forced running task, mice were head-fixed, and a custom-made treadmill was turned on during recordings. Moving grating stimuli were generated using the PsychoPy function in Python[34]. The gratings were presented with an LCD monitor (19.5 inches, 1,600 × 900 pixels, Dell), placed 25 cm in front of the center of the left eye of the mouse. Each stimulus trial consisted of a 4-s blank period (uniform gray at mean luminance) followed by a 4-s drifting sinusoidal grating (0.04 cycles per degree, 2 Hz temporal frequency). Eight drifting directions (separated by 45°, in order from 0° to 315°) were used. The timing of each moving grating stimulus and the initiation of imaging were monitored with a data acquisition module (USB-6343, National Instruments) controlled by LabVIEW (2021).

## Analysis of in vitro experiments

In vitro fluorometry analysis was performed using Python (https://www.python.org/) and Excel (Microsoft). For both Ca²⁺ and cAMP sensors, the $K_d$ value and Hill coefficient were calculated by fitting according to the Hill equation. For both Ca²⁺ and cAMP sensors, the pKa values were calculated by the inflection point of a sigmoid fit to fluorescence versus pH. For in vitro kinetics analysis (Extended Data Fig. 2g,h), half-rise time and half-decay time were calculated by single exponential fitting.

## Image analysis

Image analyses were performed with ImageJ (National Institutes of Health) and Python. For somatic cAMP imaging of cAMPinG1, cAMPinG-1mut and G-Flamp1 expressed by in utero electroporation (Fig. 2a–f), somatic cAMP imaging expressed by AAV (Extended Data Fig. 6a–d and Extended Data Fig. 7) and simultaneous RCaMP3 and cAMPinG1-ST imaging (Figs. 4 and 5a–g and Extended Data Fig. 7e), motion correction

was performed with Suite2p toolbox (https://github.com/MouseLand/suite2p/)[54]. For cAMPinG1-NE spine imaging (Fig. 2g–i), mesoscale Ca²⁺ imaging (Fig. 3o,p and Extended Data Fig. 5c,d), Ca²⁺ and cAMP imaging in astrocytes (Extended Data Fig. 6h–j) and cAMP imaging with optical stimulation using soma-targeted ChRmine (Fig. 5h–k), motion correction was performed with TurboReg[55].

Region of interest (ROI) detection for in vivo Ca²⁺ imaging was performed with Suite2p. ROI detection for HEK cell live imaging and in vivo cAMP imaging were performed with ImageJ and Cellpose[56]. ROIs for cAMPinG1-NE spine imaging (Fig. 2g–i), Ca²⁺ and cAMP in astrocytes (Extended Data Fig. 6h–j) and cAMPinG1-NE imaging with soma-targeted ChRmine (Fig. 5h–k) were drawn manually. The dynamic range was calculated as $\Delta F/F = (F − F_0)/F_0$, where $F_0$ was the average fluorescence intensity before stimulations after the subtraction of background fluorescence. No bleaching correction was performed in any analyses except Ca²⁺ imaging in acute brain slices (Fig. 3d–h). No fluorescence cross-talk correction was performed.

For cAMP imaging using acute brain slices (Extended Data Fig. 3), background subtraction was performed before calculating $\Delta F/F$. ROIs were drawn manually around somata (Extended Data Fig. 3a–h) or with Cellpose (Extended Data Fig. 3i–l) in the time-series-averaged image. $\Delta F/F$ was calculated as $(F − F_0)/F_0$, where $F$ is the fluorescence intensity at any timepoint and $F_0$ is the average fluorescence before the drug application.

For in vivo cAMPinG1-NE spine imaging (Fig. 2g–i), the period after stimulation was defined as a 15-s period starting 10 s after the end of airpuff stimulation.

For two-photon Ca²⁺ imaging in HEK293T cells (Fig. 3c), ROIs drawn based on mCerulean images with Cellpose were used for both Ca²⁺ sensor and mCerulean images. The red fluorescence intensity was divided by mCerulean fluorescence intensity in each cell for normalization.

For Ca²⁺ imaging using acute brain slices (Fig. 3d–h), background subtraction and bleach correction were performed before calculating $\Delta F/F$. ROIs were manually selected around somata in the time-series-averaged image. $\Delta F/F$ was calculated as $(F − F_0)/F_0$, where $F$ is the fluorescence intensity at any timepoint and $F_0$ is the resting baseline fluorescence measured 200 ms before stimulation. The peak amplitude was defined as the maximum value of $\Delta F/F$ after the stimuli. The rise and decay curves were fit to a single exponential. The rise time was defined as the time from the beginning of the stimulus to the timepoint of the peak fluorescence amplitude. The half-decay time was defined as the time from the maximum value of $\Delta F/F$ to half of that value.

For simultaneous Ca²⁺ imaging and cell-attached recordings in vivo (Fig. 3i–n), ROIs were manually selected around somata in the time-series-averaged image. $\Delta F/F$ was calculated as $(F−F_0)/F_0$, where $F$ is the fluorescence intensity at any timepoint and $F_0$ is resting baseline fluorescence measured 200 ms before the action potentials. Action potentials were detected by cell-attached recording of the signal. Spike events (1–4 action potentials) were identified, ensuring that no other action potentials occurred in the 1-s period before and after the first action potential. The rise and decay curves were fit to a single exponential. The half-rise time was defined as the time from the beginning of the stimulus to the timepoint of the half of the peak fluorescence amplitude. The half-decay time was defined as the time from the maximum value of $\Delta F/F$ to half of that value.

For in vivo Ca²⁺ and cAMP imaging (Figs. 4 and 5a–g and Extended Data Figs. 6a–d and 7), ROIs for RCaMP3 and cAMP indicators were drawn independently using Suite2p and Cellpose, respectively. For simultaneous RCaMP3 and cAMPinG1-ST imaging (Figs. 4 and 5a–g), the cells that had ROIs for both RCaMP3 and cAMPinG1-ST were selected for further analysis. Because the imaging using visual stimulus (Fig. 5a–g) was followed by the imaging using forced running (Fig. 4) on the same cell population, the same ROIs were used for both analyses. Motion-related neurons were defined as neurons that showed Ca²⁺ $\Delta F/F$ values of more than 0.3 during the running period (Fig. 4).

# Article

$Ca^{2+}$ responses to 4 s of visual stimulation were defined as averaged $\Delta F/F$ during the 4-s stimulation. cAMP responses to 4 s of visual stimulation were defined as averaged $\Delta F/F$ during 2-s periods that started 2 s after the end of the visual stimulus (Fig. 5a–d). Neurons that responded to repetitive 8-s visual stimuli were defined as neurons that showed $Ca^{2+}$ $\Delta F/F$ values of more than 0.3 during the stimulation period (Fig. 5e–g). The period after repetitive visual stimuli was defined as a 40-s period starting after the end of the stimuli. The DSI was calculated for cells showing $Ca^{2+}$ responses. The preferred direction ($\theta_{pref}$) of each cell was defined as the stimulus that induced the largest $Ca^{2+}$ $\Delta F/F$. The DSI was defined as DSI = $(R_{pref} - R_{pref+\pi})/(R_{pref} + R_{pref+\pi})$, where $R_{pref}$ and $R_{pref+\pi}$ are $\Delta F/F$ values at the preferred ($\theta_{pref}$) and the opposite ($\theta_{pref} + \pi$) directions, respectively. Imaging frames with notable motion artifacts were removed and supplied with the preceding frames.

For cAMPinG1-NE imaging with soma-targeted ChRmine (Fig. 5k), cAMP $\Delta F/F$ was defined as averaged $\Delta F/F$ during a 10-s period starting 20 s after the end of optical stimulation.

### Analysis of fiber photometry
Fiber photometry analysis was performed using Python. The cAMPinG1 signal was calculated as follows: (470-nm signal)/(405-nm signal). The 560-nm signal was recognized as the RCaMP3 signal.

For imaging in the V1 (Extended Data Fig. 6e–g), $\Delta R/R$ was calculated as $(R - R_0)/R_0$, where $F$ is mean fluorescence intensity in the last 10 s of the running period and $F_0$ is the mean fluorescence intensity in 10 s before the start of running. $\Delta R/R$ (pre) and $\Delta R/R$ (post) were defined as $\Delta R/R$ in response to running before and after the intraperitoneal administration, respectively.

For imaging in the dorsal striatum (Extended Data Fig. 8), the period during stimulation was defined as a 10-s running period, and the period after stimulation was defined as 10 s after the end of the running period.

### Statics and reproducibility
All experiments were conducted with at least two biological replicates, often more, involving independent transfection and mice. All statistical analyses of the acquired data were performed with Python. For each figure, a statistical test matching the structure of the experiment and the structure of the data was employed. All tests were two tailed. *$P < 0.05$; **$P < 0.01$; ***$P < 0.001$; NS, not significant ($P > 0.05$) for all statistical analyses presented in figures. No statistical tests were done to predetermine the sample size. Data acquirement and analysis were not performed blind to the conditions of the experiments. Experimental sample sizes are mentioned in the figure panel and legends.

### Reporting summary
Further information on research design is available in the Nature Portfolio Reporting Summary linked to this article.

## Data availability
The cAMPinG1 and RCaMP3 sequences are available from GenBank (accession number: LC728270 (cAMPinG1) and LC728271 (RCaMP3)). The plasmids generated in this study are available from RIKEN Bio-Resource Center (https://dna.brc.riken.jp/en/) with the following catalog numbers: pN1-cAMPinG1 (RDB19691), pAAV-eSyn-cAMPinG1-NE-WPRE (RDB19692), pAAV-eSyn-cAMPinG1-ST-WPRE (RDB19693) and pAAV-eSyn-RCaMP3-WPRE (RDB19694). Source data are provided with this paper.

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

## Acknowledgements
We thank the following researchers for kindly sharing their reagents: H. Bito, The University of Tokyo (eSyn promoter, cHS4, XCaMP-R and pCAG-Flpo plasmids); N. Saito, Doshisha University (gCarvi plasmid, and a protocol about kinetics analysis); T. Jacks, Massachusetts Institute of Technology (Addgene plasmid, 12093); V. Verkhusha, Albert Einstein College of Medicine (Addgene plasmid, 45457); J. Kim, Korea Institute of Science and Technology (Addgene plasmid, 51904); B. Khakh, the University of California, Los Angeles (Addgene plasmid, 52924); T. Kitaguchi, Tokyo Institute of Technology (Addgene plasmid, 73938); D. Kim, Janelia Research Campus (Addgene plasmid, 100854); V. Gradinaru, California Institute of Technology (Addgene plasmid, 103005); R. Campbell, The University of Tokyo (Addgene plasmid, 105864); J. Garrison, Buck Institute (Addgene plasmid, 158777); H. Zhong, Vollum Institute (Addgene plasmid, 182283); and B. Roth, University of North Carolina at Chapel Hill (Addgene kit, 1000000068). We also thank Y. Kato for technical assistance and members of the laboratories of M.S. and I.I. for their support and comments.

This work was supported in part by grants from Precursory Research for Embryonic Science and Technology (PRESTO)-JST (JPMJPR1906 to M.S.), ACT-X-JST (JPMJAX211K to T.Y.), CREST-JST (JPMJCR1921 to I.I.), Brain Mapping by Integrated Neurotechnologies for Disease Studies (Brain/MINDS) (JP19dm0207079 to S.M., JP19dm0207090 to I.I., JP19dm0207069 to S.Y., JP15dm0207001 to M.M., JP19dm0207080 to K.K. and JP17dm0207059 to M.S.), Interdisciplinary and Emerging Brain Research Program (iBrain/MINDS) (JP21wm0525018 to S.Y. and JP21wm0525004 to M.S.), Program for Technological Innovation of Regenerative Medicine (JP21bm0704060 to I.I.), AMED-PRIME (JP 23gm6510022 to M.S.), Interstellar Initiative Beyond (JP22jm0610068 to M.S.), JSPS KAKENHI (JP21K15207 to

T.Y., JP22H02718 to S.M., JP21H02485 to I.I., JP20H05775 to M.M., JP22H05161 and JP22H00460 to K.K. and JP23H02782, JP22H04922, JP21K19429 and JP20H04122 to M.S.), RIKEN Special Postdoctoral Researchers Program (to H.U.), Inamori Foundation (to M.S.), Takeda Science Foundation (to M.S.), Lotte Foundation (to M.S.), The Konica Minolta Science and Technology Foundation (to M.S.), Tokyo Biochemical Research Foundation (to M.S.), Brain Science Foundation (to M.S.), Mochida Memorial Foundation (to M.S.) and KOSÉ Cosmetology Research Foundation (to M.S.). This work was also supported by Kyoto University Live Imaging Center.

## Author contributions

T.Y. designed and developed cAMP sensors. T.Y. and M.S. designed and developed the RCaMP3. T.Y. performed most of the experiments and analyzed data. S.M. and K.K. performed electrophysiological recordings for RCaMP3 characterization. U.H. and M.M. performed $Ca^{2+}$ imaging with the FASHIO-2PM. M.T. and S.Y. performed electrophysiological and pharmacological experiments for cAMPinG1 in acute brain slices. T.Y. and M.S. designed the study and wrote the paper with input from all authors. I.I. and M.S. supervised the entire project.

## Competing interests

T.Y. and M.S. have filed a patent related to cAMPinG1. The remaining authors declare no competing interests.

## Additional information

**Extended data** is available for this paper at https://doi.org/10.1038/s41592-024-02222-9.

**Correspondence and requests for materials** should be addressed to Tatsushi Yokoyama or Masayuki Sakamoto.

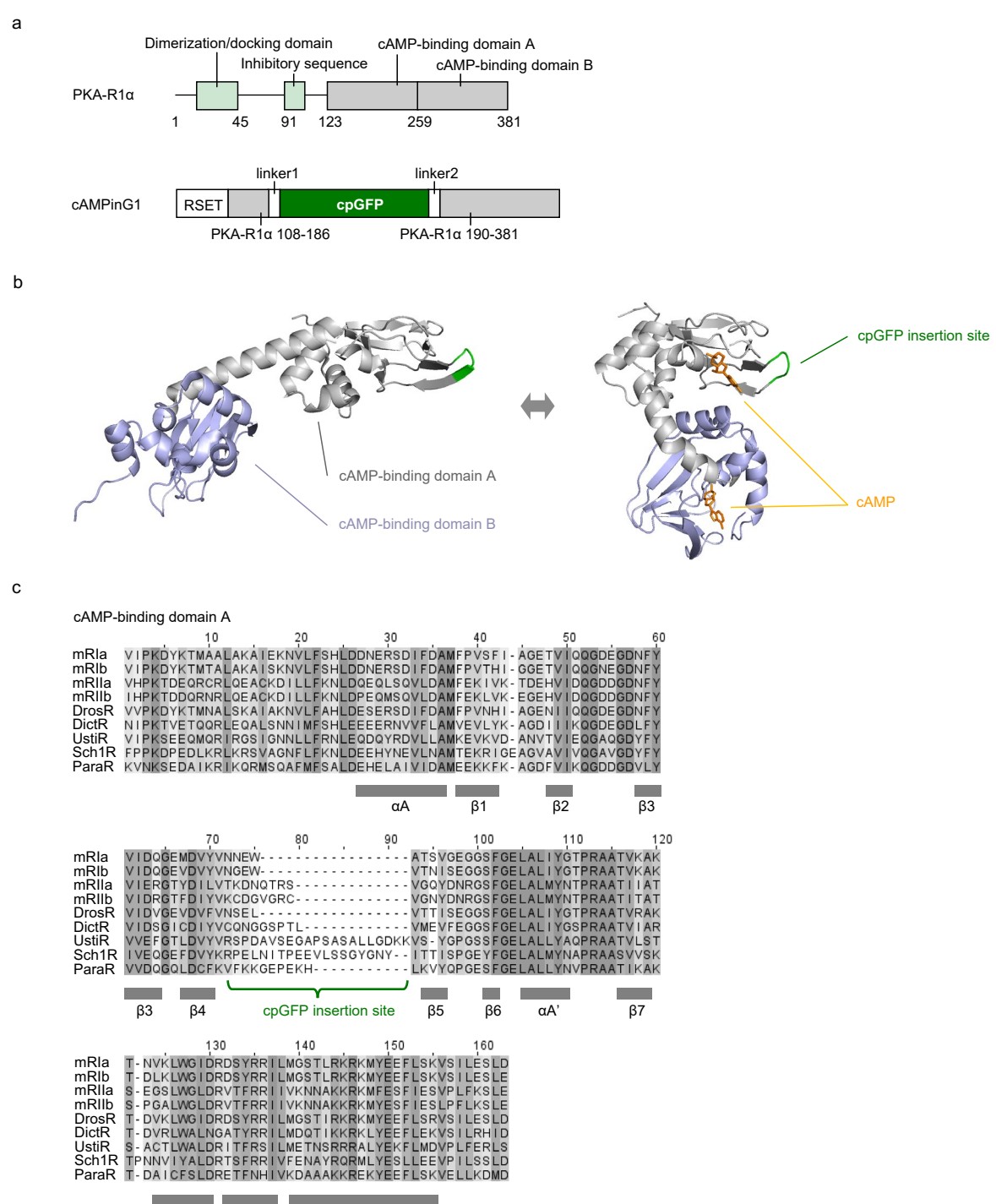

**Extended Data Fig. 1 | Design of cAMPinG1. a**, Primary structures of mouse PKA regulatory-subunit 1α (PKA-R1α) (top) and cAMPinG1 (bottom). We removed the N-terminal PKA-R1α region, which includes a dimerization/docking domain interacting with scaffold proteins and an inhibitory domain interacting with PKA catalytic subunits (PKA-C), to avoid interaction with these endogenous proteins. Instead, we fused the RSET sequence to the N-terminus of the sensor to promote stable expression. **b**, Tertiary structures of cAMP-free (left, PDB: 2QCS, catalytic-subunit is hidden) and cAMP-binding (right, PDB: 1RGS) PKA R1α. Note the cpGFP insertion site is close to cAMP in the cAMP-bound three-dimensional structure and exposed on the surface. **c**, The amino acid sequence of cAMP-binding domains of PKA regulatory-subunit. mR1a, *Mus muslucus* PKA-RIa; mR1b, *Mus musculus* PKA-R1b; mR2a, *Mus musculus* PKA-R2a; mR2b, *Mus musculus* PKA-R2b; DrosR, *Drosophila melanogaster* PKA-R; DictR, *Dictyostelium discoideum* PKA-R; UstiR, *Ustilago maydis* PKA-R; Sch1R, *Schizosaccharomyces pombe* PKA-R; ParaR, *Paramecium tetraurelia* PKA-R.

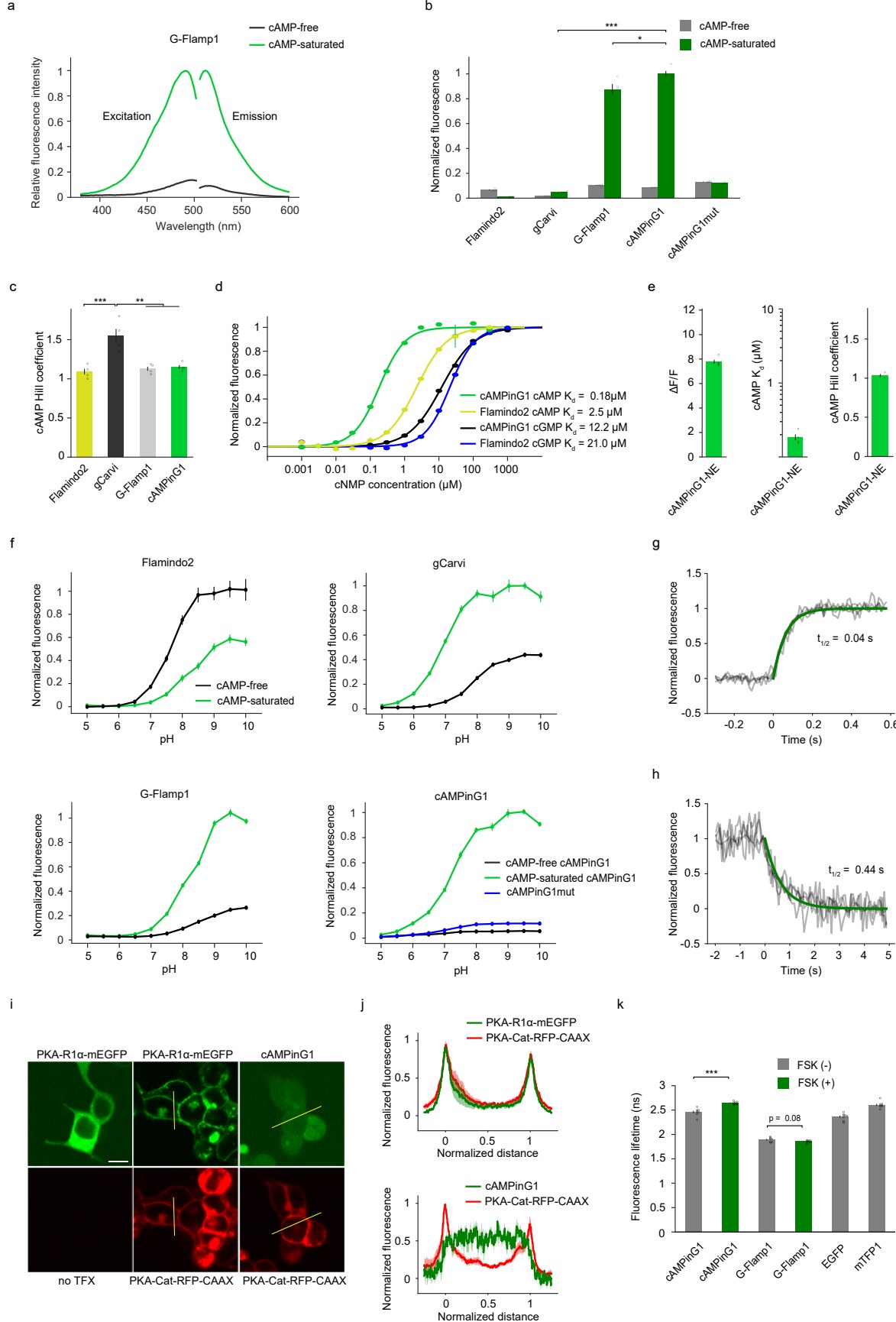

**Extended Data Fig. 2 | See next page for caption.**

**Extended Data Fig. 2 | Characterization of cAMP sensors in vitro.**
**a**, Excitation and emission spectra of cAMP-free and cAMP-saturated G-Flamp1.
**b**, Fluorescence intensities of cAMP sensors in cAMP-free and cAMP-saturated
states in HEK293T cell lysate. n = 4 wells in each sensor. Tukey's post hoc test
following one-way ANOVA. gCarvi vs cAMPinG1: $P = 4.2 \times 10^{-13}$; G-Flamp1 vs
cAMPinG1: $P = 2.4 \times 10^{-2}$. **c**, Hill coefficients of cAMP sensors. n = 4 wells in each
sensor. Tukey's post hoc test following one-way ANOVA. Flamindo2 vs gCarvi:
$P = 6.3 \times 10^{-4}$; gCarvi vs G-Flamp1: $P = 1.3 \times 10^{-3}$; gCarvi vs cAMPinG1: $P = 2.1 \times 10^{-3}$.
**d**, cGMP and cAMP titration curves of Flamindo2 and cAMPinG1. n = 4 wells in
each condition. Flamindo2: cGMP $K_d$ = 21.0 μM, cAMP $K_d$ = 2.5 μM, cAMPinG1:
cGMP $K_d$ = 12.2 μM, cAMP $K_d$ = 0.18 μM. **e**, $\Delta F/F$, cAMP $K_d$ value, and Hill coefficient
of cAMPinG1-NE. n = 4 wells in each condition. **f**, pH titration of cAMP-free and
cAMP-saturated cAMP sensors. n = 4 wells in each condition. cAMP-saturated
Flamindo2: $pK_a$ = 8.2, cAMP-free Flamindo2: $pK_a$ = 7.6, cAMP-saturated gCarvi:
$pK_a$ = 8.2, cAMP-free gCarvi: $pK_a$ = 8.5, cAMP-saturated G-flamp1: $pK_a$ = 6.9,
cAMP-free G-flamp1: $pK_a$ = 7.9, cAMP-saturated cAMPinG1: $pK_a$ = 7.2, cAMP-free

cAMPinG1: $pK_a$ = 6.5, cAMPinG1mut: $pK_a$ = 7.0. **g**, In vitro binding kinetics of
cAMPinG1 in response to cAMP microperfusion from 0 μM to 2 μM. n = 4 trials.
Half rise time = 0.04 s, $k_{on}$ = 8.0 × 10$^6$ (M$^{-1}$s$^{-1}$). **h**, In vitro dissociation kinetics of
cAMPinG1 in response to cAMP microperfusion from 0.5 μM to 0 μM. n = 4 trials.
Half decay time = 0.44 s, $k_{off}$ = 1.6 (s$^{-1}$). **i**, Representative images of binding assay
of PKA-R1α-mEGFP and cAMPinG1 with PKA-catalytic subunit (Cat)-RFP-CAAX
in HEK293T cells. Lines are for quantification of fluorescence distribution for
(j). Scale bar, 10 μm. **j**, Quantification of line fluorescence distribution. The
membrane location, as detected by the red fluorescence of PKA-Cat-RFP-CAAX, is
set at 0 and 1 on the x-axis. n = 5 cells (PKA-R1α-mEGFP), 5 cells (cAMPinG1). Scale
bar, 10 μm. **k**, Fluorescence lifetime of cAMP sensors and fluorescence proteins
expressed in HEK293T cells. n = 8 FOVs in each condition. Unpaired two-tailed
t-test. cAMPinG1 in the absence vs presence of 10 μM forskolin: $P = 1.8 \times 10^{-5}$;
G-Flamp1 in the absence vs presence of 10 μM forskolin: $P = 8.5 \times 10^{-2}$. All shaded
areas and error bars denote SEM.

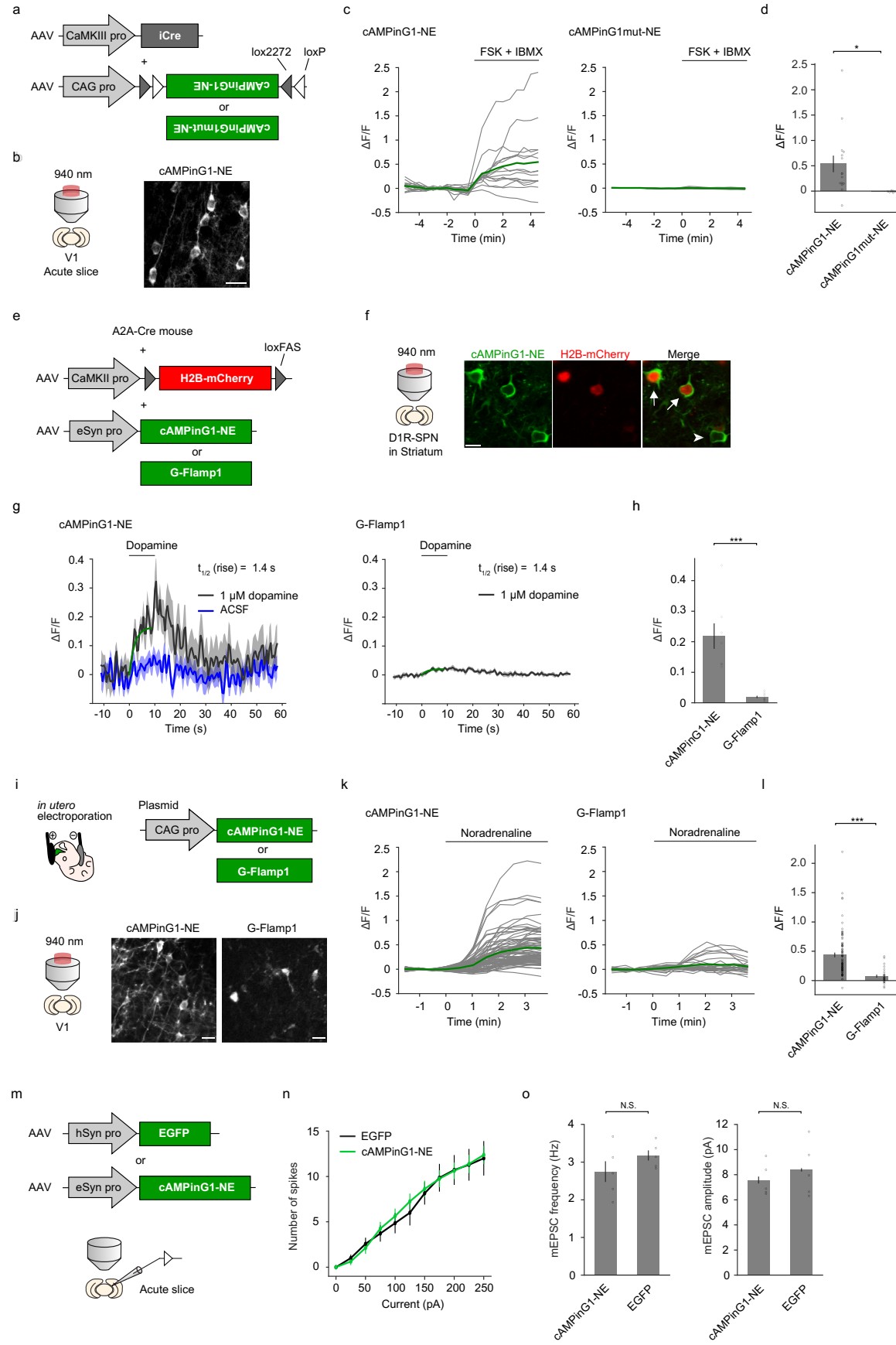

**Extended Data Fig. 3 | See next page for caption.**

**Extended Data Fig. 3 | Characterization of cAMP sensors in acute brain slices.**
**a**, Schematic of AAVs for sparse expression. **b**, Schematic of imaging settings
(left). A representative image (right). Scale bar, 20 μm. **c**, Traces of cAMPinG1-
NE (left) and cAMPinG1mut-NE (right) in response to 25 μM forskolin and
50 μM IBMX. Thin lines denote individual trace, and colored thick lines denote
average response. n = 16 neurons in 5 slices (cAMPinG1-NE), 12 neurons in 2 slices
(cAMPinG1mut-NE). **d**, ΔF/F for the forskolin (FSK) and IBMX application. n = 16
neurons in 5 slices (cAMPinG1-NE), n = 12 neurons in 2 slices (cAMPinG1mut-
NE). Unpaired two-tailed $t$-test. $P = 1.2 \times 10^{-2}$. **e**, Schematic of "Cre-Off" system
for expressing cAMP sensors in all neurons and mCherry in dopamine receptor
D1-stratal projection neurons (D1R-SPNs) in the striatum. **f**, Schematic of
imaging settings (left) and representative images of cAMPinG1-NE in an acute
brain slice (right). Fluorescence in response to 10 s local puff of 1 μM dopamine
was recorded in mCherry-positive D1R-SPNs (arrow). Dopamine receptor D2
(D2R)-positive neurons were not labeled with mCherry (arrowhead) due to the
expression of Cre recombinase. Scale bar, 10 μm. **g**, Traces of cAMPinG1-NE
to dopamine (left), cAMPinG1-NE to ACSF (left), and G-Flamp1 to dopamine
(right). The responses to dopamine were aligned with a single-exponential

fitting. Half rise time = 1.4 s (cAMPinG1-NE), 1.4 s (G-Flamp1). n = 7 neurons
in 3 slices (cAMPinG1-NE to dopamine), n = 11 neurons in 4 slices (cAMPinG1-
NE to ACSF), n = 9 neurons in 4 slices (G-Flamp1 to dopamine). **h**, ΔF/F in
response to dopamine. n = 7 neurons in 3 slices (cAMPinG1-NE), n = 9 neurons
in 4 slices (G-Flamp1). Unpaired two-tailed $t$-test. $P = 1.2 \times 10^{-3}$. **i**, Schematic
of the experimental procedure. **j**, Schematic of imaging settings (left) and
representative images (right). Scale bar, 20 μm. **k**, Traces of cAMPinG1-NE and
G-Flamp1 in response to 0.5 μM noradrenaline. Thin lines denote individual trace,
and thick lines denote average response. n = 84 neurons in 4 slices (cAMPinG1-
NE), 26 neurons in 4 slices (G-Flamp1). **l**, ΔF/F in response to noradrenaline.
n = 84 neurons in 4 slices (cAMPinG1-NE), 26 neurons in 4 slices from (G-Flamp1).
Unpaired two-tailed $t$-test. $P = 4.6 \times 10^{-6}$. **m**, Schematic of the experimental
procedure. **n**, Number of spikes evoked by current injection in expressing
cAMPinG1-NE and EGFP neurons. n = 8 neurons in each condition. **o**, mEPSC
(miniature excitatory postsynaptic current) frequency (left) and amplitude
(right) in cAMPinG1-NE and EGFP expressing neurons. n = 5 neurons in each
condition. Unpaired two-tailed $t$-test. Frequency: $P = 2.5 \times 10^{-1}$; Amplitude:
$P = 4.9 \times 10^{-1}$. All shaded areas and error bars denote SEM.

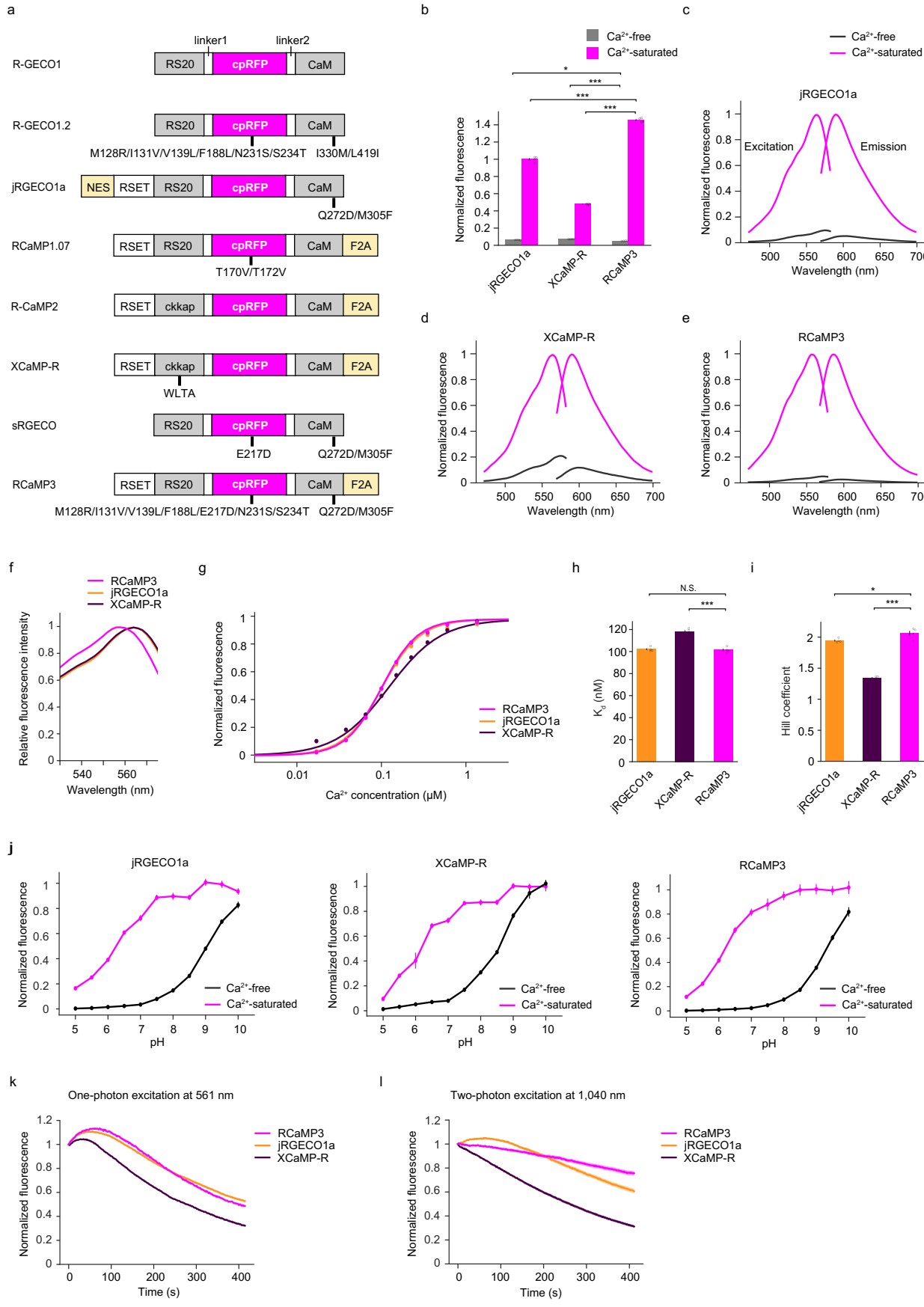

**Extended Data Fig. 4 | See next page for caption.**

**Extended Data Fig. 4 | Characterization of red Ca²⁺ sensors. a**, Primary structures of red Ca²⁺ sensors. Mutations are indicated in R-GECO1 numbering. **b**, Fluorescence intensities of red Ca²⁺ sensors in Ca²⁺-free and Ca²⁺-saturated conditions in HEK293T cell lysate. n = 4 wells in each sensor. Tukey's post hoc test following one-way ANOVA. Ca²⁺-free jRGECO1a vs Ca²⁺-free RCaMP3: $P = 2.6 \times 10^{-5}$; Ca²⁺-free XCaMP-R vs Ca²⁺-free RCaMP3: $P = 1.9 \times 10^{-7}$. Ca²⁺-saturated jRGECO1a vs Ca²⁺-saturated RCaMP3: P = $1.3 \times 10^{-9}$; Ca²⁺-saturated XCaMP-R vs Ca²⁺-saturated RCaMP3: $P = 1.4 \times 10^{-12}$. **c-e**, Excitation and emission spectra of jRGECO1a, XCaMP-R, and RCaMP3 in Ca²⁺-free and Ca²⁺-saturated states. n = 4 wells in each condition. **f**, Comparison of excitation spectra of red Ca²⁺ indicators. **g-i**, Ca²⁺ titration curves (g), $K_d$ values (h), and Hill coefficients (i) of red Ca²⁺ sensors. n = 4 wells in each sensor. Tukey's post hoc test following one-way ANOVA. $K_d$ values

(h) of jRGECO1a vs RCaMP3: $P = 9.0 \times 10^{-1}$; XCaMP-R vs RCaMP3: $P = 8.0 \times 10^{-6}$. Hill coefficients (i) of jRGECO1a vs RCaMP3: $P = 3.8 \times 10^{-2}$; XCaMP-R vs RCaMP3: P = $6.1 \times 10^{-8}$.**j**, Normalized fluorescence of red Ca²⁺ sensors in Ca²⁺-free and Ca²⁺-saturated states as a function of pH. Ca²⁺-saturated jRGECO1a: pKa = 6.3, Ca²⁺-free jRGECO1a: pKa = 9.0, Ca²⁺-saturated XCaMP-R: pKa = 5.3, Ca²⁺-free XCaMP-R: pKa = 8.6, Ca²⁺-saturated RCaMP3: pKa = 6.1, Ca²⁺-free RCaMP3: pKa = 9.4. n = 4 wells in each condition. **k,l**, One-photon (k) and two-photon (l) bleaching curves of jRGECO1a, XCaMP-R and RCaMP3 expressed in HEK293T cells. n = 63 neurons (jRGECO1a, one-photon), n = 91 neurons (XCaMP-R, one-photon), n = 76 neurons (RCaMP3, one-photon), n = 94 neurons (jRGECO1a, two-photon), n = 98 neurons (XCaMP-R, two-photon), n = 93 neurons (RCaMP3, two-photon). All shaded areas and error bars denote SEM.

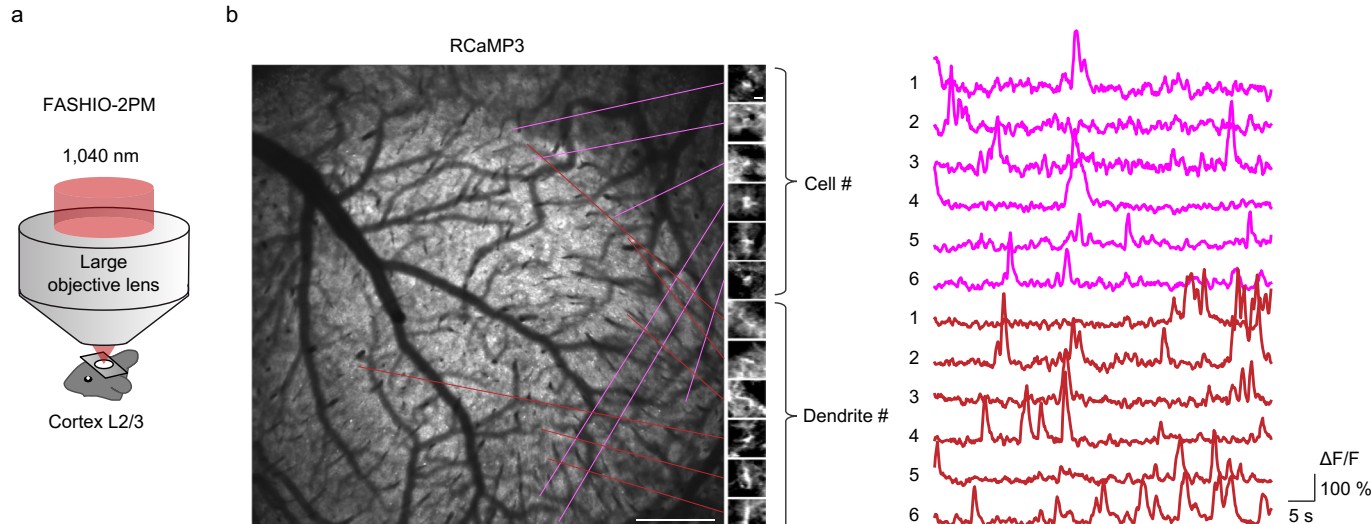

**Extended Data Fig. 5 | Two-photon mesoscale RCaMP3 imaging in layer 2/3.**
**a**, Schematic of the experimental procedure of two-photon mesoscale Ca²⁺ imaging using fast-scanning high optical invariant two-photon microscopy (FASHIO-2PM). **b**, Left: A representative full FOV of FASHIO-2PM. Scale bar, 500 μm. Right: Magnified images and Ca²⁺ traces of representative 6 somata and 6 dendrites. Scale bar, 10 μm.

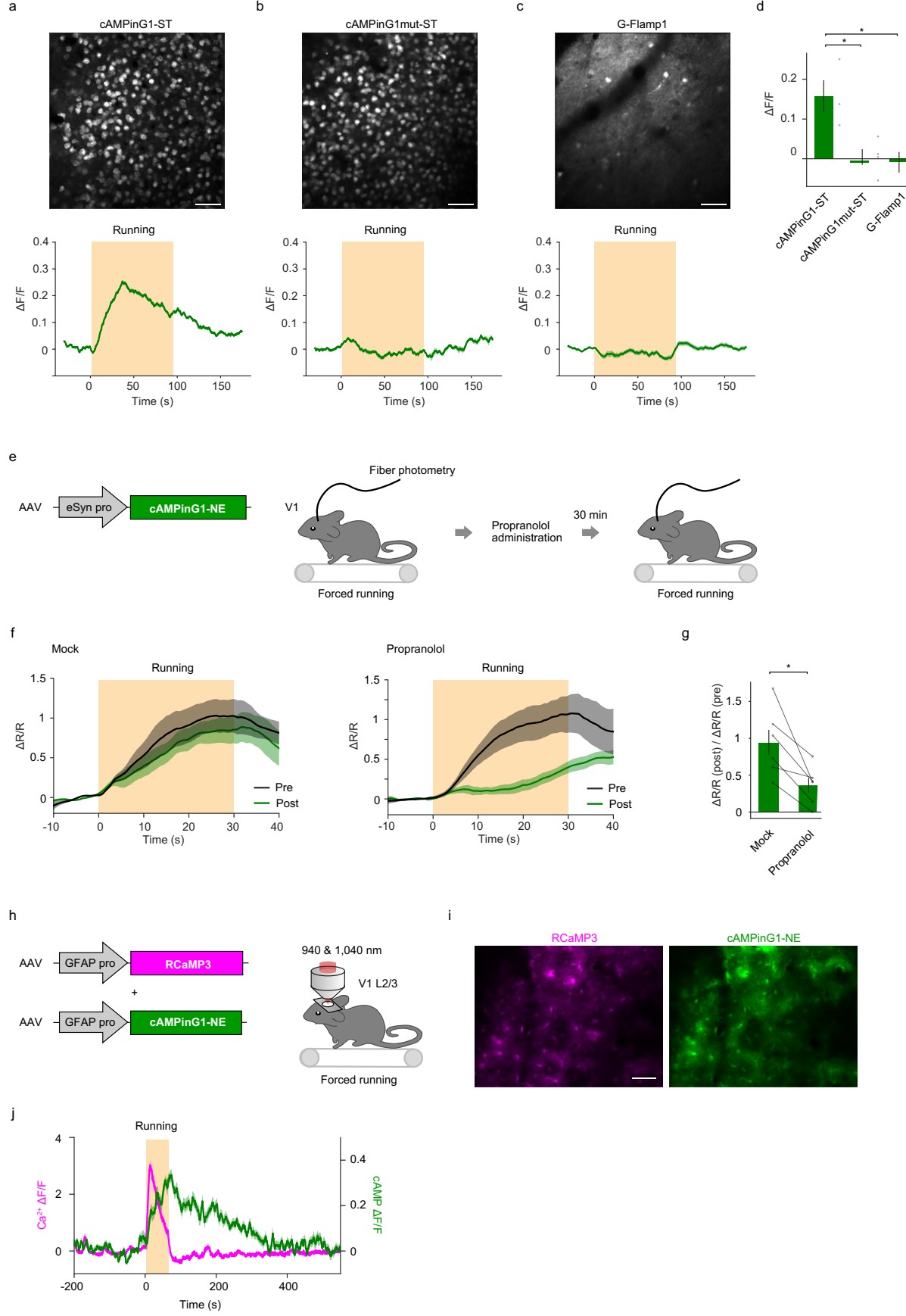

**Extended Data Fig. 6 | See next page for caption.**

**Extended Data Fig. 6 | In vivo imaging during forced running.**
**a-c**, Representative two-photon images and averaged traces of cAMPinG1-ST (a), cAMPinG1mut-ST (b), and G-Flamp1 (c) in response to forced running. n = 1156 cells in 3 mice (cAMPinG1-ST), n = 905 cells in 4 mice (cAMPinG1mut-ST), n = 74 cells in 4 mice (G-Flamp1). Scale bar, 50 μm. **d**, Averaged $\Delta F/F$ of cAMPinG1-ST, cAMPinG1mut-ST, and G-Flamp1 in response to forced running. n = 3 trials in 3 mice (cAMPinG1-ST), n = 4 trials in 4 mice (cAMPinG1mut-ST), n = 4 trials in 4 mice (G-Flamp1). Unpaired two-tailed $t$-test. cAMPinG1-ST vs cAMPinG1mut-ST: $P = 2.6 \times 10^{-2}$, cAMPinG1-ST vs G-Flamp1: $P = 4.6 \times 10^{-2}$. **e**, Schematic of the experimental procedure. cAMPinG1-NE was expressed in neurons in the V1. Fiber photometry was performed during forced running. Thirty minutes after intraperitoneal administration of propranolol, a β-adrenoceptor blocker, fiber photometry was again performed during the forced running. **f**, Averaged traces of cAMPinG1-NE in response to forced running before and after intraperitoneal administration of mock solution (left) or propranolol (right). n = 6 trials in 6 mice (mock), n = 6 trials in 6 mice (propranolol). **g**, Difference of $\Delta F/F$ in response to forced running before and after the intraperitoneal administration. n = 6 trials in 6 mice (mock), n = 6 trials in 6 mice (propranolol). Paired two-tailed $t$-test. $P = 1.4 \times 10^{-2}$. **h**, Schematic of the experimental procedure. **i**, Representative images of cAMPinG1 and RCaMP3. Scale bar, 50 μm. **j**, Averaged fluorescence transients of cAMPinG1 and RCaMP3. n = 63 cells in 3 mice. All shaded areas and error bars denote SEM.

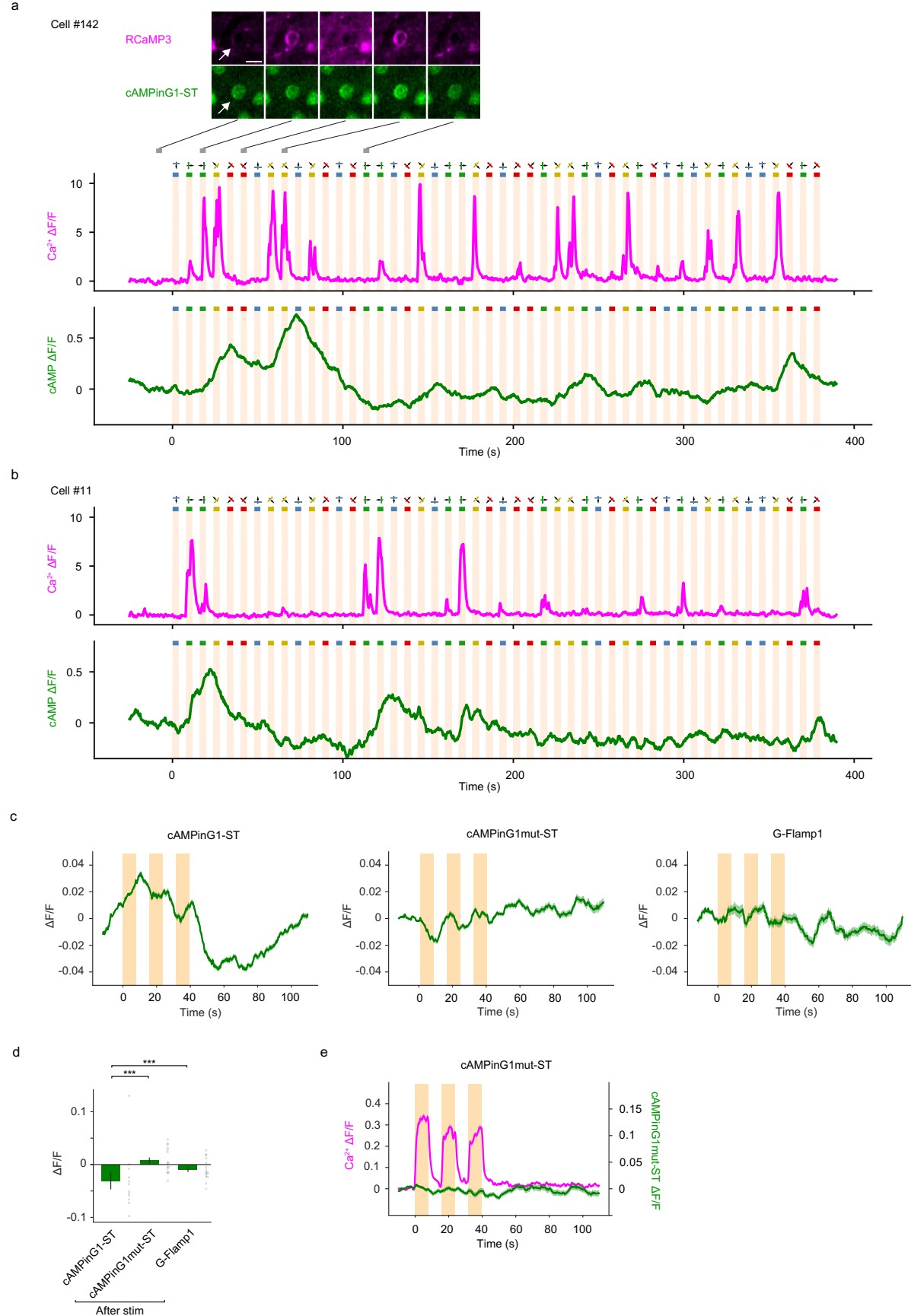

**Extended Data Fig. 7 | See next page for caption.**

**Extended Data Fig. 7 | In vivo Ca²⁺ and cAMP imaging during visual stimuli.**
**a**,**b**, Representative two-photon images and traces of RCaMP3 and cAMPinG1-ST of the cells in Fig. 5b-c, indicated by arrows. Scale bar, 10 μm. **c**, Averaged traces of cAMPinG1-ST (left), cAMPinG1mut-ST (middle), and G-Flamp1 (right) in response to visual stimuli. n = 1156 cells in 3 mice (cAMPinG1-ST), n = 905 cells in 4 mice (cAMPinG1mut-ST), n = 74 cells in 4 mice (G-Flamp1). Four trials in each cell. **d**, Averaged ΔF/F of cAMPinG1-ST, cAMPinG1mut-ST, and G-Flamp1 in response to visual stimuli. n = 12 trials in 3 mice (cAMPinG1-ST), n = 16 trials in 4 mice (cAMPinG1mut-ST), n = 16 trials in 4 mice (G-Flamp1). Unpaired two-tailed $t$-test after removing outliers with Smirnov-Grubbs' test ($P$ = 0.05). cAMPinG1-ST vs cAMPinG1mut-ST: $P$ = 1.9 × 10⁻⁵, cAMPinG1-ST vs G-Flamp1: $P$ = 7.2 × 10⁻⁴.
**e**, Averaged ΔF/F of RCaMP3 and cAMPinG1mut-ST in cells which showed Ca²⁺ response to visual stimuli. n = 486 cells in 4 mice. All shaded areas and error bars denote SEM.

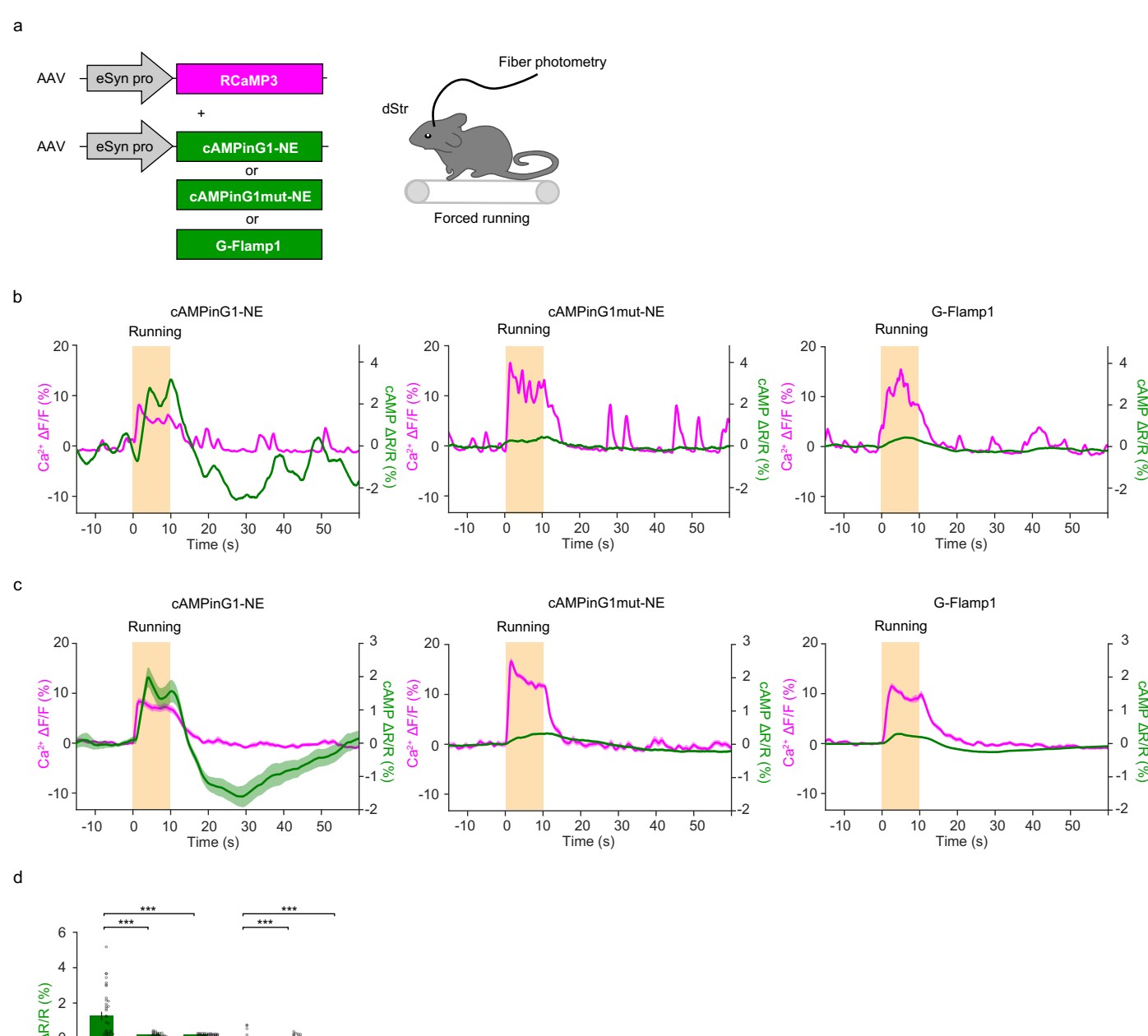

**Extended Data Fig. 8 | Dual-color fiber photometry for Ca²⁺ and cAMP in dorsal striatum neurons. a**, Schematic of the experimental procedure. AAVs encoding RCaMP3 and green cAMP sensors were injected into the dorsal striatum (dStr). Dual-color fiber photometry was performed in the dStr during a forced running task. For dual-color imaging, we employed different excitation wavelengths: 560 nm for RCaMP3 imaging and 405 nm and 470 nm for cAMPinG1 ratiometric imaging. **b**, Representative single-trial traces of cAMPinG1-NE (left), cAMPinG1mut-NE (middle), G-Flamp1 (right), and RCaMP3 signals. **c**, Averaged fluorescence traces of cAMPinG1-NE (left), cAMPinG1mut-NE (middle), G-Flamp1

(right), and RCaMP3 signals. n = 9 trials in 1 mouse (cAMPinG1-NE), n = 9 trials in 1 mouse (cAMPinG1mut-NE), n = 9 trials in 1 mouse (G-Flamp1). **d**, Averaged $\Delta F/F$ of cAMPinG1-NE, cAMPinG1mut-NE and G-Flamp1 during and after the stimulation. n = 36 trials in 4 mice (cAMPinG1-NE), n = 27 trials in 3 mice (cAMPinG1mut-NE), n = 36 trials in 4 mice (G-Flamp1). Unpaired two-tailed $t$-test. cAMPinG1-NE vs cAMPinG1mut-NE (during stim): $P = 3.7 \times 10^{-4}$; cAMPinG1-NE vs G-Flamp1 (during stim): $P = 3.6 \times 10^{-5}$; cAMPinG1-NE vs cAMPinG1mut-NE (after stim): $P = 3.1 \times 10^{-5}$; cAMPinG1-NE vs G-Flamp1 (after stim): $P = 4.2 \times 10^{-5}$. All shaded areas and error bars denote SEM.

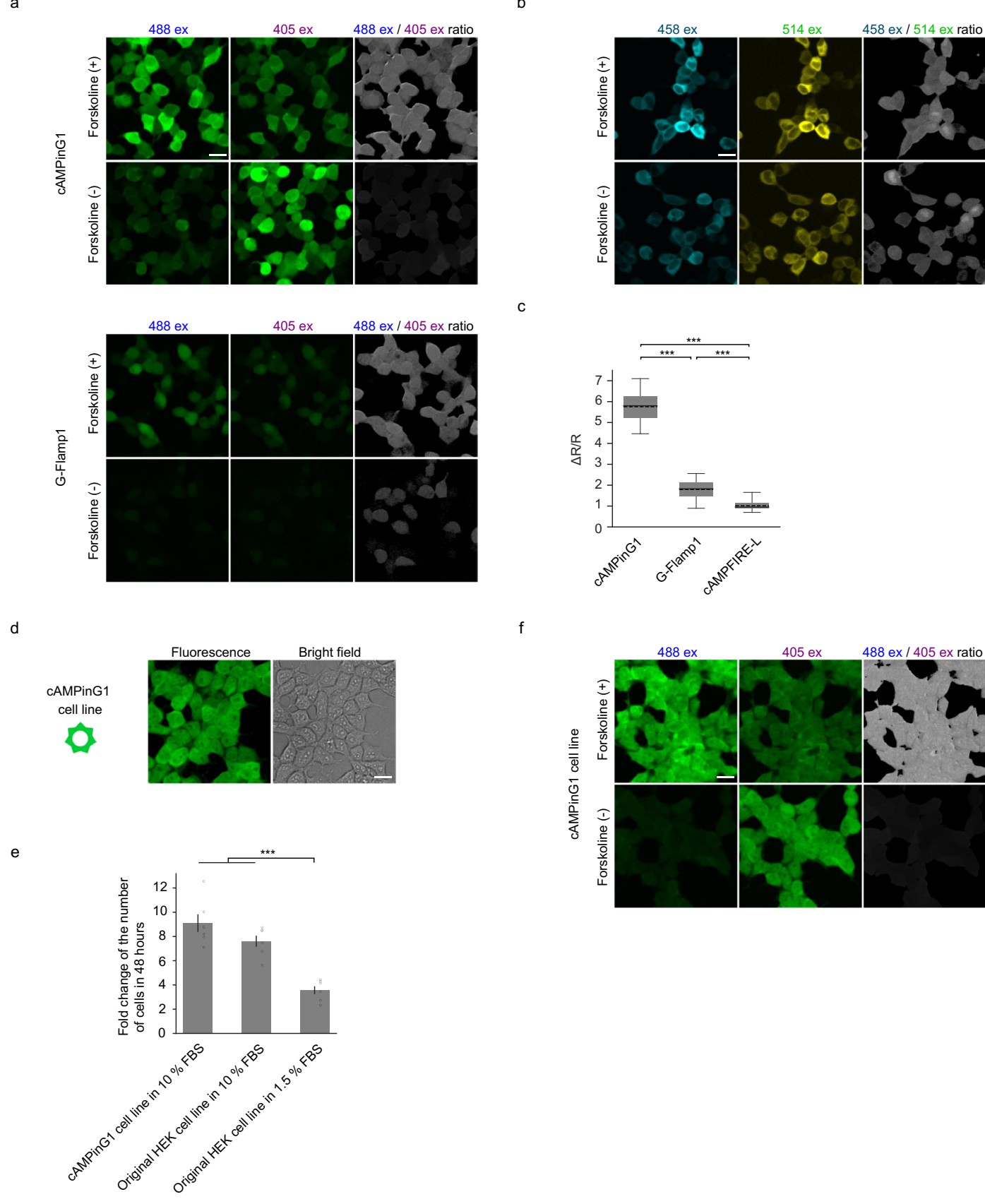

**Extended Data Fig. 9 | See next page for caption.**

**Extended Data Fig. 9 | Timelapse and single-timepoint imaging of cAMPinG1.**
**a**, Single timepoint imaging of cAMPinG1 (top) and G-Flamp1 (bottom) in the absence or presence of 10 μM forskolin. Scale bar, 20 μm. **b**, Single timepoint imaging of cAMPFIRE-L in the absence or presence of 10 μM forskolin. Scale bar, 20 μm. **c**, Relative change ($\Delta R/R$) in fluorescence ratio (R) in the absence or presence of 10 μM forskolin. For cAMPinG1 and G-Flamp1, R is the ratio of green fluorescence with 488 nm excitation to that with 405 nm excitation. For cAMPFIRE-L, R is the ratio of cyan fluorescence with 458 nm excitation to yellow fluorescence with 514 nm excitation. n = 115 (cAMPinG1 without forskolin), 155 (cAMPinG1 with forskolin), 75 (G-Flamp1 without forskolin), 102 (G-Flamp1 with forskolin), 86 (cAMPFIRE-L without forskolin), 74 (cAMPFIRE-L with forskolin) cells. Tukey's post hoc test following one-way ANOVA. cAMPinG1 vs G-Flamp1:

$P < 1.0 \times 10^{-13}$; cAMPinG1 vs cAMPFIRE-L: $P < 1.0 \times 10^{-13}$; G-Flamp1 vs cAMPFIRE-L: $P < 1.0 \times 10^{-13}$. **d**, A representative image of the cAMPinG1 stable cell line. Scale bar, 20 μm. **e**, Proliferation assay of the cAMPinG1 stable cell line and original HEK293T cell line. The original HEK cell line in 1.5 % FBS condition was for a negative control of the assay. n = 6 wells in each cell line. Tukey's post hoc test following one-way ANOVA. cAMPinG1 cell line in 10% FBS vs original cell line in 1.5% FBS: $P = 1.9 \times 10^{-5}$; Original cell line in 10% FBS vs original cell line in 1.5% FBS: $P = 5.0 \times 10^{-4}$. **f**, Single timepoint imaging of cAMPinG1 cell line in the absence (top) or presence (bottom) of 50 μM forskolin. Scale bar, 20 μm. Error bars denote SEM. Boxes indicate 25th and 75th percentiles, solid lines indicate median, dashed lines indicate mean, and whiskers indicate total range of data.

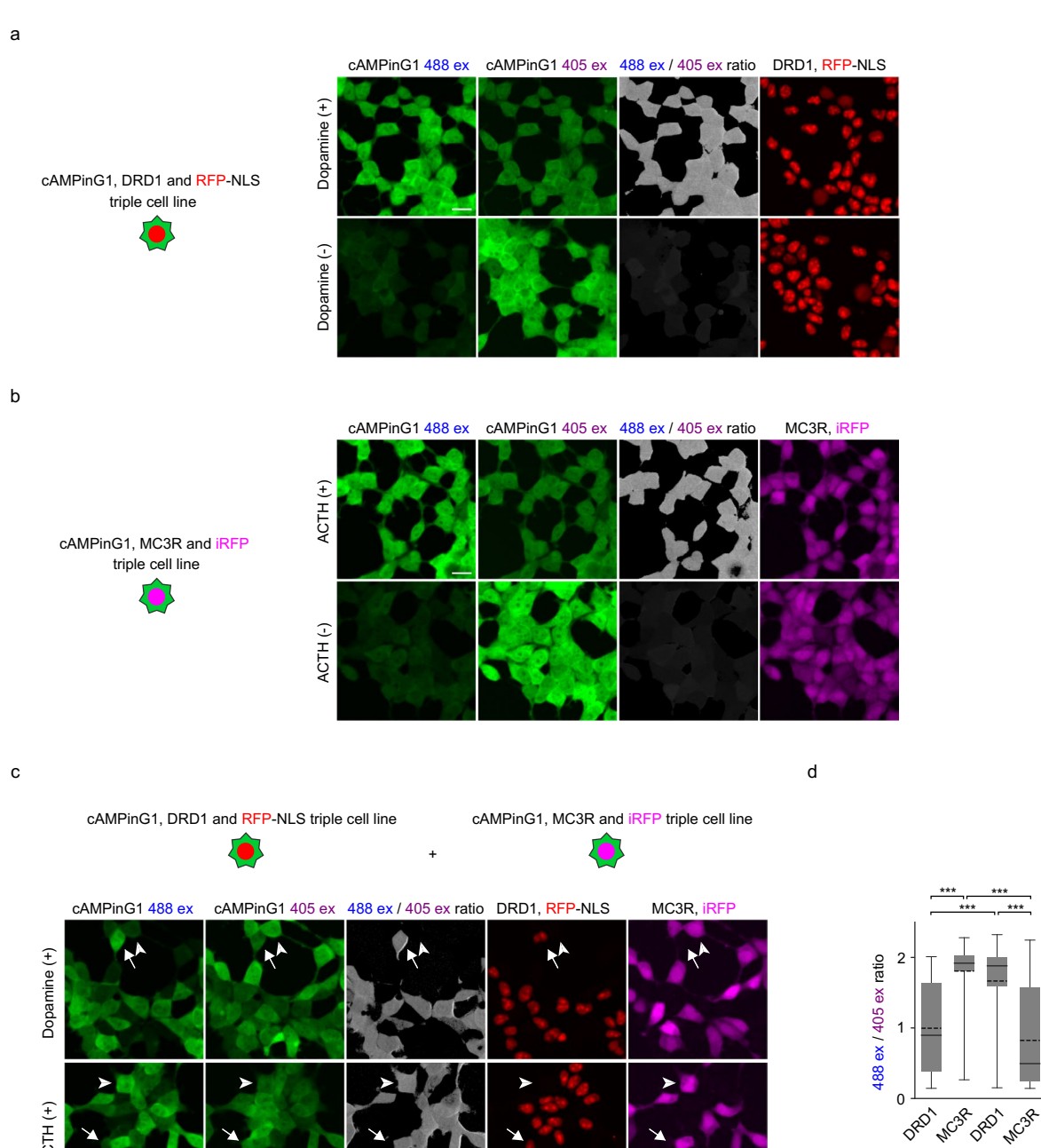

**Extended Data Fig. 10 | Triple stable cell line for drug screening. a**, Single-timepoint imaging of cAMPinG1, DRD1, and mCherry triple stable cell line in the absence (top) and presence (bottom) of 100 nM dopamine. Scale bar, 20 µm. **b**, Single-timepoint imaging of cAMPinG1, melanocortin 3 receptor (MC3R, a Gs-coupling GPCR) and iRFP670 triple stable cell line in the absence (top) or presence (bottom) of 1,000 nM ACTH. Scale bar, 20 µm. **c**, Schematic of the expression system (top). Representative images of a mixture of the cell lines with 100 nM dopamine (middle) or 1,000 nM ACTH (bottom). Representative cells expressing DRD1/RFP or MC3R/iRFP were indicated by arrows or arrowheads,

respectively. Scale bar, 20 µm. **d**, 488 ex/405 ex ratio of each cell line in the presence of dopamine or ACTH. n = 95 (ACTH + DRD1), 87 (ACTH + MC3R), 72 (dopamine + DRD1), 161 (dopamine + MC3R) cells. Boxes indicate 25th and 75th percentiles, solid lines indicate median, dashed lines indicate mean, and whiskers indicate total range of data. Tukey's post hoc test following one-way ANOVA. ACTH + DRD1 vs ACTH + MC3R: $P = 2.4 \times 10^{-11}$; ACTH + DRD1 vs dopamine + DRD1: $P = 2.1 \times 10^{-13}$. ACTH + MC3R vs dopamine + MC3R: $P = 2.1 \times 10^{-13}$; dopamine + DRD1 vs dopamine + MC3R: $P = 2.1 \times 10^{-13}$.

# Reporting Summary

## Statistics

For all statistical analyses, confirm that the following items are present in the figure legend, table legend, main text, or Methods section.

| n/a | Confirmed | |
|---|---|---|
| ☐ | ☒ | The exact sample size (*n*) for each experimental group/condition, given as a discrete number and unit of measurement |
| ☐ | ☒ | A statement on whether measurements were taken from distinct samples or whether the same sample was measured repeatedly |
| ☐ | ☒ | The statistical test(s) used AND whether they are one- or two-sided *Only common tests should be described solely by name; describe more complex techniques in the Methods section.* |
| ☒ | ☐ | A description of all covariates tested |
| ☒ | ☐ | A description of any assumptions or corrections, such as tests of normality and adjustment for multiple comparisons |
| ☐ | ☒ | A full description of the statistical parameters including central tendency (e.g. means) or other basic estimates (e.g. regression coefficient) AND variation (e.g. standard deviation) or associated estimates of uncertainty (e.g. confidence intervals) |
| ☐ | ☒ | For null hypothesis testing, the test statistic (e.g. *F*, *t*, *r*) with confidence intervals, effect sizes, degrees of freedom and *P* value noted *Give P values as exact values whenever suitable.* |
| ☒ | ☐ | For Bayesian analysis, information on the choice of priors and Markov chain Monte Carlo settings |
| ☒ | ☐ | For hierarchical and complex designs, identification of the appropriate level for tests and full reporting of outcomes |
| ☐ | ☒ | Estimates of effect sizes (e.g. Cohen's *d*, Pearson's *r*), indicating how they were calculated |

*Our web collection on statistics for biologists contains articles on many of the points above.*

## Software and code

Policy information about availability of computer code

| Data collection | Spark control software(v2.3) (TECAN), ZEN for SLM-880 (Carl-Zeiss), FV31S-SW for FVMPE-RS (Olympus), HC image (Hamatsu Photonics), pCLAMP (Molecular Devices), MCS for MOM (Sutter), Falcon for FASHIO-2PM (Nikon), LabVIEW (2021) (National Instruments), Neuroscience Studio (Doric) |
|---|---|
| Data analysis | ImageJ (Fiji 1.48), Python 3.8, Excel (Microsoft), Suite2p (https://github.com/MouseLand/suite2p), Cellpose (https://github.com/MouseLand/cellpose) |

For manuscripts utilizing custom algorithms or software that are central to the research but not yet described in published literature, software must be made available to editors and reviewers. We strongly encourage code deposition in a community repository (e.g. GitHub). See the Nature Portfolio guidelines for submitting code & software for further information.

## Data

Policy information about availability of data

All manuscripts must include a data availability statement. This statement should provide the following information, where applicable:
- Accession codes, unique identifiers, or web links for publicly available datasets
- A description of any restrictions on data availability
- For clinical datasets or third party data, please ensure that the statement adheres to our policy

cAMPinG1 and RCaMP3 plasmids and corresponding sequences were deposited to the RIKEN Bio-Resource Center (https://dna.brc.riken.jp/en/) for distribution with

the following catalog numbers: pN1-cAMPinG1 (RDB19691), pAAV-eSyn-cAMPinG1-NE-WPRE (RDB19692), pAAV-eSyn-cAMPinG1-ST-WPRE (RDB19693), pAAV-eSyn-RCaMP3-WPRE (RDB19694). The cAMPinG1 and RCaMP3 sequences are available from GenBank (accession number: LC728270 (cAMPinG1), and LC728271 (RCaMP3)).

## Human research participants

Policy information about studies involving human research participants and Sex and Gender in Research.

| Reporting on sex and gender | N/A |
| --- | --- |
| Population characteristics | N/A |
| Recruitment | N/A |
| Ethics oversight | N/A |

Note that full information on the approval of the study protocol must also be provided in the manuscript.

# Field-specific reporting

Please select the one below that is the best fit for your research. If you are not sure, read the appropriate sections before making your selection.

☒ Life sciences  ☐ Behavioural & social sciences  ☐ Ecological, evolutionary & environmental sciences

For a reference copy of the document with all sections, see nature.com/documents/nr-reporting-summary-flat.pdf

# Life sciences study design

All studies must disclose on these points even when the disclosure is negative.

| Sample size | Sample size was estimated based on prior experience and previous studies in the field. Sample sizes are reported for each experiment in the manuscript. |
| --- | --- |
| Data exclusions | For two-photon calcium and cAMP imaging, only cells recognized by Suite2p using calcium signals were included for further analysis. |
| Replication | We conducted the same experiments multiple times at least two times in different transfections (in vitro) or animals (in vivo). All experiments to replication were successful. |
| Randomization | Cells and animals were randomly allocated into experimental groups. |
| Blinding | Blinding was not performed in this study. The experimental condition were obvious to researchers and the analysis were performed objectively without human bias. |

# Reporting for specific materials, systems and methods

We require information from authors about some types of materials, experimental systems and methods used in many studies. Here, indicate whether each material, system or method listed is relevant to your study. If you are not sure if a list item applies to your research, read the appropriate section before selecting a response.

## Materials & experimental systems

| n/a | Involved in the study |
| --- | --- |
| ☒ | ☐ Antibodies |
| ☐ | ☒ Eukaryotic cell lines |
| ☒ | ☐ Palaeontology and archaeology |
| ☐ | ☒ Animals and other organisms |
| ☒ | ☐ Clinical data |
| ☒ | ☐ Dual use research of concern |

## Methods

| n/a | Involved in the study |
| --- | --- |
| ☒ | ☐ ChIP-seq |
| ☒ | ☐ Flow cytometry |
| ☒ | ☐ MRI-based neuroimaging |

# Eukaryotic cell lines

Policy information about cell lines and Sex and Gender in Research

| | |
|---|---|
| Cell line source(s) | HEK293T cells were obtained from the American Type Culture Collection (CRL-11268). |
| Authentication | The cells were authenticated based on the morphology under microscope and growth rate. |
| Mycoplasma contamination | The cell lines were not tested for mycoplasma contamination. |
| Commonly misidentified lines (See ICLAC register) | No misidentified cell lines were used in this study. |

# Animals and other research organisms

Policy information about studies involving animals; ARRIVE guidelines recommended for reporting animal research, and Sex and Gender in Research

| | |
|---|---|
| Laboratory animals | C57BL/6 and ICR wild-type animals used in this study were purchased from Japan SLC and experiments were performed using both male and female sex between 0–20 weeks of age. A2A-Cre mice ((B6.FVB (Cg)-Tg (Adora2a-cre) KG139Gsat/Mmucd)) were obtained from MMRRC and experiments were performed using male sex between 8 – 10 weeks of age.   Mice were group-housed and kept on a 12-h light/dark cycle with ad libitum food and water at room temperature. The housing condition for were controlled at room temperature of approximately 22–24°C and a relative humidity of 40–60%. |
| Wild animals | No wild animals were used in this study. |
| Reporting on sex | Male and female animals were used. |
| Field-collected samples | No filed-collected samples were used in this study. |
| Ethics oversight | All experimental procedures were performed using protocols that were approved by the Institutional Animal Care and Use Committee at Kyoto University (Lif-K23008), University of Yamanashi (A1-20), RIKEN (W2022-2-012), and The University of Tokyo (P18-118). |

Note that full information on the approval of the study protocol must also be provided in the manuscript.

