## [Peer Review File · Nature Methods]

Peer Review Information

Manuscript Title: A multicolor suite for deciphering population coding in calcium and cAMP in vivo

Corresponding author name(s): Tatsushi Yokoyama, Masayuki Sakamoto

Editorial Notes: None

Reviewer Comments & Decisions:

Decision Letter, initial version:

Dear Dr Sakamoto,

Thank you for your patience. Your Article, "A multicolor suite for deciphering population coding in calcium and cAMP *in vivo*", has now been seen by three reviewers. As you will see from their comments below, although the reviewers find your work of considerable potential interest, they have raised a number of concerns. We are interested in the possibility of publishing your paper in Nature Methods, but would like to consider your response to these concerns before we reach a final decision on publication.

We therefore invite you to revise your manuscript to address these concerns. Importantly, please do compare your cAMPinG and RCaMP3 sensors to the current best-in-class sensors in *in vivo* experiments. Please also address the other concerns voiced by the reviewers.

- * include a point-by-point response to the reviewers and to any editorial suggestions
- * please underline/highlight any additions to the text or areas with other significant changes to facilitate review of the revised manuscript
- * address the points listed described below to conform to our open science requirements
- * ensure it complies with our general format requirements as set out in our guide to authors at

www.nature.com/naturemethods

* resubmit all the necessary files electronically by using the link below to access your home page

[Redacted] This URL links to your confidential home page and associated information about manuscripts you may have submitted, or that you are reviewing for us. If you wish to forward this email to co-authors, please delete the link to your homepage.

We hope to receive your revised paper within 3-4 months. If you cannot send it within this time, please let us know. In this event, we will still be happy to reconsider your paper at a later date so long as nothing similar has been accepted for publication at Nature Methods or published elsewhere.

OPEN SCIENCE REQUIREMENTS

REPORTING SUMMARY AND EDITORIAL POLICY CHECKLISTS

DATA AVAILABILITY

We strongly encourage you to deposit all new data associated with the paper in a persistent repository where they can be freely and enduringly accessed. We recommend submitting the data to discipline-specific and community-recognized repositories; a list of repositories is provided here:

<http://www.nature.com/sdata/policies/repositories>

All novel DNA and RNA sequencing data, protein sequences, genetic polymorphisms, linked genotype and phenotype data, gene expression data, macromolecular structures, and proteomics data must be deposited in a publicly accessible database, and accession codes and associated hyperlinks must be provided in the "Data Availability" section.

MATERIALS AVAILABILITY

ORCID

Nature Methods is committed to improving transparency in authorship. As part of our efforts in this direction, we are now requesting that all authors identified as 'corresponding author' on published papers create and link their Open Researcher and Contributor Identifier (ORCID) with their account on the Manuscript Tracking System (MTS), prior to acceptance. This applies to primary research papers

only. ORCID helps the scientific community achieve unambiguous attribution of all scholarly contributions. You can create and link your ORCID from the home page of the MTS by clicking on 'Modify my Springer Nature account'. For more information please visit www.springernature.com/orcid.

Best regards,
Nina

Nina Vogt, PhD
Senior Editor
Nature Methods

Reviewers' Comments:

Reviewer #1:

Remarks to the Author:

In this study, the authors developed a high-affinity, large fluorescence intensity change cAMP sensor and an improved red calcium sensor. They characterized and demonstrated the functionality of these new sensors in in vitro and in vivo applications, making some interesting biological observations in the process. They demonstrate the broad applicability of these tools for studying the brain and cell lines, and it is clear these tools could be applied to study other organs. Barring one potential concern (major comment #1) and one possible overinterpretation of the data (major comment #2), this study is a fantastic achievement. Overall, this is an exciting paper that could have an immediate impact on both the neuroscience field as well as a broader biology community especially if DNA and viral reagents are made easily accessible.

Major comments:

1) A possible concern for this sensor is the amount of pH sensitivity, which was not reported and can typically happen with excitation ratiometric sensors that switch between A-band and B-band excitation (see Yellen & Mongeon, Current Opinion in Chemical Biology, 2015). It's important to address this particularly as it is a Methods paper and will help future research users of this sensor that may not be aware of such issues. Can the authors provide a pH sensitivity curve similar to Figure 1f at the very least for cAMPinG1, though a comparison across all 4 sensors would be extremely valuable to the community?

a. Including the cAMPinG1 mutant in the pH sensitivity assay is important as it would address how much one can use the mutant data as confirmation that pH was not underlying any observed in vivo cAMP dynamics.

2) Using ChRmine to directly stimulate neurons, the authors conclude that cell-intrinsic spiking can lead to calcium influx and cAMP production. However, as synaptic transmission is not blocked in these

in vivo studies and given the long delay between stimulation and observed cAMP increase, it's possible that the increase is mediated by G protein-coupled receptor activation. For example, the stimulated neuron may drive local interneurons or glial cells via glutamate release and these stimulated cells then release a neuromodulator back onto the stimulated cell.

a. Unless I missed something, there isn't sufficient evidence to conclude that calcium direction drives cAMP production in a cell-autonomous way. I suggest rewording the abstract, introduction, discussion and other places where the authors say "Ca²⁺-induced" or conclude that it must be calcium driving cAMP and instead say correlated with calcium. In the discussion it would be fair to speculate on this mechanism, but also speculate on polysynaptic evoked neuromodulator release.

3) It is not clear to me what the questions in Lines 50 – 52 are exactly. What does "cAMP population pattern" exactly mean? I think this needs rewording especially as this is a key sentence of the introduction that addresses outstanding questions in the field that these new tools could answer.

4) While other cAMP biosensors were recently published, the major accomplishment of cAMPinG1 is achieving the higher affinity for cAMP while also maintaining a large fluorescence change. However, the authors did not show how the higher affinity was necessary or important for biological studies, which would be necessary to demonstrate this tool's advantage over existing methods like g-Flamp. It might be that the sensor is partially saturated and limited in its dynamic range by ceiling effects. For example, is the fact that the response magnitude is so much smaller in slices (Extended Figure 3) reflective of a ceiling effect of the sensor?

Addressing this point directly would really clarify to the research community cases when it would be necessary to have this higher affinity.

a. Could the authors add discussion in the manuscript about what is known about the actual concentrations of cAMP in different neuronal (spines, dendrites, cilia, soma, nucleus) and glial compartments. This will address how well tuned the dynamic range of cAMPingG1 for certain applications.

b. Do the authors possibly have any direct comparison of g-Flamp and cAMPingG1 in similar in vivo studies as in Figure 2, 4, or 5? It would really highlight the advantage of using cAMPingG1. For example, it might even be that the suppression of cAMP seen in Figure 5 e,f is only possible with the higher affinity of cAMPinG1.

Minor comments:

1) It is becoming more apparent in the field the importance of quantitative biosensor measurements in vivo. Have the authors measured whether there is a change in lifetime of cAMPingG1?

2) Line 40 is overlooking several studies that have recorded cAMP dynamics in vivo. I agree that there is still much to discover, but there are several studies that have revealed in vivo cAMP dynamics including:

a. Zhang et al., Nature, 2021

b. Massengill et al., Nature Methods, 2022

3) Line 99 = typo, AMPingG1-ST is missing the "c"

4) Line 146 – it would help to more clearly explain why blue shifting the rcamp was beneficial

- 5) Line 182 – please clarify what you mean by “some parts of cells”. This was confusing to me.
- 6) Supplemental Table 1 – for cAMPfire-H reference it should be Massengill et al., not Crystian et al.
- 7) For Figure 2, please put individual dots for data points on top of the bar graphs.
- 8) For Figure 4f, were non-motion responsive cells confirmed to be responsive at other points in the recording. Are the authors confident these neurons express sufficient rcamp3 and are truly non-responsive versus just not expressing the red calcium sensor well.
- 9) Can the authors please add what the inter-stimulus interval was in Figure 5a.

Reviewer #2:

Remarks to the Author:

Yokoyama et al develops a green single-fluorophore sensor for cAMP, called cAMPinG1, and a red single-fluorophore sensor, called RCaMP3. cAMPinG1 uses a de novo design, whereas RCaMP3 is developed by integrating a number of known mutations from two R-GECO variants into jRGECO1a. The authors show that these sensors can be used in vivo in a multiplex manner for single-cell imaging and fiber photometry in the context of animal behavior. Notably, the authors discover direction selective cAMPinG1 signals in neurons of the mouse visual cortex that correlate with the direction selectivity of the corresponding neurons' calcium activity.

New cAMP and calcium sensors with enhanced properties are of great interests to the field. However, the enthusiasm is dampened for a number of reasons. The improvement over existing sensors seems to be moderate and is not well demonstrated in cells and neurons in direct comparison to existing sensors. Neither sensor is sufficiently characterized. Although the biological observation regarding the direction selectivity of cAMPinG1 signals is interesting, it needs critical controls and is not clear whether it is achievable (or not) using existing sensors. The authors should better acknowledge the recent progresses of in vivo imaging of existing cAMP and red calcium sensors, and demonstrate that the current sensor can achieve what previous sensors cannot do in a biological context.

Major comments.

1. Since the major goal of cAMPinG1 is for in vivo imaging, excitation radiometry is not really applicable. 2p lasers change wavelengths slowly, and 2p excitation equivalent to UV single photon excitation is associated with high background and photodamage. In fact, ratiometric excitation is not used in the demonstrated in vivo imaging experiments. With single wavelength excitation, the improvement in dynamic range is minimal.
2. The excitation of existing sensors should be at or near their respective optimal wavelength. E.g., G-Flamp1 is thought to exhibit the maximal dF/F0 at 450 nm.
3. Another improvement of cAMPinG1 is the sensitivity. However, whether the enhanced sensitivity is beneficial needs to be better demonstrated. The referenced EC50s of PKA was determined in vitro. The value in vivo can be different. Where determination of the basal cAMP concentration inside a cell has been attempted, the values are typically ~0.5–1 μM . A sensor that is too sensitive would result in elevated baseline in vivo. Indeed, the dynamic range of cAMPinG1 inside an intact cell appears to be much smaller than in lysate (Fig. 1j and ED Fig 3). Therefore, the sensor needs to be compared side-by-side with existing sensors under the same condition in intact cells and in intact neurons in neuronal

slices using physiologically relevant stimulant.

4. Another potential drawback of too high a sensitivity is the potential side effect of buffering. The authors need to determine whether or not cAMP-sensitive neuronal properties (synaptic transmission, excitability, etc) are altered by the expression of the sensor.
5. cAMPinG1 needs to be better characterized, such as its kinetics and pH sensitivity. Notably, since buffering can alter cAMP concentration and kinetics inside a cell, kinetic experiments should include measurements in intact cells, both to a puff and to a step increase of a physiological stimulant.
6. In vivo cAMP imaging has been achieved (Oe et al, 2020; Massengill et al, 2022; Wang et al, 2022). This should be clearly described in the introduction. Additionally, it is the authors' burden to demonstrate that cAMPinG1 performs better than previous sensors to a degree that crosses the high bar of Nature Methods. At this time, there is essentially no comparison in vivo.
7. The authors utilize both cAMPinG1-NE and cAMPinG1-ST in vivo and in vitro. How are these variants compared to the parental sensor? This is important given that cAMPinG1 achieves only 5% of the dynamic range (ED Fig. 3) under very strong conditions compared to Fig. 1e. Also, are there example images showing the subcellular distribution of the ST-variant?
8. The authors state that they added mutations to jRGECO1a to blue-shift the excitation spectra, generate a larger dynamic range and less aggregation. The aggregation data should be shown and quantified. Overall, the improvement as shown in Fig. 3a-e, and ED Fig. 4 seems to be moderate for the bar of Nature Methods.
9. As the field comes to realize, RS20 instead of M13 is used in RGECOs and GCaMPs. Also, since the improvement on calcium sensor is on RGECO, why naming the new sensor RCaMP3? RCaMPs already exist, and they are based on different calmodulin binding peptide and different fluorophore. Naming this new variant to RCaMP3 will cause a lot of confusion to the users. Please consider a name along the line of RGECO.
10. Since many mutations are introduced to the fluorescent protein in the new calcium sensor, whether the photostability and pH sensitivity have changed should be tested.
11. How the new calcium sensor responds to different number of action potentials should be presented and compared to the existing sensors.
12. Fig. 3d-h: Any explanation why jRGECO1a's response is only about 10% of the literature? Also, Although the new calcium sensor gives larger response, its noise also seems to be higher (although to a less degree than the signal). Why?
13. Under 2p excitation for the cAMP sensor, how much contamination of the signal from the calcium sensor are there in the cAMP signal channel? One limitation of nearly all red fluorescent protein is that they contain a green component due to bifurcating maturation. This needs to be quantified for multiplex imaging.
14. For in vivo experiments, both the cAMP and calcium sensor response should be compared with existing sensors to judgment whether the new sensors present a true improvement and what is the degree of the improvement. Relevant figures include Fig. 2, Fig. 3j, Fig. 4, and Fig. 5.
15. Pharmacological manipulation should be included for in vivo experiments to demonstrate the cAMP response and to dissect the potential pathways controlling its activity.
16. cAMP binding mutant should be used in Fig. 4 and 5 to verify that the sensor response is specific.
17. Fig. 6: Couldn't the experiment be done with existing ratiometric cAMP sensors?

Additional comments

1. For measurements of cAMP affinity, the methods on page 21 (lines 361-366) suggest that the supernatant containing the sensors and cAMP was diluted to reduce the cAMP concentration. This would also reduce the sensor concentration, making comparison between cAMP concentrations inaccurate. Please either redo the experiment or rewrite the text so that it is clear that the sensor

concentration was not held constant.

2. Fig. 3j, 4 and 5, example images before and after stimulation with sufficient magnification should be presented.
3. ED Fig. 2e is not referenced in text. Also, a quantification would strengthen the experiment.
4. Fig. 5e: Within the same cell, is there correlation between calcium amplitude and cAMPinG1 signal?
5. Fig. 5j: For ChRmine elicited cAMP response, was it elicited by a single 4-s spiral stimulation? Please clarify in text. Also, how many action potentials does this corresponding to?
6. "G-Flamp1 limited the number of cells that could be imaged and quantified cAMP dynamics in the mouse cortex". This need to be justified for cited.
7. The reviewer disagrees with this assertion: "which made fast imaging or combined use with other sensors, such as Ca²⁺ indicators, generally challenging due to the required optical settings".

Minor comments

1. Page 7 line 119: The corresponding figure (Extended Data Fig. 3) shows application of forskolin + IBMX, but text reads only forskolin.
2. Page 9 line 139: "one of the upstream of cAMP signaling" should read "one of the upstream modulators of cAMP signaling". Similarly, page 11, Line 187 should read "multiple upstream pathways, including neuromodulators". Also page 12, line 210.
3. Page 14 last paragraph: Fig. 6a,b shows dopamine application but this is not discussed in the text.
4. Page 15 first paragraph. Please define TFX and ADRB2 in the text.
5. Fig. 1e and 3b: please specify the used cAMP or calcium concentration in text or legend.
6. Page 31 line 550: Is 18 hours correct here? That seems very soon after surgery. It seems unlikely that inflammation would be reduced enough to accurately see cells.
7. ED Fig. 2d: please include cAMPinG1 cAMP concentration response curve on this plot as a comparison of cAMP vs cGMP sensitivity
8. Supplementary Table 1 should read Massengill et al., instead of Crystian et al.
9. cAMPinG1 exhibits a cAMP binding hill coefficient of 1. Any explanations given that there are two binding sites?

Reviewer #3:

Remarks to the Author:

Main findings:

The authors developed an ultrasensitive genetically encoded green cAMP indicator, cAMPinG1, with a larger dynamic range and higher cAMP affinity than existing green cAMP indicators, allowing for in vivo imaging of cAMP transients in the somata and dendritic spines of neurons in the mouse visual cortex on the order of tens of seconds. They also introduced an improved red calcium indicator, RCaMP3, enabling simultaneous measurement of population patterns in Ca²⁺ and cAMP in hundreds of neurons. Dual-color imaging revealed that cell-specific cAMP transients represented specific information downstream of Ca²⁺ signaling as well as GPCR-induced cAMP responses. The authors demonstrated the application of cAMPinG1 imaging in cultured cells for GPCR biology and suggest that their multicolor suite for Ca²⁺ and cAMP imaging will allow examination of how the information encoded in action potentials, Ca²⁺, and GPCR signaling is integrated and stored as cAMP transients at the single-cell level in both in vitro and in vivo settings.

Overall assessment:

This work is overall technically sound and introduced two novel genetically encoded indicators. This

work demonstrated the application of these two indicators via sophisticated and well-designed in vivo experiments. Overall, this work is fit for publication. The manuscript is ready for publication after responding to the following revisions. Below are some specific comments:

Major comments:

1. The authors included a comprehensive in vitro characterization of both indicators. However, they didn't include the kinetic characterization of cAMPinG1 and its benchmarking comparison with other existing cAMP indicators.
2. The authors grafted several known amino acid substitutions from R-GECO1.2 onto jRGECO1a. Among those mutations, M128R/I131V/V139L/F188L/N231S/S234T blue shift the excitation spectra. I'm a little bit confused about the rationale of making the RCaMP3 action spectra more blue-shifted. How would this make RCaMP3 a better red indicator in concurrent green cAMP imaging?
3. Based on the results, the half-decay time of cAMPinG1 is more than 100 times larger than that of RCaMP3. Is this number comparable to the physiological facts? If not, how would it impact its temporal resolution in applications?

Minor comments:

1. The authors stated that Two-photon dual-color imaging for Ca²⁺ and cAMP revealed a strong correlation between cell-specific cAMP increases and Ca²⁺ transients. However, there is no statistical analysis supporting this statement. The authors should consider including a temporal pairwise correlation between the cAMP trace and Ca trace within the same neuron.

Author Rebuttal to Initial comments

**Response to the reviewers**

We would like to thank the editor and reviewers for their careful reading and constructive
comments on our manuscript. All comments are valuable and helpful for revising and
improving our manuscript. To address the comments adequately, we have carried out new
experiments to compare our new sensors with the existing sensors *in vivo*, and to further
characterize these sensors. We describe in detail the changes in the following point-by-
point response. For clarity, the received comments are represented in *italic Arial* fonts,
while our responses are shown in Times fonts. All changes in the revised manuscript are
highlighted in yellow.

**Reviewer #1:**

First, we thank the reviewer for positive and insightful comments and constructive
suggestions.

**Major comments:**

*1) A possible concern for this sensor is the amount of pH sensitivity, which was not reported*
*and can typically happen with excitation ratiometric sensors that switch between A-band and*
*B-band excitation (see Yellen & Mongeon, Current Opinion in Chemical Biology, 2015). It's*
*important to address this particularly as it is a Methods paper and will help future research*
*users of this sensor that may not be aware of such issues. Can the authors provide a pH*
*sensitivity curve similar to Figure 1f at the very least for cAMPinG1, though a comparison*
*across all 4 sensors would be extremely valuable to the community?*

*a. Including the cAMPingG1 mutant in the pH sensitivity assay is important as it would*
*address how much one can use the mutant data as confirmation that pH was not underlying*
*any observed in vivo cAMP dynamics.*

We thank the reviewer's important comment and suggestion about pH sensitivity of the
sensor. In response to this, we measured the pH sensitivity of four cAMP sensors,
including Flamindo2, gCarvi, G-Flamp1, and cAMPinG1, and also the cAMPinG1
mutant. Notably, G-Flamp1, cAMPinG1, cAMPinG1mut had pK_a closely aligned with
the physiological pH of 7.0-7.4. In contrast, pH-dependent $\Delta F/F$ against cAMP-
dependent $\Delta F/F$ of cAMPinG1 was smallest among all the sensors evaluated. These
findings emphasize the significance of pH-dependent fluorescence change when
interpreting *in vivo* data and suggest potential avenues for further improvement of these

sensors. These quantitative results were shown as Extended Data Fig. 2f, Supplementary
Table 2, and described in line 111 in the revised manuscript.

*2) Using ChRmine to directly stimulate neurons, the authors conclude that cell-intrinsic*
*spiking can lead to calcium influx and cAMP production. However, as synaptic transmission*
*is not blocked in these in vivo studies and given the long delay between stimulation and*
*observed cAMP increase, it's possible that the increase is mediated by G protein-coupled*
*receptor activation. For example, the stimulated neuron may drive local interneurons or glial*
*cells via glutamate release and these stimulated cells then release a neuromodulator back*
*onto the stimulated cell.*

*a. Unless I missed something, there isn't sufficient evidence to conclude that calcium*
*direction drives cAMP production in a cell-autonomous way. I suggest rewording the abstract,*
*introduction, discussion and other places where the authors say "Ca²⁺-induced" or conclude*
*that it must be calcium driving cAMP and instead say correlated with calcium. In the*
*discussion it would be fair to speculate on this mechanism, but also speculate on polysynaptic*
*evoked neuromodulator release.*

We appreciate the reviewer for pointing out other possibilities to explain ChRmine-
induced cAMP increase than the cell-autonomous, Ca²⁺-dependent pathway. As suggested,
we have revised the sentences relating "Ca²⁺-induced" to "Ca²⁺-related" or have
rephrased in the revised manuscript (line 27, 57-60, 236, 314, and 321). In addition, in
the discussion section, we have changed the original sentence to: "The spike-induced
cAMP increase can be mediated by voltage-dependent Ca²⁺ channels and Ca²⁺-dependent
ACs, while other non-cell autonomous mechanisms involving local interneurons and glia
cells can also be possible." This change can be found in line 311-314 in the revised
manuscript.

*3) It is not clear to me what the questions in Lines 50 – 52 are exactly. What does "cAMP*
*population pattern" exactly mean? I think this needs rewording especially as this is a key*
*sentence of the introduction that addresses outstanding questions in the field that these new*
*tools could answer.*

We regret that this part was not articulated in the original manuscript. Following the
suggestion, we revised the original sentence to: "(1) What information is encoded in the
cAMP population pattern represented by the combination of cAMP dynamics in

individual cells of a large population?" This change can be found in line 49-51 in the
revised manuscript.

*4) While other cAMP biosensors were recently published, the major accomplishment of*
*cAMPinG1 is achieving the higher affinity for cAMP while also maintaining a large*
*fluorescence change. However, the authors did not show how the higher affinity was*
*necessary or important for biological studies, which would be necessary to demonstrate this*
*tools advantage over existing methods like g-Flamp. It might be that the sensor is partially*
*saturated and limited in its dynamics range by ceiling effects. For example, is the fact that*
*the response magnitude is so much smaller in slices (Extended Figure 3) reflective of a*
*ceiling effect of the sensor?*

*Addressing this point directly would really clarify to the research community cases when it*
*would be necessary to have this higher affinity.*

We thank the reviewer's insightful comments and constructive suggestion. In response to
the reviewer's comment, we performed side-by-side comparison of cAMPinG1-NE and
G-Flamp1 in acute brain slices. When exposed to local application of 1 μ M dopamine in
the striatum, the $\Delta F/F$ of cAMPinG1-NE was ~ 0.15 ($\sim 15\%$), which is significantly
superior to that of G-Flamp1 (~ 0.02). Similarly, the $\Delta F/F$ of cAMPinG1-NE in response
to noradrenaline bath application in the cortex was ~ 0.5 , which is superior to that of G-
Flamp1 (~ 0.1). These results strongly suggest the cAMP affinity of cAMPinG1-NE was
more suited to detect cAMP transients than that of G-Flamp1 in these conditions since *in*
*vitro* cAMP affinity of cAMPinG1-NE was much larger than G-Flamp1. These
comparative evaluation results were shown as Extended Data Fig. 3e-1 and described in
line 123-126 in the revised manuscript.

*a. Could the authors add discussion in the manuscript about what is known about the actual*
*concentrations of cAMP in different neuronal (spines, dendrites, cilia, soma, nucleus) and*
*glial compartments. This will address how well tuned the dynamic range of cAMPinG1 for*
*certain applications.*

We appreciate the reviewer's constructive suggestion. We understand the significance of
the comprehensive discussion regarding the concentrations of cAMP in different cellular
compartments in both neuronal and glial cells. In response to this, we have expanded our
discussion to include the current knowledge about cAMP concentrations. Generally, the

concentration of cAMP in cells is ~100 nM. Given this, this high affinity of cAMPinG1
($K_d = 180$ nM) should be ideally well-suited to capture these changes. The detail was
described in the discussion section in line 287-291, which includes important references
that address cAMP concentrations in the revised manuscript.

To the best of our knowledge, the actual cAMP concentration in each
compartment remains unknown due to the lack of highly sensitive cAMP indicators
before our study. We believe that further studies utilizing cAMPinG1 could elucidate
these concentrations. We have expanded on our perspective in the discussion section in
the revised manuscript (line 295-298).

*b. Do the authors possibly have any direct comparison of g-Flamp and cAMPinG1 in similar*
*in vivo studies as in Figure 2, 4, or 5? It would really highlight the advantage of using*
*cAMPinG1. For example, it might even be that the suppression of cAMP seen in Figure 5*
*e,f is only possible with the higher affinity of cAMPinG1.*

We appreciate the reviewer's comment. To address it, We conducted a comparative
analysis of cAMPinG1 and G-Flamp1 *in vivo* by utilizing two-photon microscopy and
fiber photometry. We observed that cAMPinG1 provided a clearer depiction of not only
the elevation of cAMP induced by physical stimuli but also subsequent suppression post-
stimulation than G-Flamp1. These findings suggest that cAMPinG1 is advantageous for
detecting cAMP change *in vivo*, possibly due to its higher affinity for cAMP. These new
results were shown as Fig. 2d,e, Extended Data Fig. 7f-i, Extended Data Fig. 11a,b, and
Extended Data Fig. 12 and described in line 137-138, 201-203, 227-229, and 249-251 in
the revised manuscript.

**Minor comments:**

*1) It is becoming more apparent in the field the importance of quantitative biosensor*
*measurements in vivo. Have the authors measured whether there is a change in lifetime of*
*cAMPinG1?*

We thank the reviewer for the insightful suggestion. We totally agree with the reviewer's
comment that it is becoming more apparent in the field the importance of quantitative
biosensor measurements *in vivo*. In response, we have measured the lifetime of
cAMPinG1, which revealed an increase in the lifetime in response to cAMP increase in
HEK293T cells. These results were added as Extended Data Fig. 2k and described in line
119-120 in the revised manuscript.

*2) Line 40 is overlooking several studies that have recorded cAMP dynamics in vivo. I agree*
*that there is still much to discover, but there are several studies that have revealed in vivo*
*cAMP dynamics including:*

*a. Zhang et al., Nature, 2021*

*b. Massengill et al., Nature Methods, 2022*

We appreciate the reviewer's helpful comment and suggestion. As suggested, we have
incorporated important references (Harada et al., *Scientific Reports* 2017; Oe et al., *Nature*
*Communications* 2020; Zhang et al., *Nature* 2021; Wang et al., *Nature Communications*
2022; Massengill et al., *Nature Methods* 2022) regarding *in vivo* studies. These have been
cited in line 43 as reference 4 through 8 in the revised manuscript.

*3) Line 99 = typo, AMPingG1-ST is missing the "c"*

We apologize for our mistake. We have corrected the name in the revised manuscript (line
98).

*4) Line 146 – it would help to more clearly explain why blue shifting the rcamp was beneficial*

We appreciate the reviewer's suggestion. The majority of commercially available dual-
wavelength lasers, such as Coherent's Chameleon Discovery and Spectra-Physics'
DeepSee Insight, are typically designed to operate at a fixed wavelength of approximately
1,040 nm. In contrast, the two-photon excitation peak of existing red calcium indicator
such as jRGECO1a is tuned at around 1,060-1,070 nm (Dana et al., *Elife* 2016, Figure
2—figure supplement 1), which shows a mismatch in laser wavelength. Therefore, blue-
shifted excitation spectrum of RCaMP3 is more compatible for two-photon imaging with
such lasers. We added this explanation in the revised manuscript (line 162-165).

*5) Line 182 – please clarify what you mean by "some parts of cells". This was confusing to*
*me.*

We apologize for our confusing description. We have changed "some parts of cells" to
"around 25% of cells" in the revised manuscript (line 203).

*6) Supplemental Table 1 – for cAMPfire-H reference it should be Massengill et al., not*
*Crystian et al.*

We apologize for our mistake. We have corrected the reference in Supplementary Table
2.

*7) For Figure 2, please put individual dots for data points on top of the bar graphs.*

We thank the reviewer's careful observation. In response to this comment, we put
individual data points in Fig. 2e,f.

*8) For Figure 4f, were non-motion responsive cells confirmed to be responsive at other points*
*in the recording. Are the authors confident these neurons express sufficient rcamp3 and are*
*truly non-responsive versus just not expressing the red calcium sensor well.*

We appreciate the reviewer's comment. To identify Ca²⁺ responsive neurons, we utilized
the suite2p, an image processing pipeline (Pachitariu et al., *bioRxiv* 2019). This algorithm
identifies ROIs based on neuronal activity in RCaMP3 fluorescence signals. The ROIs in
Fig. 4 match the same neurons detected in Fig. 5, which are based on visual stimuli
response or spontaneous activity, although the order of the figures is reversed. The details
of this procedure also have been described in the method section (line 829-834).
Therefore, the identified non-motion responsive cells in Fig. 4 are, indeed, are neurons
that demonstrated a response to visual stimuli or exhibited spontaneous activity during
the visual stimulation phase in Fig. 5. This provides assurance that neurons expressing
RCaMP3 adequately labeled. Therefore, the lack of response in the context of Fig. 4 is
not due to insufficient expression of the RCaMP3 but rather a true representation of their
activity under the forced running condition.

*9) Can the authors please add what the inter-stimulus interval was in Figure 5a.*

We appreciate the reviewer's suggestion. As suggested, we added the inter-stimulus
interval in Fig. 5a.

Again, we appreciate the reviewer's many helpful and constructive comments.

**Reviewer #2:**

First of all, we thank the reviewer for careful reading and providing many constructive
comments and suggestions.

*New cAMP and calcium sensors with enhanced properties are of great interests to the field.*
*However, the enthusiasm is dampened for a number of reasons. The improvement over*
*existing sensors seems to be moderate and is not well demonstrated in cells and neurons in*
*direct comparison to existing sensors. Neither sensor is sufficiently characterized. Although*
*the biological observation regarding the direction selectivity of cAMPinG1 signals is*
*interesting, it needs critical controls and is not clear whether it is achievable (or not) using*
*existing sensors. The authors should better acknowledge the recent progresses of in vivo*
*imaging of existing cAMP and red calcium sensors, and demonstrate that the current sensor*
*can achieve what previous sensors cannot do in a biological context.*

We appreciate the reviewers' comments and interest about the direction selectivity of
cAMP signals. In response to the concerns, we have conducted additional cAMP imaging
using G-Flamp1 with repetitive moving grating. Contrary to cAMPinG1 result, we did
not observe a notable increase in G-Flamp1 fluorescence, which might be attributed to its
lack of cAMP sensitivity. This result demonstrates that cAMPinG1 significantly
outperforms existing sensors for *in vivo* imaging. This result was added as Extended Data
Fig. 11a.

**Major comments.**

*1. Since the major goal of cAMPinG1 is for in vivo imaging, excitation radiometry is not really*
*applicable. 2p lasers change wavelengths slowly, and 2p excitation equivalent to UV single*
*photon excitation is associated with high background and photodamage. In fact, ratiometric*
*excitation is not used in the demonstrated in vivo imaging experiments. With single*
*wavelength excitation, the improvement in dynamic range is minimal.*

We agree with the reviewer's comment that ratiometric excitation is not applicable to *in*
*in vivo* two-photon imaging. However, we would like to respectfully point out that
ratiometric imaging using UV light excitation (405 nm) and blue light excitation (470
247 nm) are prevalently used in *in vivo* calcium imaging with GCaMP using fiber photometry.
This approach is critical for eliminating motion artifacts (Karigo et al., *Nature* 2021).
Therefore, it would perhaps be an oversimplification to state that ratiometric excitation is
not used in *in vivo* imaging experiments. In our study, we used ratiometric excitation to

detect cAMP signals by fiber photometry, and the methodology has already been
 described in the methods section in the manuscript.

Moreover, we have compared the change in cAMP level in the dorsal striatum
 induced by forced running using G-Flamp1 and fiber photometry. We found that
 cAMPinG1 was able to detect the cAMP changes more clearly than G-Flamp1. These
 results have shown as Extended Data Fig. 12. These results demonstrate the superiority
 of cAMPinG1 in detecting cAMP changes. Therefore, we firmly believe that the
 pronounced ratio change in cAMPinG1 is a significant advantage for *in vivo* imaging.

*2. The excitation of existing sensors should be at or near their respective optimal wavelength.*
 *E.g., G-Flamp1 is thought to exhibit the maximal dF/F_0 at 450 nm.*

We thank the reviewer's important comment about optimal excitation wavelength. As the
 reviewer pointed out, we performed a side-by-side comparison *in vitro* with the use of
 450 nm excitation laser wavelengths. We revealed that $\Delta F/F$ of G-Flamp1 at 450 nm was
 larger than that of cAMPinG1 *in vitro* (see the figure below).

We respect the reviewer's point regarding the excitation wavelength for G-
 Flamp1. However, we would like to respectfully point out, without any bias, that 450 nm
 excitation light is less commonly used for *in vivo* imaging of green sensors compared to
 other wavelengths, such as 473 and 488 nm. One possible reason for this is that the
 excitation efficiency at 450 nm is approximately half at the peak wavelength, ensuring
 insufficient brightness. Indeed, even in the original G-Flamp1 study, 450 nm was not used
 as the excitation for *in vivo* imaging utilizing fiber photometry (Wang et al., *Nature*
 *Communications* 2022). Our study is primarily focused on *in vivo* applications, and thus,
 we aimed to use more universally accepted, widely available, and standardized imaging
 conditions to ensure broader applicability.

**Figure**

The change in fluorescence intensity ($\Delta F/F$) of green cAMP sensors to cAMP at 450 nm
excitation in HEK293T cell lysate. n = 4 wells (gCarvi), 4 wells (G-Flamp1), 4 wells
(cAMPinG1). Tukey's post hoc following one-way ANOVA. gCarvi vs G-Flamp1: P =
1.7×10^{-9} ; G-Flamp1 vs cAMPinG1: P = 1.9×10^{-7} ; gCarvi vs cAMPinG1: P = 3.7×10^{-6}

*3. Another improvement of cAMPinG1 is the sensitivity. However, whether the enhanced*
*sensitivity is beneficial needs to be better demonstrated. The referenced EC50s of PKA was*
*determined in vitro. The value in vivo can be different. Where determination of the basal*
*cAMP concentration inside a cell has been attempted, the values are typically ~0.5–1 μ M. A*
*sensor that is too sensitive would result in elevated baseline in vivo. Indeed, the dynamic*
*range of cAMPinG1 inside an intact cell appears to be much smaller than in lysate (Fig. 1j*
*and ED Fig 3). Therefore, the sensor needs to be compared side-by-side with existing*
*sensors under the same condition in intact cells and in intact neurons in neuronal slices using*
*physiologically relevant stimulant.*

We thank the reviewer's suggestion for a side-by-side comparison of existing sensors. In
response to this comment, we have performed a side-by-side comparison of the response
of cAMPinG1-NE and G-Flamp1 to physiological stimulants in acute brain slices. We
assessed the dynamic range ($\Delta F/F$) of cAMPinG1-NE in response to dopamine local
application in the dopamine receptor D1 (D1R) expressing striatum neurons and found it
to be approximately 0.15, significantly larger than that of G-Flamp1 (~0.02). Similarly,
$\Delta F/F$ of cAMPinG1-NE in response to noradrenaline bath application in the cortex was
determined to be ~0.5, significantly larger than that of G-Flamp1 (~0.1). Therefore, these
results suggest the cAMP affinity of cAMPinG1-NE was more suited to detect cAMP
change than that of G-Flamp1 in these conditions. These results were added as Extended
Data Fig.3e-1 and described in line 123-126 in the revised manuscript.

*4. Another potential drawback of too high a sensitivity is the potential side effect of buffering.*
*The authors need to determine whether or not cAMP-sensitive neuronal properties (synaptic*
*transmission, excitability, etc) are altered by the expression of the sensor.*

We thank the reviewer's comment about the side effect of buffering. As suggested, we
have newly performed electrophysiological experiments to evaluate the side effects of
cAMPinG1-NE expression. The side effects of cAMPinG1-NE expression were not
detected in neuronal excitability and synaptic transmission. We added these results as
Extended Data Fig. 4 and described in line 126-128 in the revised manuscript.

*5. cAMPinG1 needs to be better characterized, such as its kinetics and pH sensitivity. Notably,*
 *since buffering can alter cAMP concentration and kinetics inside a cell, kinetic experiments*
 *should include measurements in intact cells, both to a puff and to a step increase of a*
 *physiological stimulant.*

We appreciate the reviewer's suggestion for further characterization of cAMPinG1
 kinetics. In response to this comment, we have conducted kinetics measurements of
 cAMPinG1-NE in acute brain slices, applying a 10-second puff of 1 μ M dopamine, a
 level that is reported within the physiological range (Iino et al., *Nature* 2020). These
 results showed that comparable rise time between cAMPinG1-NE and G-Flamp1, which
 indicated that the high cAMP affinity of cAMPinG1-NE does not introduce any buffering
 effects. These data were shown as Extended data Fig. 3g.

Also, we would like to respectfully point out that 'a step increase of a
 physiological stimulant' can include various patterns, including frequency and intensity
 required to determine what can be judged as physiological. To systematically investigate
 such diverse patterns, considering the varied physiological contexts and defining
 appropriate physiological levels and responses would be necessary for further research.
 We fully understand that the cAMP dynamics can be affected by multiple factors,
 including concentration and duration of the stimulants, potential receptor desensitization,
 and expression patterns of receptors and PDEs. Indeed, persistent activation of D1R
 through local application of 50 μ M dopamine led to desensitization with a significantly
 slower decay (see the figure below). And we could measure responses and kinetics using
 1 μ M dopamine, which is within the physiological range. Therefore, we respectfully
 mention that analysis dissecting the relation between cAMP dynamics and a step increase
 of physiological stimulants goes beyond the scope of our technological study, which
 mainly focuses on developing new biosensors. Nonetheless, we appreciate the reviewer's
 insights and will consider them in our future work to refine our understanding of cAMP
 dynamics in response to physiological stimulants.

**Figure**

The change in fluorescence intensity ($\Delta F/F$) of cAMPinG1-NE in acute brain slices of the
striatum in response to local application of 50 μM dopamine. $n = 18$ neurons. Shaded
areas denote the SEM.

*6. In vivo cAMP imaging has been achieved (Oe et al, 2020; Massengill et al, 2022; Wang et*
*al, 2022). This should be clearly described in the introduction. Additionally, it is the authors'*
*burden to demonstrate that cAMPinG1 performs better than previous sensors to a degree*
*that crosses the high bar of Nature Methods. At this time, there is essentially no comparison*
*in vivo.*

We thank the reviewer's suggestion. In response to this suggestion, we have incorporated
previous key studies on *in vivo* cAMP imaging. These include the specific references
suggested by the reviewer: Harada et al., *Scientific Reports* 2017; Oe et al., *Nature*
*Communications* 2020; Zhang et al., *Nature* 2021; Wang et al., *Nature Communications*
2022; Massengill et al., *Nature Methods* 2022. These have been cited in line 43 as
reference 4 through 8 in the revised manuscript.

Furthermore, as the reviewer suggested, we have conducted *in vivo* comparison
experiments. The results are shown as Fig. 2d,e, Extended data Fig. 7f-i, Extended data
Fig. 11a,b, Extended data Fig. 12. These findings demonstrate that cAMPinG1
significantly outperforms existing sensors in *in vivo* application.

*7. The authors utilize both cAMPinG1-NE and cAMPinG1-ST in vivo and in vitro. How are*
*these variants compared to the parental sensor? This is important given that cAMPinG1*
*achieves only 5% of the dynamic range (ED Fig. 3) under very strong conditions compared*
*to Fig. 1e. Also, are there example images showing the subcellular distribution of the ST-*
*variant?*

We appreciate the reviewer's interest about the comparative analysis of the cAMPinG1
parent sensors. To add this, we have performed *in vitro* characterization of cAMPinG1-
NE. Our analysis indicates that cAMPinG1-NE exhibits a dynamic range, cAMP affinity,
and a Hill coefficient that are comparable to those of cAMPinG1. These results were
added as Extended Data Fig. 2e and described in line 110-111 in the revised manuscript.
We also noticed the reviewer's remark about the dynamic range of cAMPinG1-NE in
response to strong stimuli (bath application of forskolin and IBMX) being minimal at 5%.
We would like to respectfully clarify that the actual rate of fluorescent change is
approximately 50%, which we do not consider an insignificant value. Regrettably, in the

case of cAMPinG1-ST, despite our rigorous efforts, we could not accomplish *in vitro*
characterization, potentially due to its lower concentration arising from soma-targeting
signals compared to other parental sensors.

We have also included an example image of cAMPinG1-ST co-expressed with
RCaMP3 as Extended Data Fig. 7a. This image clearly illustrates that the fluorescent
signal is predominantly localized to the soma, minimizing contamination between
neurons caused by their neuropil. The distinct localization is particularly apparent when
compared to the localization of cAMPinG1-NE, as shown in Extended Data Fig. 7d. We
explained the detail in line 191-193 in the revised manuscript.

*8. The authors state that they added mutations to jRGECO1a to blue-shift the excitation*
*spectra, generate a larger dynamic range and less aggregation. The aggregation data should*
*be shown and quantified. Overall, the improvement as shown in Fig. 3a-e, and ED Fig. 4*
*seems to be moderate for the bar of Nature Methods.*

We thank the reviewer's suggestion about quantitative analysis of the aggregation
tendency of our calcium indicator. As suggested, we have assessed the aggregation
tendency of RCaMP3 in comparison to jRGECO1a. We injected AAV into the primary
visual cortex to introduce calcium sensors. Four weeks after the injection, we performed
quantification of fixed brain tissue. Our results revealed that the aggregation tendency of
RCaMP3 is comparable to that of jRGECO1a. These results were added as Extended Data
Fig. 6a,b and was described in line 174-175 in the revised manuscript.

*9. As the field comes to realize, RS20 instead of M13 is used in RGECOs and GCaMPs. Also,*
*since the improvement on calcium sensor is on RGECO, why naming the new sensor*
*RCaMP3? RCaMPs already exist, and they are based on different calmodulin binding peptide*
*and different fluorophore. Naming this new variant to RCaMP3 will cause a lot of confusion*
*to the users. Please consider a name along the line of RGECO.*

To address the reviewer's concern, we have replaced 'M13' with 'RS20' in the revised
manuscript (line 353 and 913) and Fig. 3a and Extended Data Fig. 5a. We also fully
recognize the confusion arising from the naming rules of the red calcium indicator in this
field. In terms of primary structure, our red calcium indicator is composed of RS20,
cpmApple, Calmodulin (CaM), and F2A, which aligns with the structure of R-CaMP1.07
(Ohkuma et al., *Plos One* 2013). The F2A incorporation is known not only for enhancing
cytoplasmic localization but also for increasing the Ca²⁺-dependent fluorescence changes.

Thus, despite the numerous mutations introduced, our sensor stands as the closest analog
to R-CaMP1.07 in terms of the primary structure. We note that R-CaMP2, the initial
successor to R-CaMP1.07, has already been developed by Dr. Haruhiko Bito's group
(Inoue et al., *Nature Methods* 2015). Furthermore, our calcium indicator significantly
outperforms R-CaMP2 in sensitivity, as it shows a fluorescence change that is
significantly superior to that of XCaMP-R, the successor to R-CaMP2. Therefore, we
named 'RCaMP3' to emphasize the improvement over predecessors. We believe that this
name is concise, indicative, and can efficiently emphasize the advancement of the red
calcium indicator. Also, to clarify that the primary structure of RCaMP3 is closest to
RCaMP1.07, we have included the primary structure of R-CaMP1.07 as Extended Data
Fig. 5a.

*10. Since many mutations are introduced to the fluorescent protein in the new calcium sensor,*
*whether the photostability and pH sensitivity have changed should be tested.*

We thank the reviewer's helpful comment about the photostability and pH sensitivity. In
response to the reviewer's comment, we have newly assessed the photostability and pH
sensitivity of RCaMP3, finding them to be comparable with those of jRGECO1a. These
new results are added as Extended Data Fig. 5j-l and described in line 160-162 in the
revised manuscript.

*11. How the new calcium sensor responds to different number of action potentials should be*
*presented and compared to the existing sensors.*

We thank the reviewer for this comment. In response to this comment, we conducted a
quantitative analysis to assess the relation between the fluorescent changes and the
number of spikes in RCaMP3 and jRGECO1a in L2/3 pyramidal neurons, utilizing
simultaneous calcium imaging with loose-seal cell-attached recording *in vivo*. The
fluorescence response of both RCaMP3 and jRGECO1a demonstrated a linear correlation
with the spike number. Notably, the peak amplitude of RCaMP3 was consistently higher
than that of jRGECO1a across all spike numbers. These results are summarized as Fig.
3n and described in line 171-174 in the revised manuscript.

*12. Fig. 3d-h: Any explanation why jRGECO1a's response is only about 10% of the literature?*
*Also, Although the new calcium sensor gives larger response, its noise also seems to be*
*higher (although to a less degree than the signal). Why?*

We appreciate the reviewer's careful observations. This experiment was performed on
acute brain slices, using an LED as the excitation light source. Contrary to two-photon
imaging, this approach resulted in considerable light scattering, leading to intense noise,
including fluorescence contamination from non-targeted neurons. This interference could
potentially mask the fluorescent changes of the calcium sensors.

To rectify the contamination issue, we conducted a detailed analysis of the
fluorescence change in response to single action potentials *in vivo* by utilizing two-photon
microscopy with loose-seal cell-attached recording. Again, the amplitude of fluorescence
change in RCaMP3 was significantly larger than in jRGECO1a, though the kinetics is
comparable. These results were added as Fig. 3k-n.

*13. Under 2p excitation for the cAMP sensor, how much contamination of the signal from the*
*calcium sensor are there in the cAMP signal channel? One limitation of nearly all red*
*fluorescent protein is that they contain a green component due to bifurcating maturation. This*
*needs to be quantified for multiplex imaging.*

We thank the reviewer for highlighting this point. To evaluate the potential signal
contamination from the red calcium sensor to the green channel, we conducted *in vivo*
two-photon two-color imaging with soma-targeted cAMPinG1 mutant (cAMPinG1mut-
ST) and RCaMP3. We observed fluorescence changes in RCaMP3 in response to visual
stimuli, whereas no response was observed in cAMPinG1mut-ST. Therefore, under our
experimental conditions, we found no indication of leakage from the RCaMP3 signal into
the green (cAMPinG1) channel. This result was added as Extended Data Fig.11c and
described in line 229-230 in the revised manuscript.

*14. For in vivo experiments, both the cAMP and calcium sensor response should be*
*compared with existing sensors to judgment whether the new sensors present a true*
*improvement and what is the degree of the improvement. Relevant figures include Fig. 2, Fig.*
*3j, Fig. 4, and Fig. 5.*

We appreciate the reviewer's comment. To address this, we performed the following
additional experiments to compare with existing sensors and judge the superiority of our
new sensors.

Regarding cAMP sensors, we conducted a comparative analysis of cAMPinG1
and G-Flamp1 *in vivo* by utilizing two-photon microscopy and fiber photometry. We

observed that cAMPinG1 provided a clearer depiction of not only the elevation of cAMP
induced by physical stimuli but also subsequent suppression post-stimulation than G-
Flamp1. These new results were added as Fig. 2d,e, Extended Data Fig. 7f-i, Extended
Data Fig. 11a,b, and Extended Data Fig. 12, and described in line 137-138, 201-203, 227-
229, and 249-251 in the revised manuscript.

Regarding calcium sensors, we conducted a comparative analysis of RCaMP3
and jRGECO1a by utilizing *in vivo* two-photon imaging together with loose-seal cell-
attached recording. Our results revealed that the dynamic range of RCaMP3 to report
single action potentials is significantly superior to those of jRGECO1a, although the
kinetics are comparable. Furthermore, we performed a quantitative analysis to assess the
relation between the fluorescent changes and the number of spikes in RCaMP3 and
jRGECO1a. Notably, the peak amplitude of RCaMP3 was consistently higher than that
of jRGECO1a across all spike numbers. These new results were summarized as Fig. 3i-n
and described in line 171-174 in the revised manuscript.

*15. Pharmacological manipulation should be included for in vivo experiments to demonstrate*
*the cAMP response and to dissect the potential pathways controlling its activity.*

We agree with the reviewer's suggestion about the importance of dissecting the potential
pathways controlling cAMP. To address this, we conducted *in vivo* cAMP imaging with
fiber photometry with cAMPinG1-NE in the primary visual cortex combined with
pharmacological manipulation to reveal the upstream pathways of cAMP during forced
running. Our results showed that the administration of propranolol, a well-known β -
blocker, significantly reduced the increase in cAMP level observed during forced running.
These findings indicate that the cAMP elevation in the primary visual cortex depended
on β adrenoceptors under these conditions. These new results were added as Extended
Data Fig. 8 and described in line 198-199 in the revised manuscript.

*16. cAMP binding mutant should be used in Fig. 4 and 5 to verify that the sensor response*
*is specific.*

We appreciate the reviewer's suggestion about the use of the cAMPing binding mutant in
*in vivo* experiment. As suggested, we performed *in vivo* two-photon imaging using soma-
targeted cAMPinG1 mutant (cAMPinG1mut-ST) to examine the responses to forced
running and visual stimuli. Our observation revealed that the cAMPinG1 mutant did not
exhibit any response to these stimuli. These data show that the response of cAMPinG1 is

consequent to cAMP binding. These results were added as Extended Data Fig. 7f-i and
Extended Data Fig. 11a,b and described in line 201-203 and 227-228 in the revised
manuscript.

*17. Fig. 6: Couldn't the experiment be done with existing ratiometric cAMP sensors?*

We appreciate the reviewer's comment. To address this, we conducted single-timepoint,
ratiometric imaging of cAMPinG1, G-Flamp1, and cAMPFIRE-L, which is one of the
best FRET-type cAMP indicators (Massengill et al., *Nature Methods* 2022). When we
compare the relative change ($\Delta R/R_0$) in fluorescence ratio upon forskolin stimulation,
cAMPinG1 exhibited superior performance. These findings confirm that cAMPinG1 is
advantageous for both intensimetric and ratiometric applications *in vitro*. These results
are added as Extended Data Fig. 13a-c and described in line 257-259 in the revised
manuscript.

**Additional comments**

*1. For measurements of cAMP affinity, the methods on page 21 (lines 361-366) suggest that*
*the supernatant containing the sensors and cAMP was diluted to reduce the cAMP*
*concentration. This would also reduce the sensor concentration, making comparison*
*between cAMP concentrations inaccurate. Please either redo the experiment or rewrite the*
*text so that it is clear that the sensor concentration was not held constant.*

We apologize for our confusing description. Sensor concentration was constant among all
wells because we applied the same amount of lysate to the solution which included
different concentrations of cAMP. We have corrected this part in the revised manuscript
(line 390-395).

*2. Fig. 3j, 4 and 5, example images before and after stimulation with sufficient magnification*
*should be presented.*

We acknowledge the reviewer's suggestion. In response to this, we added magnified
example images in Fig. 3j, Extended Data Fig. 7b,d, and Extended Data Fig. 10a.

*3. ED Fig. 2e is not referenced in text. Also, a quantification would strengthen the experiment.*

We thank the reviewer's careful observation. In response to this, we mentioned about it
in the result section in line 111-112 in the revised manuscript. We also added the
quantification result as Extended Data Fig. 2j.

*4. Fig. 5e: Within the same cell, is there correlation between calcium amplitude and*
*cAMPinG1 signal?*

We thank the reviewer for the comment about the correlation between Ca^{2+} and cAMP
signal during the visual stimuli. The results presented in Fig. 5c,d aim to be
straightforward and elucidative. They display the relative intensity of Ca^{2+} and cAMP
signal within the same neurons when subjected to drifting gratings. These results
highlighted that both RCaMP3 and cAMP exhibit strong responses to certain visual
stimulus orientations. Conversely, for stimuli that lack orientation selectivity, both
RCaMP3 and cAMP showed weak responses. Therefore, these results suggest a positive
correlation between calcium response and cAMP signal within the same neuron.

*5. Fig. 5j: For ChRmine elicited cAMP response, was it elicited by a single 4-s spiral*
*stimulation? Please clarify in text. Also, how many action potentials does this corresponding*
*to?*

We apologize for our unclear description. We have changed "4-second spiral scanning"
to "4-second, 16 Hz spiral scanning" in the Methods section in the revised manuscript
(line 749-750).

We also appreciate the reviewer's question. We acknowledge and agree with the
reviewer on the understanding of the number of induced action potentials in this condition.
However, our primary focus in this study was directed towards the characterization and
comparative analysis of our new indicators. Therefore, we respectfully state that a
detailed analysis exploring the relation between physiological stimulants and cAMP
dynamics is beyond the scope of our technological development.

*6. "G-Flamp1 limited the number of cells that could be imaged and quantified cAMP dynamics*
*in the mouse cortex". This need to be justified for cited.*

We thank the reviewer's comment. To respond to this point, we have conducted a
comparative analysis of cAMPinG1 and G-Flamp1 *in vivo* by utilizing two-photon
microscopy. We observed that cAMPinG1 provided a clearer depiction of not only the

elevation of cAMP induced by physical stimuli but also subsequent suppression post-
stimulation than G-Flamp1, possibly due to its higher affinity for cAMP. These results
suggest that cAMPinG1 is suitable for more accurate quantification of cAMP dynamics
in the mouse cortex. These results were added as Fig. 2d,e, Extended Data Fig. 7f-i,
Extended Data Fig. 11a,b.

Furthermore, we also examined the fluorescent intensities of these two sensors
and found that the fluorescence intensity of G-Flamp1 is very low level, being only one-
tenth the brightness of cAMPinG1. These results were shown as Extended Data Fig. 7d,e,
and described in line 199-201. The fluorescence difference also diminished the number
of neurons available for imaging and quantification of cAMP dynamics, highlighting the
limitations of G-Flamp1 for *in vivo* application.

*7. The reviewer disagrees with this assertion: "which made fast imaging or combined use*
*with other sensors, such as Ca²⁺ indicators, generally challenging due to the required optical*
*settings".*

We appreciate the reviewer's comment to correct our statement. In response, we have now
deleted this sentence.

**Minor comments**

*1. Page 7 line 119: The corresponding figure (Extended Data Fig. 3) shows application of*
*forskolin + IBMX, but text reads only forskolin.*

We apologize for our mistake for missing the experimental condition. In response to this
comment, we have changed "forskolin/IBMX" in the revised manuscript (line 122).

*2. Page 9 line 139: "one of the upstream of cAMP signaling" should read "one of the upstream*
*modulators of cAMP signaling". Similarly, page 11, Line 187 should read "multiple upstream*
*pathways, including neuromodulators". Also page 12, line 210.*

We appreciate the reviewer's suggestion. As suggested, we have made the recommended
corrections in the revised manuscript (line 148, 208, and 233).

*3. Page 14 last paragraph: Fig. 6a,b shows dopamine application but this is not discussed in*
*the text.*

We thank the reviewer for pointing it out. In the absence of dopamine, there was a
correlation between the fluorescence intensities of cAMPinG1 and RFP. However, in the
presence of dopamine, the fluorescence intensity of cAMPinG1 increases, and no such
correlation between the fluorescence intensities of cAMPinG1 and RFP is observed. We
described this point in the revised manuscript (line 267-269).

*4. Page 15 first paragraph. Please define TFX and ADRB2 in the text.*

We apologize for our mistakes regarding abbreviations. In Fig. 6, the TFX has been
revised to 'transfection.' (line 1058, and 1064) Also, ADRB2 has been defined as 'Gs-
coupling β 2 adrenoceptor' in the revised manuscript (line 270-271).

*5. Fig. 1e and 3b: please specify the used cAMP or calcium concentration in text or legend.*

We apologize for our mistakes for missing experimental conditions. In Fig. 1b, we added
the cAMP concentration (300 μ M) to the figure legend (line 921). In Fig. 3b, we also
added calcium concentration (3.9 μ M) to the figure legends (line 964).

*6. Page 31 line 550: Is 18 hours correct here? That seems very soon after surgery. It seems
unlikely that inflammation would be reduced enough to accurately see cells.*

We appreciate the reviewer's careful observation. We understand the reviewer's concern
regarding the timing of imaging after the surgery. However, our previous studies have
shown that *in vivo* imaging is possible even after immediate surgery if the surgery
procedure is performed properly (Bando et al., *Cell Reports* 2019). Our imaging movie
(Supplementary Video 2) and fluorescence traces of neurons further support that imaging
from numerous neurons simultaneously and with sufficient sensitivity. Given these
considerations, we believe that the impact of inflammation is negligible, and conducting
imaging 18 hours post-surgery does not introduce significant distortions or artifacts to our
results.

*7. ED Fig. 2d: please include cAMPinG1 cAMP concentration response curve on this plot as
a comparison of cAMP vs cGMP sensitivity*

We thank the reviewer's suggestion. As suggested, we have generated a new graph to
compare the sensitivity of cAMPinG1 and Flamindo2 to cAMP and cGMP as Extended
Data Fig. 2d.

8. Supplementary Table 1 should read Massengill et al., instead of Crystian et al.

We apologize for our mistake. We now have corrected the reference in Supplementary
Table 1.

9. cAMPinG1 exhibits a cAMP binding hill coefficient of 1. Any explanations given that there
are two binding sites?

We thank the reviewer for asking intriguing question. When considering genetically
encoded calcium indicators, they predominantly utilize a calmodulin domain which has
four Ca²⁺-binding regions. The Hill coefficients of recently developed fast green calcium
indicators are 2.1 for jGCaMP8f (Zhang et al., *Nature* 2023), 3.1 for jGCaMP7f (Dana et
al., *Nature Methods* 2019), and 1.3 for XCaMP-Gf (Inoue et al., *Cell* 2019). The
variability in these coefficients implies that there is no direct correlation between the
number of Ca²⁺-binding sites and the linearity of Ca²⁺-dependent fluorescence changes.

This lack of direct correlation seems to be analogous for cAMP indicators where
the number of cAMP binding sites probably does not correlate with the Hill coefficient.
However, if we consider the cAMP binding sites of cAMPinG1 specifically, a possible
explanation for cAMPinG1's behavior is that the cAMP affinity of cAMP-binding domain
B (CBD-B) is more than 40-fold higher than that of CBD-A (Lorenz et al., *Biochemical
Journal* 2017), suggesting that the structural change of CBD-B has less effect on that of
CBD-A and subsequent fluorescence change.

Again, we appreciate the reviewer's careful reading and providing constructive comments.

**Reviewer #3:**

First, we thank the reviewer for positive and constructive comments and suggestions.

**Major comments:**

1. The authors included a comprehensive *in vitro* characterization of both indicators. However,
 they didn't include the kinetic characterization of cAMPinG1 and its benchmarking
 comparison with other existing cAMP indicators.

We thank the reviewer for pointing it out. To address this comment, we performed kinetics
 measurements for cAMPinG1 utilizing a microperfusion system as previously described
 (Kawata et al., *PNAS* 2022). The apparent kinetic parameters of cAMPinG1 were $k_{on} =$
 $8.0 \times 10^6 \text{ M}^{-1} \text{ s}^{-1}$ and $k_{off} = 1.6 \text{ s}^{-1}$, corresponding to a binding half-rise time of 0.04 s to
 2 μM cAMP and unbinding half-decay time of 0.44 s. This finding indicates that kinetics
 of cAMPinG1 is not slower compared to other existing cAMP indicators, such as gCarvi
 ($k_{on} = 1.38 \times 10^6 \text{ M}^{-1} \text{ s}^{-1}$, $k_{off} = 3.31 \text{ s}^{-1}$, as per literature values, Kawata et al., *PNAS* 2022)
 and G-Flamp1 ($k_{on} = 3.48 \times 10^6 \text{ M}^{-1} \text{ s}^{-1}$, $k_{off} = 7.9 \text{ s}^{-1}$, as per literature values, Wang et al.,
 *Nature Communications* 2022).

In a separate experiment, we also evaluated the kinetics of the cAMP sensors in
 response to the local application of dopamine to striatum neurons in acute brain slices.
 The result demonstrated that the rise kinetics ($t_{1/2}$) of cAMPinG1-NE was 2.9 s, which is
 comparable to that of G-Flamp1 ($t_{1/2} = 2.4\text{s}$).

These results were added as Extended Data Fig. 2g,h and Extended Data Fig. 3g
 and were described in line 111 in the revised manuscript.

2. The authors grafted several known amino acid substitutions from R-GECO1.2 onto
 jRGECO1a. Among those mutations, M128R/I131V/V139L/F188L/N231S/S234T blue shift
 the excitation spectra. I'm a little bit confused about the rationale of making the RCaMP3
 action spectra more blue-shifted. How would this make RCaMP3 a better red indicator in
 concurrent green cAMP imaging?

The majority of utilized commercial dual-wavelength lasers, such as Coherent's
 Chameleon Discovery and Spectra-Physics' DeepSee Insight, are designed to operate at
 a fixed wavelength of approximately 1,040 nm. In contrast, the two-photon excitation
 peak of jRGECO1a is at around 1,060 – 1,070 nm, which shows a mismatch laser
 wavelength (Dana et al., *Elife* 2016, Figure 2—figure supplement 1). Therefore, the blue-
 shifted excitation spectrum of RCaMP3 is well-suited for two-photon imaging. We added
 this explanation in the revised manuscript (line 162-165).

The use of blue-shifted red fluorescent indicators may not be optimal for dual-
 color imaging with green fluorescent indicators due to the potential risk of signal
 contamination. To assess this possibility, we conducted *in vivo* two-photon two-color

imaging with soma-targeted cAMPinG1 mutant (cAMPinG1mut-ST) and RCaMP3. We
observed fluorescence changes in RCaMP3 in response to visual stimuli, whereas no
response was observed in cAMPinG1mut-ST. Therefore, under our experimental
conditions, we found no indication of signal leakage from RCaMP3 signals into the
cAMPinG1 signal. Rather, there is an advantage in detecting the RCaMP3 fluorescence
signal with a high signal-to-noise ratio. This result was added as Extended Data Fig. 11c
and described in line 229-230 in the revised manuscript.

*3. Based on the results, the half-decay time of cAMPinG1 is more than 100 times larger than*
*that of RCaMP3. Is this number comparable to the physiological facts? If not, how would it*
*impact its temporal resolution in applications?*

We thank the reviewer's insightful observations and concern regarding the distinct
differences in the half-decay time between cAMPinG1 and RCaMP3. Our results showed
that the half-decay time of cAMPinG1 was on the order of tens of seconds in Fig. 2f,
which is more than 100 times larger than that of RCaMP3. This finding is consistent with
previous studies (Wang et al., *Nature Communications* 2022; Massengill et al., *Nature*
*Methods* 2022), although they did not quantify the decay time. Given the consistency with
previous studies, we are confident that the identified half-decay time of cAMPinG1 is
indeed representative of physiological conditions.

**Minor comments:**

*1. The authors stated that Two-photon dual-color imaging for Ca²⁺ and cAMP revealed a*
*strong correlation between cell-specific cAMP increases and Ca²⁺ transients. However,*
*there is no statistical analysis supporting this statement. The authors should consider*
*including a temporal pairwise correlation between the cAMP trace and Ca trace within the*
*same neuron.*

We thank the reviewer for the constructive suggestion. The results presented in Fig. 5d is
intended to show the correlation between Ca²⁺ and cAMP signal within the same neurons
when subjected to drifting gratings. These results elucidate that both RCaMP3 and
cAMPinG1 exhibit a strong response to visual stimuli, demonstrating orientation
selectivity. Conversely, a weak response is observed for stimuli lacking this selectivity
from both RCaMP3 and cAMPinG1 measurements. Therefore, these results imply a
positive correlation between the Ca²⁺ response and the cAMP signal within each neuron.
Furthermore, in response to the reviewer's suggestion, we conducted a statistical analysis

of the fluorescent intensity of RCaMP3 and cAMPinG1 in response to the stimuli in the
 180° direction opposite the preferred one (see the figure below). Neurons showing a
 direction selectivity index (DSI) below 0.4 based on Ca²⁺ transients demonstrated a larger
 change in fluorescence intensity of cAMPinG1 compared to those with DSI above 0.4.
 This additional analysis further supports the observed strong correlation between cAMP
 increases and Ca²⁺ transients.

**Figure**
 The change in fluorescence intensity ($\Delta F/F$) of RCaMP3 (magenta) and cAMPinG1
 (green) in response to the stimuli in a direction 180° opposite of the preferred one.
 Neurons showing a direction selectivity index (DSI) < 0.4 in Ca²⁺ response (n = 94 cells
 in 3 mice) and neurons showing a direction selectivity index (DSI) \geq 0.4 in Ca²⁺ response
 (n = 101 cells in 3 mice). Unpaired two-tailed t-test. $P = 3.0 \times 10^{-5}$. Shaded areas denote
 the SEM.
 Again, we appreciate the reviewer's helpful and constructive comments.

Decision Letter, first revision:

Dear Dr. Sakamoto,

Thank you for submitting your revised manuscript "A multicolor suite for deciphering population coding in calcium and cAMP *in vivo*" (NMETH-A51721A). It has now been seen by the original referees and their comments are below. The reviewers find that the paper has improved in revision, and therefore we'll be happy in principle to publish it in Nature Methods, pending minor revisions to satisfy the referees' final requests and to comply with our editorial and formatting guidelines.

When resubmitting the manuscript, please ensure to include a rebuttal and to discuss or better explain the relevant remaining issues in the manuscript.

TRANSPARENT PEER REVIEW

ORCID

Best regards,
Nina

Nina Vogt, PhD
Senior Editor
Nature Methods

Reviewer #1 (Remarks to the Author):

The authors have fully addressed my concerns/comments and provide nice additional characterization of cAMP sensors (pH, lifetime, on/off kinetics) that will be valuable to future users of this sensor.

Reviewer #2 (Remarks to the Author):

The authors did a remarkable job in addressing the concerns and improving the manuscript. The reviewer, however, has remaining concerns regarding potential buffering effects, which appear to be evident in the included data.

Major comments.

1. Buffering effect due to the high affinity of cAMPinG1 sensor remains to be a concern. In Fig. 1j, ED Fig. 3c and ED Fig. 3k, in response to the very strong stimulation of forskolin (or to norepinephrine), the responses peak in several minutes. This is much slower than known kinetics of cAMP accumulation in the corresponding cells under these conditions. The reviewer thinks that this slowness is because the expressed sensor buffers cAMP to a degree that endogenous cAMP production takes a lot time to keep up with. This is corroborated by the results that the response amplitudes in HEK cells (Fig. 1j) are only ~40% of the maximal dynamic range and are ~10-fold smaller in neurons (ED Fig. 3). This is explained by that neuron has higher basal cAMP concentrations, and the sensor has been largely saturated by basal level of cAMP. Are there alternative explanations?

Please note that the seemingly faster kinetics in striatal neurons is not an evidence against the point above because cAMP production would desensitize in the striatal neurons even in the presence of a constant concentration of dopamine.

2. ED Fig. 4: Are the properties in the examined neuronal types and under the experimental conditions known to be cAMP dependent? Can the authors test a cAMP-dependent change to a non-saturating cAMP manipulation (e.g., via the application of a moderate concentration of norepinephrine)?

3. There is no comparison between cAMPinG1 and GFlamp1 in HEK cells. This is important for evaluating that whether cAMPinG1's advantage over GFlamp1 is across all the systems, versus experiment dependent.

Additional comments.

1. Fig. 1e, the reviewer maintains the opinion that, for comparing the dynamic range, each sensor should be tested at its optimal condition. If the authors insist to keep Fig. 1e as it is, then the 450 nm excitation data should be shown immediately adjacent either in the same panel, or as an independent panel.

2. The above opinion also applies to Fig. 3c. Also, for this figure, how is the variability in expression controlled (some plasmids express better than others)?

3. Lines 53-54: it should be point out that larger dynamic range is achieved either using ratiometric excitation, or using a wavelength that is less ideal for the alternative sensor.

Minor comments.

1. In many places in the main text, the experiment conditions need to be better described. For example, lines 126-128: what type of neuron? Acute slices versus, cultured slices, vs primary neuronal culture? Viral infected vs IUE? There are many similar situations throughout the text that need clarification. This will greatly enhance the readability by saving time to go deep into Methods.

2. ED Fig. 12: If ratio is used, then the y labels ($\Delta F/F$) is not correct. Should be " $\Delta R/R$ ".

3. Line 1118: the reference page number is not correct.

Reviewer #3 (Remarks to the Author):

The authors have addressed all my major and minor comments.

Author Rebuttal, first revision:

**Response to the reviewers**

We would like to thank all reviewers for their careful reading and constructive comments
on our manuscript. All comments are valuable and helpful for revising and improving our
manuscript. We describe in detail the changes in the following point-by-point response.
For clarity, the received comments are represented in *italic Arial* fonts, while our
responses are shown in Times fonts. All changes in the revised manuscript are highlighted
in yellow.

**Reviewer #1**

*The authors have fully addressed my concerns/comments and provide nice additional*
*characterization of cAMP sensors (pH, lifetime, on/off kinetics) that will be valuable to future*
*users of this sensor.*

We wish to express our appreciation to the reviewer for insightful comments, which have
helped us significantly improve our paper.

**Reviewer #2**

First of all, we thank the reviewer for careful reading and providing many constructive
comments and suggestions.

**Major comments.**

*1. Buffering effect due to the high affinity of cAMPinG1 sensor remains to be a concern. In*
*Fig. 1j, ED Fig. 3c and ED Fig. 3k, in response to the very strong stimulation of forskolin (or*
*to norepinephrine), the responses peak in several minutes. This is much slower than known*
*kinetics of cAMP accumulation in the corresponding cells under these conditions. The*
*reviewer thinks that this slowness is because the expressed sensor buffers cAMP to a degree*
*that endogenous cAMP production takes a lot time to keep up with. This is corroborated by*
*the results that the response amplitudes in HEK cells (Fig. 1j) are only ~40% of the maximal*
*dynamic range and are ~10-fold smaller in neurons (ED Fig. 3). This is explained by that*
*neuron has higher basal cAMP concentrations, and the sensor has been largely saturated*
*by basal level of cAMP. Are there alternative explanations? Please note that the seemingly*
*faster kinetics in striatal neurons is not an evidence against the point above because cAMP*

*production would desensitize in the striatal neurons even in the presence of a constant*
*concentration of dopamine.*

We appreciate the reviewer's comment regarding the buffering effect. Referring to the G-
Flamp1 study (Wang et al., *Nature Communications* 2022, Figure 2c), it was observed
that the half-rise time under forskolin application was approximately 3-4 minutes in the
HEK293T cells. This response time is closely aligned with that of cAMPinG1 under
similar conditions. Moreover, the half rise time in response to 1 μ M dopamine application
is also comparable, at approximately 1.4 seconds in both cAMPinG1 and G-Flamp1
(Extend Data Fig.3g). Therefore, cAMPinG1 kinetics is not exceptionally slow due to the
buffering effects, but rather comparable range, supporting the validity of cAMPinG1
kinetics.

We acknowledge the possibility that the cAMP indicator may be coupled with
basal cAMP concentrations *in vivo*. However, we also respectfully point out that this is
not the sole factor contributing to the variance in response amplitudes. These differences
can also be attributed to differences in the optical systems (one-photon vs two-photon),
as well as to light scattering and associated noise in cultured cells and acute brain slices.
Therefore, we believe that a direct comparison of the maximal dynamic range between
HEK293T cells and acute brain slices is not reasonable. Furthermore, it should be
highlighted that in side-by-side comparisons, cAMPinG1 demonstrates superior
performance compared to G-Flamp1 in both acute brain slices and living animals (Figure
2a-e, Extend Data Fig.3g-l, Extend Data Fig.6a-d, Extend Data Fig.7c,d, Extend Data
Fig.8).

Nonetheless, the possibility of buffering effect remains a consideration with
existing cAMP biosensors. To assist future cAMPinG1 users, we specifically highlighted
this aspect in the discussion section in the revised manuscript (line 239).

*2. ED Fig. 4: Are the properties in the examined neuronal types and under the experimental*
*conditions known to be cAMP dependent? Can the authors test a cAMP-dependent change*
*to a non-saturating cAMP manipulation (e.g., via the application of a moderate concentration*
*of norepinephrine)?*

We thank the reviewer's comment. Previous studies have already demonstrated that the
elevation of intracellular cAMP levels induced by forskolin and norepinephrine affects
the firing rate of neurons (Cudmore and Turrigiano., *Journal of Neurophysiology* 2004;
Joseph et al., *Journal of Neurophysiology* 2007; Mueller et al., *Journal of Neuroscience*

2008). Based on these studies, we expect a cAMP-dependent change in excitability and
synaptic transmission under our experimental conditions.

In addition, we would like to respectfully point out that the application of a
moderate concentration of norepinephrine involves significant challenges. This is due to
the necessity of careful consideration of various parameters is essential to select the most
appropriate protocol, including the concentration and application methods. It is also
possible that these parameters might not be consistent across various brain regions and
cell types. Thus, we believe it extends beyond the scope of our technological study, which
focuses on developing new biosensors.

*3. There is no comparison between cAMPinG1 and GFlamp1 in HEK cells. This is important*
*for evaluating that whether cAMPinG1's advantage over GFlamp1 is across all the systems,*
*versus experiment dependent.*

We have already conducted a comparison between cAMPinG1 and G-Flamp1 in
HEK293T cells as shown in Extended Data Fig.9a,c. We performed ratiometric imaging
by measuring the ratio of excitation at 488 nm and 405 nm following forskolin application
in HEK293T cells. These results clearly demonstrated the advantage of cAMPinG1 over
G-Flamp1. Based on this result, we conclude that cAMPinG1 is superior to GFlamp1 in
HEK293T cells, at least in the context of ratiometric imaging.

**Additional comments.**

*1. Fig. 1e, the reviewer maintains the opinion that, for comparing the dynamic range, each*
*sensor should be tested at its optimal condition. If the authors insist to keep Fig. 1e as it is,*
*then the 450 nm excitation data should be shown immediately adjacent either in the same*
*panel, or as an independent panel.*

We appreciate the reviewer's comment. As suggested, we added 450 nm excitation data
as Fig.1e and described the result in the revised manuscript (line 86-88).

*2. The above opinion also applies to Fig. 3c. Also, for this figure, how is the variability in*
*expression controlled (some plasmids express better than others)?*

We thank the reviewer's comment. We measured the fluorescent intensities of jRGECO1a
and RCaMP3 in HEK293T cells following ionomycin application by 1,040 nm and 1,070
107 nm excitation. The two-photon excitation peak of jRGECO1a is reported to be around

108 1,060-1,070 nm (Dana et al., *Elife* 2016, Figure 2—figure supplement 1). When
 comparing fluorescence intensity ratios at the two wavelengths, RCaMP3 demonstrated
 a significantly higher ratio than jRGECO1a. This result suggests that blue-shifted
 RCaMP3 is more suitable for 1040 nm excitation. Also, for consistency throughout the
 experiment, we used the same promoter (CAG) and backbone vector for evaluation.

**Figure**

The ratio of fluorescence intensities of red Ca²⁺ indicators at 1,040 nm and 1,070 nm
 excitation in live HEK293T cells in the presence of ionomycin application. The ratio of
 both indicators was normalized to that of jRGECO1a. n = 285 (jRGECO1a), n = 220
 (RCaMP3) cells. Unpaired two-tailed t-test. P = 2.9 × 10⁻¹⁰. Error bars denote SEM.

*3. Lines 53-54: it should be point out that larger dynamic range is achieved either using*
 *ratiometric excitation, or using a wavelength that is less ideal for the alternative sensor.*

Our primary goal is to develop a practical tool for *in vivo* applications. Comparative
 analysis between cAMPinG1 and G-Flamp1, particularly in acute brain slices and living
 animals, has clearly demonstrated that cAMPinG1 exhibits significantly larger dynamic
 ranges in response to drugs and physical stimuli compared to G-Flamp1. Notably, the
 orientation-selective increase and subsequent decrease in cAMP level in the primary
 visual cortex, which are evident with cAMPinG1, are not apparently detected with G-
 Flamp1 (Extend Data Fig. 7c,d). This contrast is highlighted to emphasize the superior *in*
 *vivo* performance of cAMPinG1.

Furthermore, while the dynamic range of G-Flamp1 differs significantly from
 cAMPinG1 in the *in vitro* assay at the 490 nm excitation, we do not consider that of G-
 Flamp1 to be inferior. Instead, the differences of *in vivo* imaging are primarily attributed
 to the cAMP affinity, as previously discussed in our response to the reviewer #1. Thus, in
 consideration of the reviewer's comment, we revised the original statement to: "Here, we

present cAMPinG1, a green cAMP indicator with a high dynamic range and more than 4-
136 fold higher cAMP affinity than the existing green cAMP indicators.” in the revised
manuscript (line 46-47).

**Minor comments.**

*1. In many places in the main text, the experiment conditions need to be better described.*
*For example, lines 126-128: what type of neuron? Acute slices versus, cultured slices, vs*
*primary neuronal culture? Viral infected vs IUE? There are many similar situations throughout*
*the text that need clarification. This will greatly enhance the readability by saving time to go*
*deep into Methods.*

We thank the reviewer’s comment. Due to the word count limitations, we were unable to
explain all detail of the experimental conditions in the main text. To address this, we have
endeavored to clearly depict the experimental conditions in the figures and their legends
in Extended Data Fig.3m-o.

*2. ED Fig. 12: If ratio is used, then the y labels ($\Delta F/F$) is not correct. Should be “ $\Delta R/R$ ”.*

We apologize for our mistake. As suggested, we revised the $\Delta F/F$ to $\Delta R/R$ in the Extended
Data Fig.8 and 9.

*3. Line 1118: the reference page number is not correct.*

We apologize for our mistake. We now have revised the reference in the revised
manuscript (line 491-492).

Again, we appreciate the reviewer’s careful reading and providing constructive comments.

**Reviewer #3**

*The authors have addressed all my major and minor comments.*

We thank again the reviewer for constructive comments and suggestions.

Final Decision Letter:

Dear Dr Sakamoto,

I am pleased to inform you that your Article, "A multicolor suite for deciphering population coding in calcium and cAMP *in vivo*", has now been accepted for publication in Nature Methods. The received and accepted dates will be February 10th, 23 and February 21st, 2024. This note is intended to let you know what to expect from us over the next month or so, and to let you know where to address any further questions.

Over the next few weeks, your paper will be copyedited to ensure that it conforms to Nature Methods style. Once your paper is typeset, you will receive an email with a link to choose the appropriate publishing options for your paper and our Author Services team will be in touch regarding any additional information that may be required. It is extremely important that you let us know now whether you will be difficult to contact over the next month. If this is the case, we ask that you send us the contact information (email, phone and fax) of someone who will be able to check the proofs and deal with any last-minute problems.

Please note that *Nature Methods* is a Transformative Journal (TJ). Authors may publish their research with us through the traditional subscription access route or make their paper immediately open access through payment of an article-processing charge (APC). Authors will not be required to make a final decision about access to their article until it has been accepted. Find out more about Transformative Journals

You may wish to make your media relations office aware of your accepted publication, in case they consider it appropriate to organize some internal or external publicity. Once your paper has been scheduled you will receive an email confirming the publication details. This is normally 3-4 working

days in advance of publication. If you need additional notice of the date and time of publication, please let the production team know when you receive the proof of your article to ensure there is sufficient time to coordinate. Further information on our embargo policies can be found here: <https://www.nature.com/authors/policies/embargo.html>

If you are active on Twitter/X, please e-mail me your and your coauthors' handles so that we may tag you when the paper is published.

Best regards,
Nina

Nina Vogt, PhD
Senior Editor
Nature Methods